# A small-molecule SARS-CoV-2 inhibitor targeting the membrane protein

Ellen Van Damme[1], Pravien Abeywickrema[2,21], Yanting Yin[2,21], Jiexiong Xie[1,21], Sofie Jacobs[1,21], Mandeep Kaur Mann[3], Jordi Doijen[1], Robyn Miller[2], Madison Piassek[2], Simone Marsili[4], Murali Subramanian[5,15], Leah Gottlieb[2,16], Rana Abdelnabi[6,7], Michiel Van Gool[4], Nick Van den Broeck[8], Ines De Pauw[8], Annick Diels[9], Peter Vermeulen[9], Koen Temmerman[9], Trevor Scobey[10], Melissa Mattocks[11], Alexandra Schäfer[10], Dirk Jochmans[6], Steven De Jonghe[6], Pieter Leyssen[6], Winston Chiu[6], Mayra Diosa Toro[12,17], Marleen Zwaagstra[12], Anouk A. Leijs[13], Heidi L. M. De Gruyter[13], Christophe Buyck[14], Klaas Van Den Heede[1,18], Frank Jacobs[5], Christel Van den Eynde[1], Laura Thijs[8], Valerie Raeymaekers[8], Seth Miller[2,19], Amanda Del Rosario[2], Johan Neyts[6,7], Danielle Peeters[9], Ralph S. Baric[10,11], Frank J. M. van Kuppeveld[12], Eric J. Snijder[13], Martijn J. van Hemert[13], Mario Monshouwer[5], Sujata Sharma[2], Ruxandra Draghia-Akli[3,20,22 ✉], Anil Koul[1,22 ✉] & Marnix Van Loock[1,22 ✉]

The membrane (M) protein of betacoronaviruses is well conserved and has a key role in viral assembly[1,2]. Here we describe the identification of JNJ-9676, a small-molecule inhibitor targeting the coronavirus M protein. JNJ-9676 demonstrates in vitro nanomolar antiviral activity against SARS-CoV-2, SARS-CoV and sarbecovirus strains from bat and pangolin zoonotic origin. Using cryogenic electron microscopy (cryo-EM), we determined a binding pocket of JNJ-9676 formed by the transmembrane domains of the M protein dimer. Compound binding stabilized the M protein dimer in an altered conformational state between its long and short forms, preventing the release of infectious virus. In a pre-exposure Syrian golden hamster model, JNJ-9676 (25 mg per kg twice per day) showed excellent efficacy, illustrated by a significant reduction in viral load and infectious virus in the lung by 3.5 and 4 $\log_{10}$-transformed RNA copies and 50% tissue culture infective dose ($TCID_{50}$) per mg lung, respectively. Histopathology scores at this dose were reduced to the baseline. In a post-exposure hamster model, JNJ-9676 was efficacious at 75 mg per kg twice per day even when added at 48 h after infection, when peak viral loads were observed. The M protein is an attractive antiviral target to block coronavirus replication, and JNJ-9676 represents an interesting chemical series towards identifying clinical candidates addressing the current and future coronavirus pandemics.

The *Coronaviridae* is a large family of enveloped, positive-stranded RNA viruses[3]. The *Coronavirinae* subfamily consists of four genera, of which alphacoronaviruses and betacoronaviruses infect mammals, and deltacoronaviruses and gammacoronaviruses mainly infect birds[4]. Seven coronaviruses have been described to infect humans and are thought to originally reside in zoonotic reservoirs such as bats and mice, or in intermediate hosts such as cattle, camels and palm civets[5]. Climate change, increasing pressure on animal environments, closer proximity to wildlife and an increasing global population are linked to zoonotic spillover[6–8].

In the past two decades, betacoronaviruses have caused serious epidemics, including those caused by severe acute respiratory syndrome coronavirus (SARS-CoV) in 2002–2003[9,10], Middle East respiratory syndrome coronavirus (MERS-CoV) first identified in 2012[11] and, most recently, SARS-CoV-2, which paralysed the world and caused the COVID-19 pandemic[12]. Of these coronaviruses, SARS-CoV and

[1]Global Public Health R&D, Janssen Pharmaceutica, Beerse, Belgium. [2]Discovery Technologies & Molecular Pharmacology, Janssen Research & Development, Spring House, PA, USA. [3]Global Public Health R&D, Janssen Research & Development, Spring House, PA, USA. [4]Therapeutics Discovery, Janssen-Cilag, Toledo, Spain. [5]Translational PK/PD & Investigative Toxicology (TPPIT), Janssen Research & Development, Beerse, Belgium. [6]Virology, Antiviral Drug & Vaccine Research Group, Department of Microbiology and Transplantation, Rega Institute for Medical Research, KU Leuven, Leuven, Belgium. [7]VirusBank Platform, Leuven, Belgium. [8]Charles River Laboratories, Beerse, Belgium. [9]Discovery Technologies & Molecular Pharmacology, Janssen Research & Development, Beerse, Belgium. [10]Department of Epidemiology, University of North Carolina at Chapel Hill Gillings School of Global Public Health, Chapel Hill, NC, USA. [11]Department of Microbiology and Immunology, University of North Carolina at Chapel Hill School of Medicine, Chapel Hill, NC, USA. [12]Virology Section, Division of Infectious Diseases and Immunology, Department of Biomolecular Health Sciences, Faculty of Veterinary Medicine, Utrecht University, Utrecht, The Netherlands. [13]Molecular Virology Laboratory, Leiden University Center of Infectious Diseases, Leiden University Medical Center, Leiden, The Netherlands. [14]In Silico Discovery (ISD), Computer-Aided Drug Design (CADD), Janssen Pharmaceutica, Beerse, Belgium. [15]Present address: Gilead Sciences, Foster City, CA, USA. [16]Present address: Red Nucleus, Philadelphia, PA, USA. [17]Present address: Eurofins BioPharma Product Testing, Leiden, The Netherlands. [18]Present address: Independent Researcher, Mechelen, Belgium. [19]Present address: Spark Therapeutics, Philadelphia, PA, USA. [20]Present address: Research & Development, Novavax Inc., Gaithersburg, MD, USA. [21]These authors contributed equally: Pravien Abeywickrema, Yanting Yin, Jiexiong Xie, Sofie Jacobs. [22]These authors jointly supervised this work: Ruxandra Draghia-Akli, Anil Koul, Marnix Van Loock. ✉e-mail: rdraghia@gmail.com; anil.koul@lshtm.ac.uk; mvloock@its.jnj.com

SARS-CoV-2 are closely related and belong to the *Sarbecovirus* subgenus of the betacoronaviruses[4]; both viruses presumably originated from a zoonotic reservoir and made the cross-species jump to humans, while successfully adapting to this novel host species. Since 2020, more than 770 million cases, as well as more than 6.9 million COVID-19-related deaths have been recorded globally[13].

Although the World Health Organization recently declared the end of the emergency phase of the COVID-19 pandemic, the impact of the virus is still ongoing[14], including the continuous emergence of new SARS-CoV-2 variants[14,15]. Given that the sarbecoviruses have been the cause of several outbreaks with tremendous impact on public health, economies and societies around the world and are expected to cause another outbreak within the next 10 years[16], there is a great need for new therapeutics, vaccines and other interventional strategies that could help us to treat patients and prevent another catastrophic pandemic.

Current drugs available to treat COVID-19 include remdesivir (also known by the trade name Veklury)[17,18] and molnupiravir (also known by the trade name Lagevrio)[19], both of which target the RNA-dependent RNA polymerase; and nirmatrelvir and ritonavir (also known by the trade name Paxlovid)[18,20] and ensitrelvir (emergency use)[21], which target the main protease (Mpro). Although these are well-conserved targets with key functions in the viral replication cycle, in vitro and/or in vivo resistance against these drugs has been observed in certain cases[22–24]. Patients can benefit from drugs with other targets either as a monotherapy or as part of combination regimens.

The SARS-CoV-2 30-kb genomic RNA is well described[25–27], and here we therefore focus on the M protein. This protein functions as a master regulator of assembly[1,2,28], being involved in critical interactions directing both the encapsidation of the viral nucleocapsid[2] and the morphogenesis of the coronavirus envelope. The M protein is the most abundant SARS-CoV-2 envelope protein[1]. A cryogenic electron microscopy (cryo-EM) structure using purified recombinant proteins was recently elucidated[2,29]. The M protein contains a short *N*-glycosylated ectodomain followed by three transmembrane domains (three-helix bundles) and a cytosolic intravirion C-terminal domain (a β-sheet sandwich)[2].

M protein forms a homodimer that can adopt two distinct conformational states: an elongated long form (86 Å (height) × 50 Å (width)) and a short form (72 Å (height) × 57 Å (width))[2,30]. It is believed that the M protein dimer is in a conformational equilibrium between these two states[2]. An elongated M protein is associated with a rigid virion, clusters of spikes and a narrow range of membrane curvature. By contrast, the short M protein conformation induces flexibility and lowers spike density[30]. Both forms of the M protein are required for virus assembly and are present in virions[2]. This conformational plasticity and the ability of M protein to forge protein–protein interactions helps to regulate functions such as virion size and membrane composition[1,2,28,30,31].

Here we describe JNJ-9676, a small-molecule inhibitor of SARS-CoV-2 and SARS-CoV with a novel mode of action. JNJ-9676 has double-digit nanomolar in vitro potency against sarbecoviruses including SARS-CoV, all tested variants of SARS-CoV-2, and bat and pangolin SARS-like coronaviruses, as tested in various cell lines. The cryo-EM structure of JNJ-9676-bound SARS-CoV-2 M protein elucidates the molecular basis of inhibition and provides a structural rationale for the resistance mutations, thereby identifying it as an M protein inhibitor. JNJ-9676 is efficacious in vivo in a pre-exposure Syrian golden hamster model, with a lowest effective dose of 25 mg per kg twice daily (BID). JNJ-9676 significantly reduced the viral load and infectious virus in the lung by $3.5 \log_{10}$[RNA copies per mg lung] and $4 \log_{10}$[$TCID_{50}$ per mg lung], respectively. At the same dose, histopathology scores were reduced to the baseline, similar to those of uninfected hamsters. When treatment was administered in a post-exposure hamster model, significant efficacy could be shown even when the compound was added at 48 h after infection, when peak viral loads were observed.

## JNJ-9676 inhibits sarbecoviruses

Following the same methodology as a high-throughput screen using structures that have passed phase-one trials[32], a follow-up screening campaign for small-molecule inhibitors of SARS-CoV-2 was performed in VeroE6-eGFP cells using a diversity set of compounds from the Janssen proprietary library. JNJ-9676 is a representative analogue (Fig. 1a) of a compound series that was identified in the screen. JNJ-9676 exhibits in vitro antiviral activity against SARS-CoV-2 B1 with a 50% effective concentration ($EC_{50}$) ranging from 14 to 22 nM in a variety of cell types and is equipotent against SARS-CoV-2 Omicron B.1.1.529 ($EC_{50}$, 26 nM) and SARS-CoV-2 Delta B.1.617.1 ($EC_{50}$, 14 nM). The molecule is active against SARS-like animal viruses with $EC_{50}$ values ranging from 4 to 6 nM against bat WIV-1, bat SHC014 and pangolin coronavirus (Pg-CoV) (Fig. 1b). Extended-spectrum activity was shown against other betacoronaviruses such as MERS-CoV, HCoV-OC43 and mouse hepatitis virus (MHV), although, with a greater than tenfold lower potency compared with SARS-CoV and SARS-CoV-2 antiviral activity (Extended Data Table 1). Moreover, JNJ-9676 showed single-digit micromolar potency against the prototypic gammacoronavirus infectious bronchitis virus (IBV), but no activity was observed against mildly pathogenic human alphacoronaviruses HCoV-229E and HCoV-NL63 or porcine deltacoronavirus (PDCoV).

Next, JNJ-9676 was tested in three-dimensional (3D) primary human nasal epithelium cultured at the air–liquid interface and infected with SARS-CoV-2 (B1 variant). JNJ-9676 was highly effective in this model and reduced the production of viral RNA with an $EC_{50}$ of $94.0 \pm 3.4$ nM and a 90% effective concentration ($EC_{90}$) of $132.0 \pm 36.7$ nM, similar to nirmatrelvir ($EC_{50}$, $109.1 \pm 55.6$ nM; $EC_{90}$, $268.3 \pm 111.6$ nM) (Fig. 1c).

## JNJ-9676 is an M protein inhibitor

To understand the mechanism of action of this chemical series and determine the stage of the viral replication that is targeted by the compound, we performed time-of-addition (ToA) experiments in HeLa-hACE2 cells. A single viral replication cycle takes 8–10 h, which can be divided into early (1 h post-infection (h.p.i.)), post-entry/replication (3 h.p.i.) and post-replication (5 h.p.i.) stages[33,34]. The biogenesis of infectious progeny was inhibited completely by JNJ-9676, even if compound treatment was delayed until 5 h.p.i. (Extended Data Fig. 1a,b). This suggests that JNJ-9676 may interfere with early events as well as the biogenesis of infectious viral progeny.

To identify the molecular target of JNJ-9676, drug-resistant viruses were selected through serial passaging of the SARS-CoV-2 B1 strain in the presence of gradually increasing concentrations of JNJ-9676 in an in vitro resistance selection (IVRS) assay (Extended Data Fig. 1c). When comparing the mutation profiles of JNJ-9676-selected virus and untreated controls (DMSO), an increased number of mutations was observed in the M protein of treated viruses: L29F, A40P, A85S, A98D, N117K, P132S, E135V, L138I, L138P, S173P and Q185K (Extended Data Fig. 1d–f). The resistance dynamic change curve (Extended Data Fig. 1d) and the generation of resistance mutations shows an increase in the concentration of JNJ-9676 needed for full breakthrough. All mutations reside near the dimer interface of the M protein, indicating a putative binding pocket for the compound, or an effect on the conformational equilibrium between the long and short forms of the M protein dimer (Fig. 1d,e). In particular, the Q185 residues from the two protomers are facing each other at the dimer interface both in the short and long forms, while, in the short form, the P132 residue is in close proximity to E115 in the hinge region of the M protein, a key residue for the structural transition between the two different dimer conformations[2]. Follow-up resistance selection experiments with JNJ-9676 for SARS-CoV-2 variants (Omicron and Delta strain) confirmed the key resistance mutations identified with the SARS-CoV-2 B1 strain (Extended Data Fig. 1d and Extended Data Table 2).

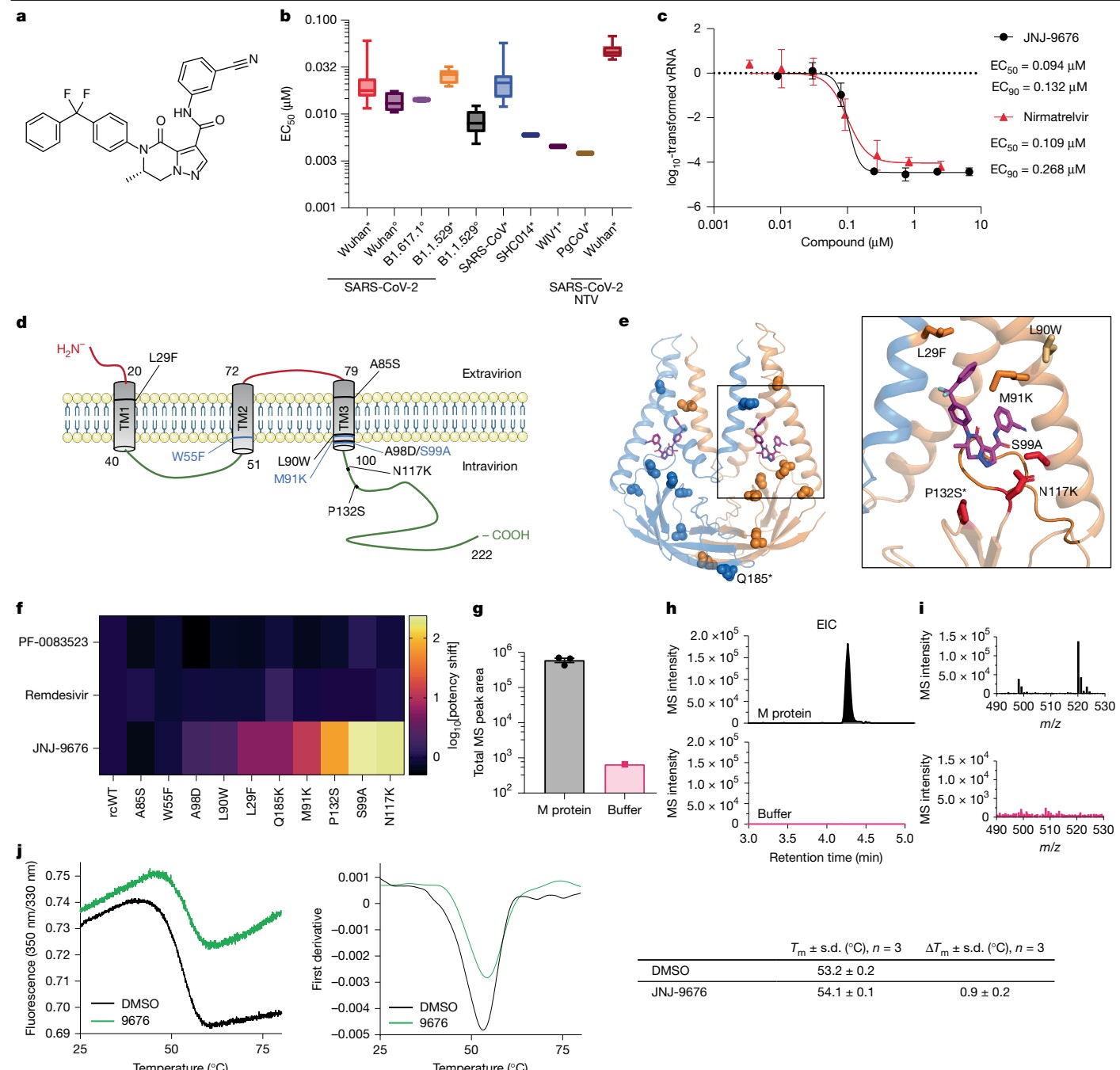

**Fig. 1 | JNJ-9676 targets the M protein. a**, The structure of JNJ-9676, (*S*)-*N*-(3-cyanophenyl)-5-(4-(difluoro(phenyl)methyl)phenyl)-6-methyl-4-oxo-4,5,6,7-tetrahydropyrazolo[1,5-a]pyrazine-3-carboxamide. **b**, The mean EC$_{50}$ values of JNJ-9676 against sarbecoviruses (SARS-CoV-2 B1 strain, *n* = 21 (A549-hACE2 cells), *n* = 6 (VeroE6-eGFP cells); SARS-CoV-2 B1.617.1, *n* = 2 (VeroE6-eGFP cells); SARS-CoV-2 B1.1.529, *n* = 12 (A549-hACE2 cells), *n* = 11 (VeroE6-eGFP cells); SARS-CoV, *n* = 12 (A549-hACE2 cells); SHC014, *n* = 1 (A549-hACE2 cells); WIV1, *n* = 1 (A549-hACE2 cells); and Pg-CoV Guangdong, *n* = 1 (A549-hACE2 cells)) assessed in A549-hACE2 cells (asterisks) or VeroE6-eGFP cells (circles). The EC$_{50}$ for nirmatrelvir in SARS-CoV-2-infected A549-hACE2 cells is also shown. The box plots show the 25th–75th percentile (box limits), median (centre line) and the whiskers show the spread between minimum and maximum values. All replicates listed are biological replicates. **c**, The effect of JNJ-9676 and nirmatrelvir on RNA copy numbers in nasal epithelial cultures infected with SARS-CoV-2 (48 h.p.i., apical). *n* = 3, biological replicates. Data are mean ± s.d. **d**, The transmembrane structure of the SARS-CoV-2 M protein. IVRS mutation residues (black), important residues in the cryo-EM structure (blue), the intravirion domain (green), the extravirion domain (red) and transmembrane domains (grey) are indicated. **e**, The M protein structure annotated with mutations identified in IVRS. Left, mutations identified in more than 2 IVRS samples. Right, mutations in the binding site. The asterisks indicate mutations potentially altering the equilibrium between the long and short forms of the dimer. **f**, The mean EC$_{50}$ fold changes in drug resistance potency of site-directed mutants (SDMs). *n* = 3, biological replicates. **g**, ASMS evaluation of M protein with compound (black) and buffer control with breakthrough (pink). *n* = 3 technical replicates. Data are mean ± s.e.m. **h**, M-protein-enriched extracted ion chromatogram (EIC) with a 3 ppm mass error tolerance window and the corresponding MS spectrum at the M protein EIC peak apex (inset). **i**, Buffer control EIC with a 3 ppm mass error tolerance window with the corresponding MS spectrum at the M protein EIC peak apex (inset). **j**, NanoDSF melting profile with the fluorescence ratio (350 nm/320 nm) and the first derivative plotted against temperature; the table indicates technical replicates (*n* = 3).

Using reverse genetic engineering techniques to study the impact of mutations on viral replication and fitness, based on IVRS or cryo-EM, the individual M protein mutations (P132S, L29F, L90W, N117K, Q185K, W55F, A85S, M91K, A98D and S99A) were introduced into the viral genome using a full-length infectious cloning system. These mutant viruses were able to produce infectious progeny and did not show altered growth kinetics or plaque size compared with the wild-type virus (Extended Data Fig. 1g,h). To assess the effect of these mutations on the susceptibility to JNJ-9676, we implemented a standard antiviral assay with the recombinant viruses. Although several mutations in the M protein led to some phenotypic resistance to the compound, the most pronounced effect was observed with the P132S-, S99A- and N117K-mutant viruses. These mutations caused an increase in the $EC_{50}$ value of JNJ-9676 of 43-fold, 97-fold and 145-fold, respectively (Fig. 1f and Extended Data Table 3).

We confirmed JNJ-9676 target engagement with purified recombinant M protein using biophysical techniques. We obtained reproducible JNJ-9676 recovery with M protein relative to a buffer-only control using offline affinity-selection–mass spectrometry (ASMS) (Fig. 1g–i and Extended Data Fig. 2). Furthermore, we found that incubation of M protein with JNJ-9676 yielded a 0.9 °C stabilization in melting temperature ($\Delta T_m$) using nano differential scanning fluorimetry (nanoDSF) confirming drug–target interaction (Fig. 1j).

Moreover, the high amino acid sequence similarity (>90%) within the M protein sarbecoviruses aligns well with potent antiviral activity while more distantly related betacoronaviruses, such as MHV, HCoV-OC43 and MERS-CoV, clearly exhibited reduced sensitivity to JNJ-9676 (Extended Data Fig. 3a–c).

## JNJ-9676 changes M protein conformation

To understand the binding mode of JNJ-9676, we used cryo-EM to perform a thorough structural analysis of recombinant SARS-CoV-2 M protein in a complex with JNJ-9676. We strategically used FabB and FabE fragments as fiducial markers in cryo-EM analysis; FabB and FabE lock the M protein in the short and long conformation, respectively[2]. This approach led to achieving a global nominal resolution of 3.1 Å for the SARS-CoV-2 M in a complex with JNJ-9676 and FabE and 3.06 Å for M bound to FabB. However, in the SARS-CoV-2 M complex with JNJ-9676 and FabE map (3.06 Å resolution), we could not identify the ligand density corresponding to the JNJ-9676, and the M protein maintained its long-form conformation. By contrast, the map of SARS-CoV-2 M in a complex with JNJ-9676 and FabB (3.06 Å resolution) showed a clearly defined compound density for JNJ-9676.

In the report detailing the long- and short-form conformations, the M protein dimer was found to exhibit a $C_2$ symmetry. However, in the JNJ-9676–M–FabB complex, the M protein dimer adopts a $C_1$ symmetry, with protomer A (chain A) showing superior density compared with protomer B (chain B) (Fig. 2a,b and Extended Data Fig. 4d). Protomer A was therefore selected to elucidate the binding mode of JNJ-9676.

Notably, the M protein displayed significant JNJ-9676-induced conformational changes relative to both reported structures. Compared with the M protein dimer short-form structure (PDB: 7VGS; root-mean-squared deviation (r.m.s.d.) of 1.455 Å for all Cα atoms), notable conformational shifts were prominent at the cytoplasmic termini of transmembrane domain 1 (TM1) (6.6 Å shift), TM2 (5.9 Å shift) and TM3 (7.1 Å shift) (Fig. 2c,d). Compared with the long form (PDB: 7VGR; r.m.s.d. of 4.924 Å for all Cα atoms of M protein) shifts at the cytoplasmic ends of TM1 (2.5 Å shift), TM2 (1.6 Å shift) and TM3 (1.2 Å shift) were also observed (Extended Data Fig. 4). In summary, JNJ-9676 binding induces substantial conformational changes in the M protein dimer, resulting in a novel conformational state.

Our cryo-EM structure elucidates the unique binding mode of JNJ-9676 (Figs. 1a and 2). The compound adopts a rotated L-shaped configuration and binds to an induced pocket formed by TM2 and TM3 of

protomer A, and TM1 of protomer B. The side-chain conformations of Q36 and Y95 in the apo structures clashed sterically with the overlaid JNJ-9676, necessitating their reorganization to accommodate the ligand. The exocyclic amide group of JNJ-9676 makes two significant hydrogen bond interactions with the side chains of N117 and S99, which aligns with the 145-fold and 97-fold increase in $EC_{50}$ observed for the N117K and S99A drug resistance mutations, respectively (Fig. 1f and Extended Data Table 3). Detailed ligand interactions are summarized in Fig. 2e,f.

## JNJ-9676 is efficacious in vivo

The antiviral effect of JNJ-9676 was studied in Syrian golden hamsters infected with the SARS-CoV-2 B1 strain[35]. JNJ-9676 has a pharmacokinetic profile in Syrian golden hamsters that allows sustained exposure using a twice a day dose regimen (Extended Data Fig. 5a). Moreover, JNJ-9676 has a favourable pharmacokinetic profile in dog, rat and cynomolgus macaques (Extended Data Table 4).

First, the antiviral efficacy was assessed in a pre-exposure infection model, wherein the drug was dosed orally starting 1 h before viral infection (Fig. 3a). JNJ-9676 was tested at three different doses: 75 mg per kg, 25 mg per kg and 8.33 mg per kg BID. A dose-dependent decrease in lung viral load, as measured by quantitative PCR with reverse transcription (RT–qPCR) analysis of viral RNA (Fig. 3b and Extended Data Table 4) and end-point titrations of infectious viral progeny (Fig. 3c and Extended Data Table 4) was observed at 75 and 25 mg per kg BID (Extended Data Fig. 5c). At 25 mg per kg BID, JNJ-9676 was able to reduce the viral RNA load and infectious virus in the lung with 3.5 $\log_{10}$[RNA copies per mg lung] and 4 $\log_{10}$[$TCID_{50}$ per mg lung], respectively. The lung tissue was given a cumulative score based on the severity of the different histopathological lesions as visually observed in the lungs of SARS-CoV-2-infected hamsters. JNJ-9676 significantly reduced the cumulative histopathological lung score at 75 mg per kg BID ($P$ = 0.0015) and 25 mg per kg BID ($P$ = 0.0093) (Fig. 3d,e). As a reference, molnupiravir (300 mg per kg BID) was used[36]; although a decline in antiviral parameters was observed, these were not significant.

Next, the antiviral efficacy of JNJ-9676 was assessed in Syrian golden hamsters in a post-exposure model in which the compound was administered after viral infection (Fig. 3f).

After obtaining proof of concept of antiviral efficacy when JNJ-9676 (75 mg per kg BID) was first administered at 10 h.p.i. (Extended Data Fig. 5d–g), the compound was administered at 10, 24 or 48 h.p.i., with 48 h.p.i. representing the peak of viral load in the lungs[37,38]. Even when the start of treatment with JNJ-9676 (75 mg per kg BID) was delayed to 48 h.p.i., viral RNA load and levels of infectious virus in the lung were reduced by 1.4 $\log_{10}$[RNA copies per mg lung] and 1.8 $\log_{10}$[$TCID_{50}$ per mg lung], respectively (Fig. 3g,h, Extended Data Fig. 5c and Extended Data Table 4). Although not significant, similar trends were observed for nirmatrelvir (250 mg per kg BID). Taken together, JNJ-9676 was efficacious against SARS-CoV-2 in both a pre-exposure and a post-exposure therapeutic hamster model with a lowest efficacious concentration of 25 mg per kg BID. These findings position JNJ-9676 as a potential drug candidate for preventing and treating infections caused by sarbecoviruses.

## Discussion

The development of highly potent, safe and effective antiviral therapeutics against pathogenic coronaviruses has a major role in building an arsenal of drugs against the ongoing SARS-CoV-2 pandemic. It is crucial, not only to address patient needs during potential future coronavirus outbreaks, but also as prophylaxis and in post-exposure settings to curb an outbreak or ensure that we have treatment solutions while vaccines are developed. The recurrent emergence of new variants, despite population immunity to SARS-CoV-2, and the potential development of

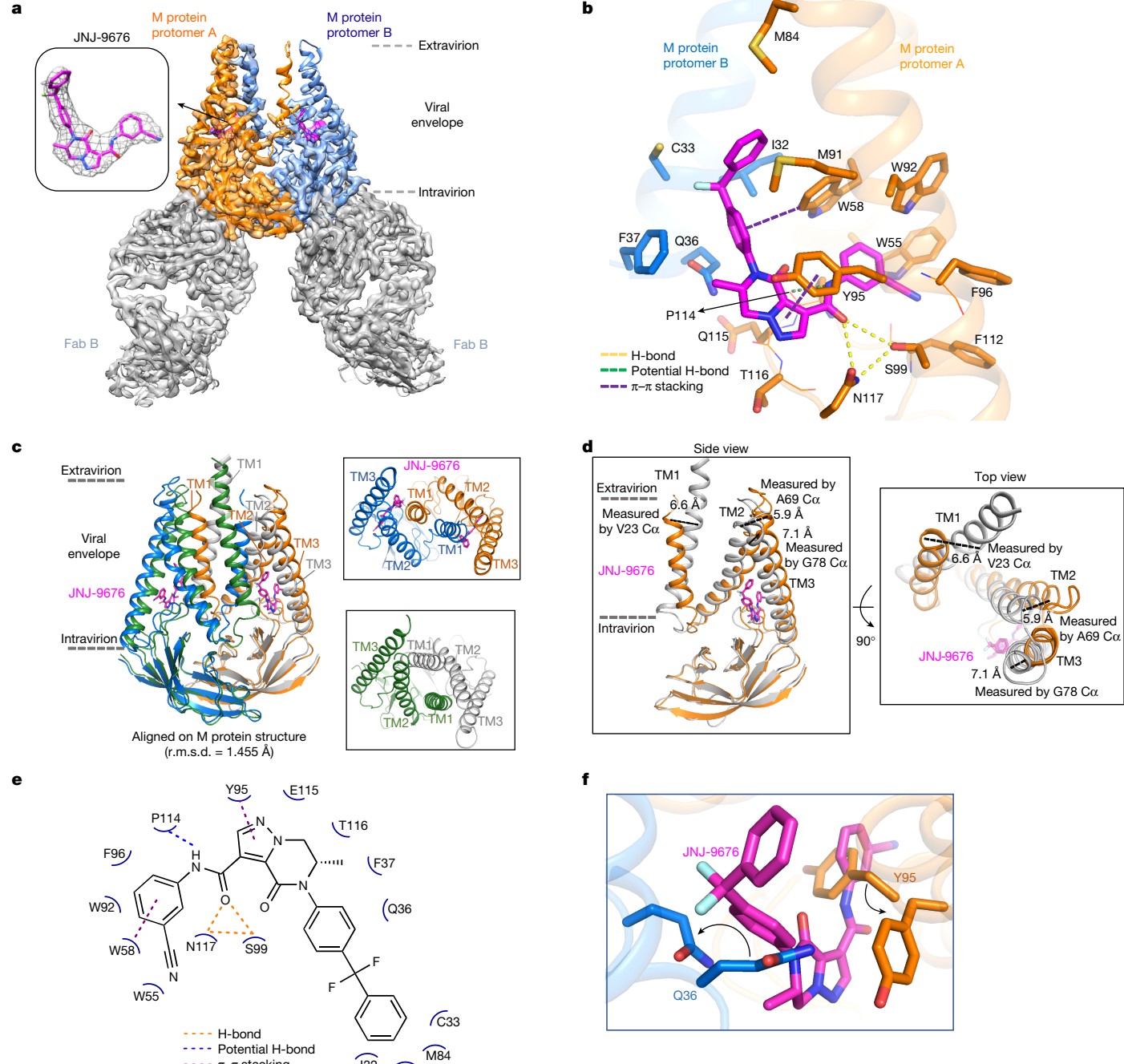

**Fig. 2 | Cryo-EM insights into the JNJ-9676-binding environment. a**, Cryo-EM map of the M protein dimer–FabB complex with JNJ-9676 composed of M protein protomers A (orange), B (blue) and FabB (grey) with JNJ-9676 (inset, magenta sticks) depicted as a density map (grey mesh). The faces of the viral envelope are indicated by dashed lines. **b**, The JNJ-9676-binding pocket on protomer A, with chains shown as a cartoon ribbon, and the interacting residues (within 4 Å), side chain and main chain shown as sticks and a line. Key interactions are indicated by dashed lines (hydrogen (H) bonding, yellow; π–π stacking, purple). **c**, Superimposed structural comparison of the M protein dimer in the absence (short-form, grey/green; PDB: 7VGS) and the presence of JNJ-9676 (orange/blue). Alignment achieved using the whole M protein dimer structure. **d**, Protomer A comparison: JNJ-9676-bound versus JNJ-9676-unbound M protein (short form). **e**, Two-dimensional (2D) interaction patterns of JNJ-9676 and M protein depicted by dashed lines. **f**, The ligand-binding pocket structural comparison using protomer A: JNJ-9676-bound versus JNJ-9676-unbound (apo) M protein (short form). Y95 and Q36 side chains both shift to accommodate JNJ-9676 for an induced fit. Residue shifts are indicated by curved arrows.

vaccine-resistant strains and a group of difficult to vaccinate patients underscore the need for additional antiviral treatments that can reduce severe outcomes and persistent infections[39]. Antiviral drugs can provide multiple benefits, including lowering the viral load, disease severity, time to sustained clinical recovery and number of deaths, and thereby alleviate the burden on patients and healthcare systems and aid in the management of breakthrough infections in vaccinated individuals.

Currently approved drugs to treat COVID-19[17,18,20] have their limitations in terms of use in patients: remdesivir is only available as an intravenous formulation, although trials are ongoing with orally available novel prodrugs of GS-441524[40,41]. Molnupiravir is approved in multiple countries, including the United States (emergency use), but not in the European Union[42], and cannot be administered to pregnant women or children[19]. The required ritonavir boost in the Paxlovid oral antiviral

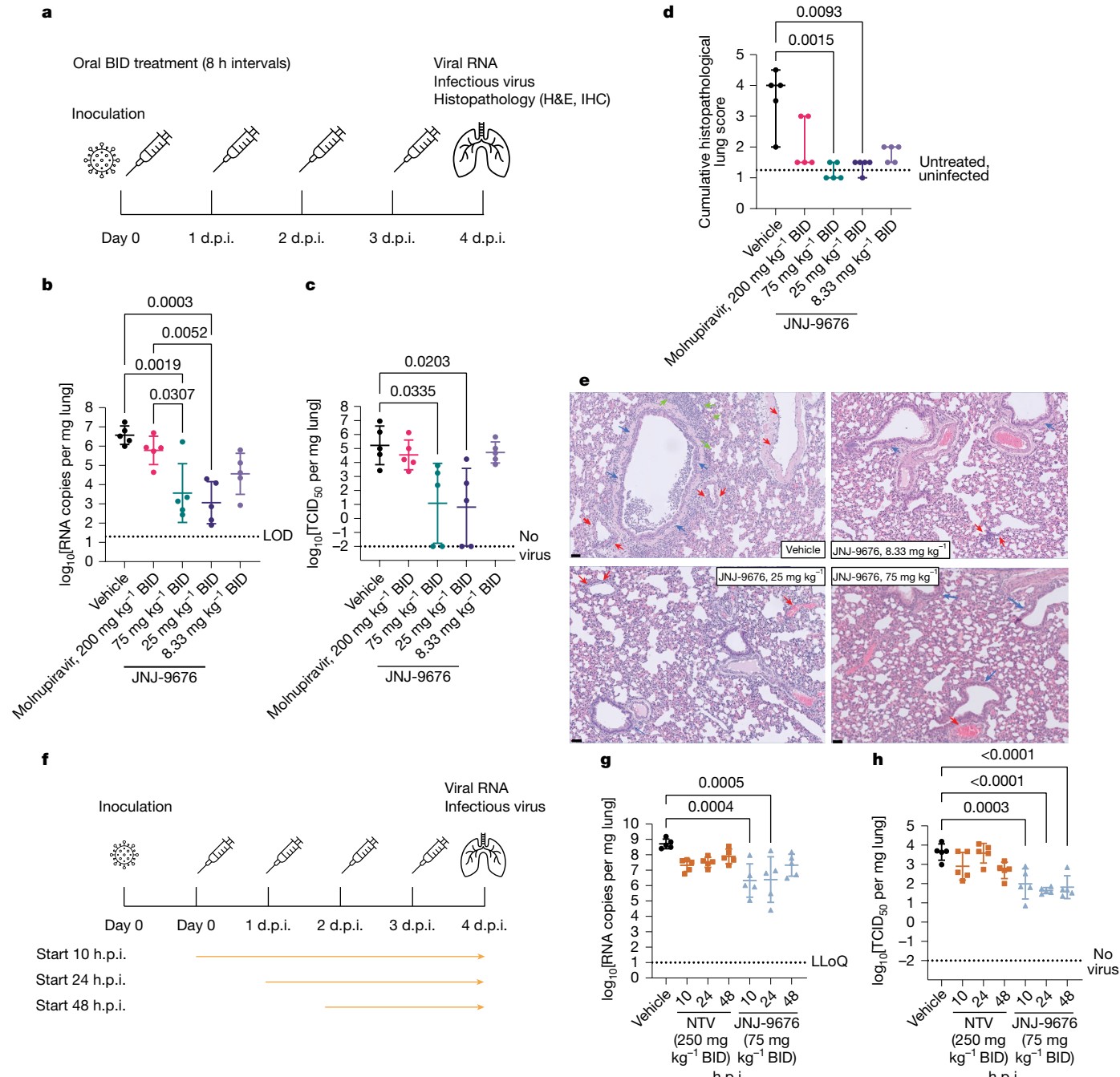

**Fig. 3 | In vivo antiviral activity of JNJ-9676 against SARS-CoV-2. a**, Schematic of a pre-exposure Syrian golden hamster experiment (data are shown in **b**–**e**). **b,c**, Individual datapoints per treatment group. $n = 5$ animals per group. Data are mean ± s.d. The mean differences between groups are calculated using one-way analysis of variance with Šídák's multiple-comparison correction. **b**, The viral load in the indicated lung. The dotted line represents the lower limit of detection (LOD). **c**, Infectious virus in the lung. The dotted line represents the lower limit of quantification (LLoQ). **d**, The cumulative histopathology score. $n = 5$ animals per group. Individual datapoints per treatment group represent the median with the 95% confidence intervals. The dotted line represents a median lung score of 1.25 in healthy, untreated, non-infected animals. The differences between groups were calculated using the nonparametric Kruskal–Wallis tests with Benjamini–Hochberg false-discovery-rate

multiple-comparison correction. **e**, Haematoxylin and eosin (H&E) staining of left lung lobe. Top left, the focal area of bronchopneumonia (green arrows), perivascular (red arrows) and peribronchial inflammation (blue arrows). Bottom left, no bronchopneumonia. Limited perivascular (red arrows) inflammation is indicated. Top right, no bronchopneumonia. Limited but significant perivascular (red arrows) inflammation and normal bronchi (blue arrows) are indicated. Bottom right, no bronchopneumonia. Normal bronchial (blue arrows) and vascular (red arrows) structures are indicated. **f**, Schematic of a therapeutic Syrian golden hamster experiment. **g,h**, Individual data points per treatment group. $n = 5$ per group. Data are mean ± s.d. The mean differences between groups were calculated as in **b** and **c**. The viral load (**g**) and infectious virus (**h**) in the lung is shown. IHC, immunohistochemistry. The doses reflect the amount of compound given for each administration.

(nirmatrelvir and ritonavir) is linked to the risk for ritonavir-mediated drug–drug interactions[43]; moreover, virological rebound[44] is recognized as a major concern with this drug. Another 3CL protease inhibitor,

ensitrelvir, is in clinical development (approved in Japan)[21]. Multiple other antivirals have been brought to the clinic but have failed to meet clinical or safety end points[45–47]. In as many as 45% of patients[48],

there is risk of treatment resistance[49], which could lead to rebound with monotherapy[50]. Adopting a multidrug strategy for ongoing and future coronavirus infections, including drugs with different modes of action, requires new antiviral medications directed at new viral targets[51] to identify potent direct-acting antivirals. To find such molecules, we designed a screening campaign that identified a chemistry with a previously undescribed mode of action targeting the M protein.

Our screens identified a highly potent selective sarbecovirus inhibitor, JNJ-9676, with nanomolar in vitro potency against all tested variants of SARS-CoV-2, as well as against SARS-CoV and several SARS-like zoonotic viruses. Furthermore, computational data show that the full sequence of the M protein is >87% conserved between known sarbecoviruses, and the binding pocket of JNJ-9676 shows >90% homology, thereby supporting a broader zoonotic sarbecovirus coverage. The coverage of the bat reservoirs of coronaviruses is important[52] because bats are natural hosts for coronaviruses[4] and the likelihood of zoonotic spillovers is growing[6,7].

Genotyping of drug-resistant variants that were selected by repeated serial passaging in the presence of JNJ-9676 revealed an accumulation of mutations in the M protein, suggesting that it is the target of the compound. Among the coronavirus structural proteins, the M protein is well-conserved in the sarbecovirus subgenus, is the most abundant structural protein and is involved in viral assembly, membrane budding and morphogenesis of virions[1,53]. Key resistance mutations were observed in the M protein that led to a greater than 100-fold reduction of JNJ-9676's antiviral activity. In global databases of SARS-CoV-2 clinical isolates, these mutations (A40P, A98D, N117K, E135V, L138I, L138P, S173P, Q185K, W55F, S99A, M91K) were absent or occurred at extremely low frequency (L29F (0.126%), A85S (0.223%), P132S (0.008%)) suggesting that they arose specifically in response to compound treatment. The shift in $EC_{50}$ with the introduction of key mutations in the viral genome further points towards the M protein as the likely drug target.

It has yet to be clarified how the emergence of resistance mutation in vitro will translate to in a clinical setting. In case of nirmatrelvir, a subset of mutations occurring in vitro was found in patients[54]. However, it is unclear what the impact on non-immunocompromised patients is, as escape mutants often have compromised viral fitness[24,55]. Nevertheless, the emergence of resistance mutations needs to be monitored in the clinic across human predicted doses and an extended period of time.

The M protein, which is localized in the endoplasmic reticulum–Golgi intermediate compartment, exists in two conformational states: the long and the short form[2]. Here we demonstrated that JNJ-9676 binds to the M protein dimer and forces the protein into an alternative conformational state with a compound-induced binding pocket. A limitation of this study is that no dissociation constant value could be determined using surface plasma resonance due to inefficient capture of M protein on the surface and inability to generate a stable baseline. Owing to the conformational change induced by JNJ-9676, the M protein is impaired in its function, leading to an absence of infectious viral particles in the supernatants of treated cells.

Drug disposition characteristics of JNJ-9676 allowed for a dose regimen that provide sustained exposure and resulted in strong efficacy in SARS-CoV-2 hamster models. In a prophylactic model, 25 mg per kg BID was the lowest effective concentration that significantly decreased the viral load, infectious virus and histopathology scores in the lung by 3.5 $\log_{10}$[RNA copies per mg lung] and 4 $\log_{10}$[$TCID_{50}$ per mg lung], and to the baseline, respectively. Moreover, when JNJ-9676 was administered at peak viral load (48 h.p.i.), viral RNA load and levels of infectious virus in the lungs were still significantly reduced by 1.4 $\log_{10}$[RNA copies per mg lung] and 1.8 $\log_{10}$[$TCID_{50}$ per mg lung], respectively. This exemplifies that JNJ-9676 may be beneficial to treat coronavirus infections in both a prophylactic and therapeutic setting.

In conclusion, we have demonstrated that JNJ-9676 displays antiviral activity against sarbecoviruses by induction of a binding pocket and the introduction of a new conformational state of the M protein. Targeting conserved structural proteins has been described before when influenza matrix and nucleoprotein were found to be druggable[56,57]. Our data paves the way for treatment of sarbecovirus infections by the disruption of M-protein-driven assembly mechanisms and provides a structural basis for the development of next-generation virus assembly inhibitors. The M protein is a target[58] that holds great potential for the development of anti-coronavirus drugs that can be used as a stand-alone treatment or in combination with antivirals targeting other viral functions. The strong potency warrants further development of this compound class and further clinical studies for endemic SARS-CoV-2, as well as an investigation into the impact of antiviral therapies on long-COVID.

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

## Methods

### Ethics and inclusion statement

The research process was a collaboration between Europe-based and US-based researchers; citations reflect the global nature and interest of coronavirus infections. Roles and responsibilities were defined between the authors based on their specific area of expertise to ensure the highest quality standards. Animal studies were approved by local ethics committees (see the designated sections). No human participants were involved. All experiments using pathogens were conducted in the appropriate biosafety-containment-level laboratories.

### Statistics and reproducibility

In Fig. 3e, representative images are shown of H&E-stained left lung lobes of Syrian golden hamsters infected with SARS-CoV-2 and vehicle or JNJ-9676 treated. In this experiment, a full cross-section of the left lung of each of the five animals per group was assessed by a skilled pathologist.

In Extended Data Fig. 2g, uncut western blots are shown of purified M proteins. These blots were generated once as a quality control of the protein obtained.

In Extended Data Fig. 6a, a micrograph from the SARS-CoV-2 M–FabB–JNJ-9676 data collection is shown. To obtain this representative image, 12,988 images were taken.

### Compounds

The synthesis of JNJ-9676 is described in patent WO-2024/008909 and in the Supplementary Methods. Molnupiravir was ordered at MedChemExpress (HY-135853) and nirmatrelvir was synthesized according to literature procedures[59]. For in vitro experiments, JNJ-9676, molnupiravir or nirmatrelvir was dissolved in 100% dimethyl sulfoxide (DMSO) as a 5–100 mM stock. For in vivo experiments, JNJ-9676 was dissolved in 100% polyethylene glycol 400 (PEG400) as stocks of 75, 25 or 8.33 mg ml$^{-1}$, molnupiravir was dissolved in 100% PEG as a stock of 300 mg ml$^{-1}$ and nirmatrelvir as a stock of 250 mg ml$^{-1}$.

### Cells

VeroE6-eGFP cells were cultures as described previously[32]. Human epithelial cell line A549 stably expressing hACE2 (A549-hACE2) were obtained from InvivoGen for SARS-CoV-2 and SARS-CoV experiments, or from the American Type Culture Collection (ATCC, CCL-185) for experiments with zoonotic viruses. The cells were cultured as instructed. Pooled donor nasal epithelial cells grown in air–liquid interface format were obtained from Epithelix as a fully differentiated culture and maintained in MucilAIR medium (Epithelix). All cells were maintained at 37 °C in 5% $CO_2$ unless otherwise noted. All cell cultures were checked for mycoplasma contamination and found negative.

### Viruses

SARS-CoV-2 strains B1 (BetaCov/Belgium/GHB-03021/2020, EPI_ISL_407976), Delta B.1.617.2 (hCoV-19/Belgium/rega-7214/2021, EPI_ISL_2425097) and Omicron B1.1.529 BA.1 (hCoV-19/Belgium/1-SPL21-p1/2021, EPI_ISL_7413964) were obtained from the University of Leuven, Belgium.

SARS-CoV (Frankfurt strain FFM1; GenBank Accession Number: AY291315) was obtained from Goethe University.

Virus stocks were obtained after six passages in VeroE6-eGFP cells, after which stocks were aliquoted, flash-frozen, and stored at −80 °C.

Recombinant viruses, derived from infectious clones of PgCoV GD/2019, RsSHC014, and WIV-1 expressing nanoluciferase, were derived and isolated as working stocks as previously described[60–62].

### Antiviral assays

**SARS-CoV-2 (B1 strain) and SARS-CoV.** JNJ-9676 antiviral activity and compound toxicity against SARS-CoV-2 (B1 strain) and SARS-CoV (FFM1 strain) was determined in a high-content imaging (HCI)-based infection assay in A549-hACE2 cells as described previously[63].

**SARS-CoV-2 (B1 strain, Delta variant and Omicron BA.1 variant).** Antiviral activity of JNJ-9676 against the SARS-CoV-2 B1 strain, Delta variant and Omicron BA.1 variant in VeroE6-eGFP cells was described previously[64].

Cytotoxicity was evaluated on day 5 in treated but uninfected cells using an MTS (3-(4,5-dimethylthiazol-2-yl)-5-(3-carboxymethoxyphenyl)-2-(4-sulfophenyl)-2H-tetrazolium, inner salt) reduction assay[65].

**Zoonotic sarbecoviruses.** The antiviral assays against Pg-CoV, WIV-1 and SHC014 (zoonotic sarbecoviruses) were conducted in A549-hACE2 cells, as previously described[61,62,66,67]. Antiviral assays against other human coronaviruses are detailed in the Supplementary Methods.

**SARS-CoV-2 (B1) in air–liquid interface MucilAIR cultures.** JNJ-9676 or vehicle (0.2% DMSO) was prepared in MucilAIR medium (Epithelix) and added on day 1 to the basal compartment of 24-well MucilAIR plates (Corning Costar clear PS plates, Merck) with pooled nasal epithelial cells. Then, 1 h later, the inserts were infected with SARS-CoV-2 (multiplicity of infection (MOI) of 0.1, B1 strain) for 1 h at 37 °C followed by three PBS washes. At 24 and 48 h, apical washes were collected. After 48 h.p.i., cells were lysed using 200 µl of RLT buffer (Qiagen). Automated RNA extraction was performed using the MagNA Pure instrument (Roche) using the MagNA Pure 96 DNA and Viral NA Small Volume Kit for the apical washes and the MagNA Pure 96 Cellular RNA Large Volume Kit for cell extracts. For the apical washes, an external lysis step (Roche lysis buffer) was included before the RNA extraction. One-step reverse transcription quantitative polymerase chain reaction (RT–qPCR) was performed on extracts using the LightCycler Multiplex RNA Virus Master kit (Roche) and SARS-CoV-2 primers and probe (located in nucleocapsid gene; https://stacks.cdc.gov/view/cdc/84525) and in-house designed β-actin primers and probes. Absolute quantification was performed using a logarithmic dilution series of SARS-CoV-2 nucleocapsid RNA fragment standard (in-house generated), on the LightCycler 480 real-time PCR instrument (Roche).

Toxicity was assessed by exposing non-infected inserts to the same concentration of JNJ-9676 as for antiviral treatment by TEER measurements using an EVOM3 (World Precision Instruments), representative of the cell layer's integrity or health. Brefeldin (0.3 µM; internally synthesized) was used as a toxicity control.

The data were further analysed using GraphPad Prism v.8.

### Viral yield reduction assay using SARS-CoV-2 Omicron BA.1 variant

A549-hACE2 cells (8,000 cells per well, 96-well black polystyrene tissue-culture-treated plates (Sigma-Aldrich)) were seeded onto pre-spotted DMSO-dissolved compound in a nine-point dilution series. Columns containing DMSO were used as controls. On day 2, the cells were infected for 2 h with SARS-CoV-2 virus (MOI, 0.1) after which the cells were washed with PBS, the compound was refreshed, and the plates were incubated for an additional 48 h at 37 °C. On day 4, a cytotoxicity read-out was performed, using the ATPlite reagent and a Viewlux instrument (PerkinElmer). In parallel, the supernatant was collected from the inoculated plates for RNA extraction using either the MagNA Pure instrument (Roche) and the MagNA Pure 96 DNA and Viral Small Volume Kit, or QIAamp Viral RNA Mini kit (Qiagen). One-step RT–qPCR was performed using the LightCycler Multiplex RNA Virus Master kit (Roche), and SARS-CoV-2 primers and probe as described above.

### Antiviral activity of JNJ-9676 against site-directed mutant viruses

IVRS experiments with JNJ-9676 and the generation of site-directed mutants of SARS-CoV-2 are described in the Supplementary Methods and Supplementary Table 1. The impact of these mutations on the

antiviral activity of JNJ-9676 was assessed by an HCI-based antiviral assay[63]. An overview of the MOI used is presented in Extended Data Table 5. The analysis was performed with Phaedra HCI analysis software (v.1.0.10.202309011029). The fold change in $EC_{50}$ for a mutant virus compared with the recombinant WT virus was determined. Calculated potency shifts were transformed to logarithmic scale and visualized as a heat map using GraphPad Prism v.9.5.1.

**Cloning, protein expression and purification of SARS-CoV-2 M.** The gene encoding SARS-CoV-2 M protein (1–222, UniProt: P0DTC5) was synthesized and cloned into a pcDNA3.4 vector, with an added C-terminal linker sequence, an ALFA-tag and a C-tag (SNSLEVLFQGP-SRGGSGAAAGSGSGSGSPSRLEEELRRRLTE-GS-EPEA).

SARS-CoV-2 M was transfected into Expi293F cells (Invitrogen) according to the manufacturer's protocol and incubated at 37 °C with 8% $CO_2$ for 72 h. The cells were collected by centrifugation at 1,000$g$, washed with 1× PBS, flash-frozen and stored at −80 °C.

Cell pellets of SARS-CoV-2 M were thawed and resuspended in lysis buffer (20 mM HEPES pH 7.5, 250 mM NaCl, 5% glycerol (v/v), protease inhibitor (Roche), 50 U ml⁻¹ of nuclease). The cell suspension was homogenized using a glass Dounce homogenizer and then lysed using a M110Y microfluidizer (Microfluidics). The cell lysate was centrifuged at 167,900$g$ for 1 h to collect the membranes. The membranes were resuspended in the same buffer and solubilized by adding lauryl maltose neopentyl glycol (LMNG, Anatrace) and cholesteryl hemisuccinate (CHS, Anatrace) to a final concentration of 1% and 0.1% (w/v), respectively. After incubation for 2 h at 4 °C, the supernatant was collected by centrifugation at 167,900$g$ for 30 min and incubated with C-tag resin (Thermo Fisher Scientific) for 2 h at 4 °C with gentle rotation. The resin was washed with 10 column volumes (CV) of wash buffer (20 mM HEPES pH 7.5, 250 mM NaCl, 1.25% glycerol (v/v), 1 mM EDTA, 0.0025% LMNG (w/v), 0.00025% CHS (w/v)). The protein was eluted using 3 CV of elution buffer (20 mM HEPES pH 7.5, 150 mM NaCl, 1.25% glycerol (v/v), 1 mM EDTA, 0.0025% LMNG (w/v), 0.00025% CHS (w/v), 3 mM C-tag peptide (Vivitide)). The protein was further purified by size-exclusion chromatography on the Superose 6 Increase 10/300 GL column (Cytvia) in buffer (20 mM HEPES pH 7.5, 150 mM NaCl, 0.001% LMNG (w/v), 0.0001% CHS (w/v), 0.00033% glycol diosgenin (GDN; w/v)).

**Cloning, protein expression and purification of FabB.** The sequence encoding the heavy chain of FabB[2] was modified to contain a truncated C terminus to block Fab dimer formation (-CKPCICTVPEVSS) and cloned into the pcDNA3.4 vector with an added C-terminal 6×His-tag containing a linker (GS-GS-HHHHHH). The sequence encoding the light chain of FabB[2] was cloned into a pcDNA3.4 vector with an added N-terminal gLUC signal sequence (MGVKVLFALICIAVAEA).

pcDNA3.4 vectors containing FabB heavy chain and light chain were co-transfected into Expi293F cells (Invitrogen) according to the manufacturer's protocol and incubated for 96 h at 37 °C with 8% $CO_2$.

Conditioned medium was loaded onto a 10 ml HisTrap excel column (Cytvia) at a flow rate of 8 ml min⁻¹. The column was washed with 6 CV of wash buffer (20 mM sodium phosphate pH 6.5, 150 mM NaCl, 20 mM imidazole) and eluted with over 5 CV using a 39.2–500 mM imidazole gradient prepared in buffer (20 mM sodium phosphate pH 6.5, 150 mM NaCl, 500 mM imidazole). Peak fractions of FabB were subsequently purified on to a HiLoad 16/600 Superdex 75 pg column (Cytvia) in buffer (20 mM sodium phosphate pH 6.5, 150 mM NaCl).

**Purification and formation of SARS-CoV-2 M–FabB complex.** SARS-CoV-2 M and FabB were mixed at a 1:2.5 ratio and incubated on ice for 1 h. The SARS-CoV-2 M–FabB complex was loaded into a Superose 6 Increase 10/300 GL column (Cytvia) with buffer (20 mM HEPES pH 7.5, 150 mM NaCl, 0.001% LMNG (w/v), 0.0001% CHS (w/v), 0.00033% GDN (w/v)). The peak fractions containing the SARS-CoV-2 M–FabB complex

were pooled, 100 µM JNJ-9676 was added and incubated for 1 h on ice. The sample was diluted to 0.2–0.8 mg ml⁻¹ with size-exclusion chromatography (SEC) buffer containing 100 µM JNJ-9676 for cryo-EM.

**Nano differential scanning fluorometry.** Experiments were performed in a total volume of 10 µl. A Prometheus NT.Plex instrument (NanoTemper Technologies) was used to measure the melting temperatures. The samples were prepared in a 384-well plate with 0.5 mg ml⁻¹ purified recombinant SARS-CoV-2 M and 100 µM of JNJ-9676 in 20 mM HEPES pH 7.5, 150 mM NaCl, 0.001% LMNG (w/v), 0.0001% CHS (w/v), 0.00033% GDN (w/v) and 1% DMSO (v/v). The samples loaded into standard-grade glass capillaries were measured under a temperature range of 25–95 °C with a temperature gradient of 1 °C min⁻¹, and the intrinsic protein fluorescence at 330 and 350 nm was recorded. The data were analysed using PR.ThermControl v.2.1.6 (NanoTemper Technologies) (technical replicates ≥ 3).

**Offline ASMS**
The offline ASMS experiment consisted of the preparation of three sample types: compound QC, protein target (M protein) and no-protein control (breakthrough).

For the preparation of SEC filter plates for offline ASMS, 130 µl of pre swollen Bio-Gel P10 resin slurry was added to each well of a low-protein-binding Millipore HTS 384 HV filter plate (hereafter, size-exclusion plate) with a 0.45 µm Durapore (PVDF) membrane (MZHCN0W10). The size-exclusion plate was placed into a 4 °C refrigerated centrifuge, centrifuged at 1,000$g$ for 2 min and the flow-through was discarded. Each cartridge was then washed a total of four times using 50 µl buffer containing 20 mM HEPES, pH 7.5, 150 mM NaCl, 0.001% LMNG, 0.0001% CHS, 0.00033% GDN and 2% DMSO, whereby the flowthrough from each wash was discarded after centrifugation at 1,000$g$ for 2 min. The ASMS assay plate was prepared using an Echo acoustic liquid handler, and an aliquot of 20 nl of 5 mM compound dissolved in 100% DMSO was transferred from the source plate into four separate wells of a 384-well, natural, polypropylene V-bottom plate (781280). An aliquot of purified recombinant M protein stock solution was thawed on ice, then diluted using assay buffer to a working concentration of 5 µM and 2% DMSO. Then, 20 µl of the resulting working protein stock was dispensed into three wells containing compound to yield a final concentration of 5 µM (3 technical replicates). To control for compound breakthrough of the SEC resin, either in-solution or through micelle partitioning, a separate working stock was prepared without protein and dispensed as a 20 µl aliquot into the remaining compound well. The plate was centrifuged at 1,000$g$ for 1 min at room temperature and incubated at 25 °C for 30 min.

All of the samples were transferred to the size-exclusion plate, which was quickly centrifuged at 1,000$g$ for 2 min at 4 °C to minimize compound breakthrough. The resulting flowthrough was diluted with 15 µl MS-grade water (Honeywell) to reduce the detergent concentration and centrifuged further at 2,000$g$ for 5 min at room temperature to collect any insoluble precipitate.

The compound QC sample was prepared separately without additional handling, whereby a 5 nl aliquot of 5 mM compound in DMSO was transferred from the source plate into a 384-wellplate and combined with 25 µl of 49% acetonitrile, 2% DMSO solution.

All liquid chromatography–mass spectrometry (LC–MS) analyses were performed on an Agilent 1290 Infinity II uHPLC system coupled to an Agilent 6545XT qTOF using the Agilent MassHunter (v.10.0) software. A 4 µl sample injection was loaded with water as a loading solvent onto the reversed-phase column (2.1 × 35 mm ACQUITY UPLC BEH C18 column, 130 Å, 1.7 µm), heated to 40 °C. LC separation was performed using mobile phases consisting of water (solvent A) and acetonitrile (solvent B), each containing 0.2% formic acid. The LC method used a constant flow rate of 0.1 ml min⁻¹ and consisted of a 1 min wash with 5%

solvent B, a steep gradient from 5% to 20% B over 0.1 min, a subsequent shallow gradient from 5% to 95% B over 1.9 min, followed by a hold for 1 min and a return to 5% B in 0.1 min with a 0.9 min hold. The MS instrument was operated in positive polarity mode with centroided data acquisition, where the source was set to a 350 °C drying gas temperature and 13 l min⁻¹ drying gas flow rate; 375 °C sheath gas temperature and 12 l min⁻¹ sheath gas flow rate; capillary voltage of 3,300 V; nozzle voltage of 500 V; nebulizer pressure of 50 psi; fragmentor of 125 V; and skimmer of 50 V. A reference mass solution consisting of purine and HP-0921 (Agilent, G1969-85001) was prepared according to the manufacturer's instructions and infused to apply automatic mass correction to all spectra acquired from 110 to 1,100 $m/z$ at a rate of 1 spectrum per second.

MS data processing was performed using Agilent MassHunter Qualitative Analysis (v.10.0), where the $[M+H]^+$, $[M+Na]^+$, and $[M+K]^+$ masses were extracted and merged using a mass error tolerance window of 3 ppm.

**Cryo-EM.** QuantiFoil Au 1.2/1.3 300 mesh grids were subjected to glow discharge using the PELCO easiGlow Discharge Cleaning System. A total of 3 µl recombinant M protein sample (0.8 mg ml⁻¹), prepared as described above, was applied to the EM grids, which were vitrified with a Vitrobot (Thermo Fisher Mark IV) using the following settings: blot time 4 s, blot force 0, wait time 0 s, inner chamber temperature 4 °C, and 100% relative humidity. Flash-freezing in liquid ethane cooled by liquid nitrogen was performed. Cryo-EM data collection was automated on the 200 kV Thermo Scientific Glacios microscope controlled by EPU software. Micrographs were taken at ×105,000 magnification using a Facon4 detector (Gatan) in counting mode. Each 6 s exposure recorded 40 frames with a total dose of 40 e⁻ Å⁻². The calibrated physical pixel size for all digital micrographs was 0.910 Å. All details corresponding to individual datasets are summarized in Extended Data Table 6.

Cryo-EM data collection and image quality were monitored using cryoSPARC Live v.3.2. Image preprocessing steps, including patch motion correction, patch contrast transfer function (CTF) estimation, blob particle picking (100–200 Å diameter) and extraction, were performed simultaneously. A total of 12,988 raw micrographs was recorded during a 4-day data collection session using the Glacios microscope. Acceptable 2D classes served as templates for particle repicking. One round of live 2D image classification yielded approximately 1.2 million good particle images. These particles were used for 3D reconstruction. The first round of five starting 3D models were calculated, resulting in one major 3D class, followed by a second round of four 3D classes. One major class underwent non-uniform 3D refinement and local refinement using 484,610 particles and was further refined to a 3D EM map with an average resolution of 3.06 Å.

Resolutions were estimated by applying a soft mask around the protein complex density using the gold-standard (two halves of data refined independently) FSC = 0.143 criterion. Before visualization, all density maps were sharpened by applying different negative temperature factors along with the half maps and used for model building. Local resolution was determined using ResMap. Detailed statistics about the cryo-EM data processing can be found in Extended Data Fig. 6a–f.

**Cryo-EM model building, refinement and validation.** Human SARS-CoV-2 M protein dimer (short form) in a complex with FabB (PDB: 7VGS) was used as the initial model for atomic model building of the EM map. For the M–FabB complex model building, the M protein was manually built using COOT[68]. The FabB was fitted into the 3D map using Chimera and then further refined manually with COOT followed by real-space refinement in Phenix[69]. Detailed data collection and structural refinement statistics are provided in Extended Data Table 6 and Extended Data Fig. 6g. Structure representations were generated using Pymol (v.2.0)[70] and Chimera[71].

**Pre-exposure Syrian golden hamster model.** Housing conditions and experimental procedures were performed according to project 062/2020, approved by the ethics committee of KU Leuven, Belgium license number LA1210186. The hamster infection model of SARS-CoV-2 has been described previously[72]. Statistical power analysis as well as the limitations of the study size warranted 5 animals per group to obtain statistical significance in Syrian golden hamster studies. After arrival, the animals were randomly assigned to groups. No blinding was performed during the experiment. Female hamsters (Janvier Laboratories), 8–10 weeks old, were inoculated intranasally with 50 µl containing $2 \times 10^6$ TCID₅₀ SARS-CoV-2 B1 (day 0). Animals were treated according to the schedule (Fig. 3a), with vehicle or JNJ-9676 (75, 25 or 8.33 mg per kg per dose, formulated in 100% PEG400). Animals were dosed BID at 08:00 and 16:00. Viral RNA and infectious virus levels in the right lung were quantified using RT–qPCR and end-point virus titration, whereas left-lung samples were subjected to histopathological scoring, as described previously[72] (Fig. 3b–e).

For histological examination, the fixed lung tissue sections (5 µm) were analysed after staining with haematoxylin and eosin and scored blindly for lung damage by an expert pathologist. The scored parameters, (cumulative score, 1 to 3), were as follows: congestion, intra-alveolar haemorrhagic, apoptotic bodies in the bronchus wall, necrotizing bronchiolitis, perivascular oedema, bronchopneumonia, perivascular inflammation, peribronchial inflammation and vasculitis.

All statistical analyses were performed in GraphPad Prism v.9.5.0 and validated using R (v.3.6.1). A log₁₀ transformation was applied to the lung viral-load data (RNA and infectious virus) to approximate normality. The mean differences between the treatment groups and the vehicle group were estimated using the one-way analysis of variance with Šídák's multiplicity correction to account for multiple testing.

In the case that normality could not be assumed for the outcome variable or in case of lung histopathology, the nonparametric Kruskal–Wallis test by ranks was applied. The post hoc Dunn's test with the Benjamini–Hochberg's multiplicity correction was applied to account for multiple testing. A significance level of 0.05 was used.

**Post-exposure Syrian golden hamster model.** Housing conditions and experimental procedures were performed as described and approved by the ethics committee of Johnson & Johnson Research & Development (Belgium), license number LA1100119. Statistical power analysis as well as the limitations of the study size warranted 5 animals per group to obtain statistical significance in Syrian golden hamster studies. After arrival, the animals were randomly assigned to groups. No blinding was performed during the experiment. Female Syrian golden hamsters (Janvier Laboratories) aged 8–10 weeks were anaesthetized by isoflurane inhalation and inoculated intranasally with 100 µl of PBS containing $1 \times 10^4$ TCID₅₀ SARS-CoV-2 (day 0). The animals were treated orally starting at 10, 24 or 48 h.p.i. and continued to be dosed BID at 10 h intervals with vehicle or JNJ-9676 (75 mg per kg per dose in PEG400) (Fig. 3f). The animals were dosed BID at 08:00 and 16:00. On day 4 after infection, the hamsters were euthanized by CO₂ inhalation. Whole right lungs were homogenized by bead disruption using the Precellys homogenizer (Bertin Instruments). Viral RNA and infectious virus levels were quantified in the lung homogenate supernatant by RT–qPCR and end-point virus titration, respectively (Fig. 3g,h). RNA was extracted using the MagNA Pure 96 DNA and Viral NA Small Volume Kit following the Viral NA universal SV 4.0 protocol (Roche). RT–qPCR was performed using the LightCycler Multiplex RNA Virus Master kit (Roche), and SARS-CoV-2 primers and probe as described above. For end-point titrations, a 1:10 serial dilution of the lung homogenate was prepared in 1× MEM (without phenol red (Thermo Fisher Scientific) supplemented with 2% FCS (Biowest), 2 mM alanyl-glutamine (Sigma-Aldrich) and 0.04% gentamicin (Thermo Fisher Scientific). This dilution series was then added to confluent Vero E6 cells in a 96-well plate and incubated for 72 h at 37 °C. The infectious viral titres of the samples were determined

by microscopically scoring the virus-induced cytopathic effects and quantified as the $TCID_{50}$ $ml^{-1}$ according to the Reed–Muench calculation method[73]. The $TCID_{50}$ $ml^{-1}$ values were normalized to the total weight of the right lung and expressed as $TCID_{50}$ per mg tissue.

The statistical analysis was performed as described above.

## Reporting summary

Further information on research design is available in the Nature Portfolio Reporting Summary linked to this article.

## Data availability

All data supporting the findings of this study are available within the Article. All accession codes are provided in the Article. Cryo-EM maps have been deposited at the Electron Microscopy Data Bank (EMD-43745), and the atomic coordinates of the M−FabB complex structures have been deposited at the PDB (8W2E). The synthesis and chemical characterization of all compounds described here in are provided in the Supplementary Methods. No cropped images of western blots are shown, the uncropped images of the western blots are presented in Extended Data Fig. 2g. Source data are provided with this paper.

## Code availability

In this Article, no custom code or mathematical algorithms were used. All HCI analysis was performed in Phaedra HCI analysis software (v.1.0.10.202309011029). All antiviral data ($EC_{50}/_{90}$, $CC_{50}$) were processed using GraphPad Prism (v.8 or v.9). Antiviral data in air−liquid interface cultures were processed in LightCycler software (Roche) and GraphPad Prism (v.8). Toxicity $CC_{50}$ values were calculated in GraphPad Prism (v.8). NanoDSF data were analysed using PR.ThermControl v.2.1.6 (NanoTemper Technologies). ASMS data processing was performed using Agilent MassHunter Qualitative Analysis (v.10.0). Cryo-EM structure representations were generated using PyMOL (v.2.0) and Chimera (v1.17.3). Cryo-EM data collection and image quality were monitored using cryoSPARC Live v.3.2 Image. Local resolution was determined using ResMap. For the M−FabB complex model building, the M protein was manually built using COOT. The FabB was fitted into the 3D map using Chimera and then further refined manually with COOT followed by real-space refinement in Phenix. The data were processed using the Bruker TOPSPIN program v.4.1, and $^1H$ and $^{13}C$ chemical shifts were analysed using ACD/Spectrus software 2023 v.1.1. All statistical analyses for in vivo experiments were performed in GraphPad Prism (v.9) and validated using R (v.3.6.1). The amino acid sequences for the M protein were downloaded from https://www.ncbi.nlm.nih.gov/ (on 31 January 2023) and aligned through a pairwise sequence alignment using the Needleman−Wunsch algorithm through the EMBOSS-Needle tool from EMBL-EBI (https://www.ebi.ac.uk/jdispatcher/psa/emboss_needle). All visualizations of the sequence alignments were made using Tableau Software (online version). Graphs and figures were generated using Microsoft PowerPoint (v.2308 Build 16731.20460), GraphPad Prism (v.8 and 9), PyMOL Molecular Graphics System (v.2.0), Chimera (v.1.17.3), CryoSparc (v.4.4.1), 3D-FSC (v.1.0), Grace (v.5.1.25) and Image Lab (v.6.0.1); the software is made available by Janssen Pharmaceutica.

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

**Acknowledgements** We thank K. Vercauteren and A.-T. Henze (Akkodis Belgium) for medical writing and coordination support on behalf of Janssen Pharmaceutica; L. Vijgen, M. Burcin, L. Zhang, E. Parmee and L. Lombardo for the comments and discussions; B. C. Arquero for the analysis of the NMR spectra of JNJ-9676; and M. Van Heerden for the histopathology scoring. This work was supported by Janssen Research & Development. This project has also been funded in part with federal funds from the Office of the Administration for Strategic Preparedness and Response, Biomedical Advanced Research and Development Authority (BARDA), under OTA numbers HHSO100201700018C and HHSO100201800012C. The panCoV nLUC drug assays were developed under NIH AID AI171292 (RSB) and applied to this project.

**Author contributions** E.V.D. contributed to the conceptualization and study design, critical revision of results, manuscript writing, editing and supervision. P.A. contributed to conceptualization of target engagement and structure determination studies, designed constructs, guided protein production method development and coordinated activities for expression, protein production, target engagement and cryo-EM. Y.Y. contributed to the conceptualization of cryo-EM experiments, structure modelling and analysis, figure preparation, manuscript writing and editing. J.X. contributed to the conceptualization, design, execution, data analysis and visualization of vRNA reduction experiments, IVRS assay, reverse-engineer platform to generate mutant SARS-CoV-2 strains, $EC_{50}$ shift experiments, and manuscript writing and editing. S.J. contributed to the in vivo part of this manuscript by aiding with study design, execution of experiments, data analysis and visualization, and manuscript writing and editing. M.K.M. contributed to the purification of recombinant proteins, prepared samples for cryo-EM studies and performed NanoDSF experiments, data collection and analysis, figure preparation, and manuscript writing and editing. J.D. contributed to the dose−response testing in air–liquid interface nasal epithelial cultures, ToA studies and HCI-based antiviral assays by aiding with study design, execution of experiments, data analysis and visualization, and manuscript writing and editing. R.M. performed expression of recombinant proteins. M.P. contributed to the purification of recombinant proteins. S. Marsili contributed to structure analysis, figure preparation, and manuscript writing and editing. M.S. and F.J. conceptualized and analysed experiments describing drug disposition properties, and contributed to data analysis, and manuscript writing and editing. L.G. contributed to the conceptualization of ASMS experiments, ASMS data collection, ASMS analysis, manuscript writing and editing. R.A. coordinated the execution of the hamster infection studies carried out at KU Leuven. M.V.G. contributed to the conceptualization and study design, critical revision of results, manuscript writing and editing and supervision. N.V.d.B. contributed to the conceptualization, execution, data analysis and visualization of vRNA reduction experiments and $EC_{50}$ shift experiments, and contributed to the design and execution of the cloning strategy to generate mutant SARS-CoV-2 strains and manuscript writing and editing. I.D.P. contributed to the dose–response testing in air–liquid interface nasal epithelial cultures, ToA studies, viral-yield-reduction experiments and potency-shift experiments by aiding study design, execution of experiments and data analysis. A.D., P.V., K.T. and D.P. generated the SARS-CoV and SARS-CoV-2 data in A549-hACE2 cells. T.S. contributed to the in vitro part of this manuscript by aiding with drug inhibition assay design, execution of experiments, data analysis and revision of results, manuscript writing and editing. M. Mattocks contributed to the in vitro part of this manuscript by aiding with drug inhibition assay design, data analysis and revision of results, manuscript writing and editing. A.S. contributed to the in vitro part of this manuscript by aiding with drug inhibition assay design, data analysis and revision of results, manuscript writing and editing. D.J., S.D.J., P.L. and W.C. developed and coordinated the high-throughput antiviral screenings assay and coordinated the throughput antiviral screening assay and hit validation at KU Leuven. M.D.T. and M.Z. performed antiviral assays and analysed results. A.A.L. and H.L.M.d.G. performed MERS antiviral experiments and analysed results. C.B. contributed to the identification of this chemical series of inhibitors from screening to hit data analysis of the cryo-EM structure, the revision of results and manuscript editing. K.v.d.H. contributed to structure analysis, figures preparation, manuscript writing and editing. C.V.d.E., V.R. and L.T. contributed to the dose−response testing in air–liquid interface nasal epithelial cultures, ToA studies and HCI-based antiviral assays by aiding with study design, execution of experiments, data analysis and visualization, manuscript writing and editing. S. Miller and A.D.R. were involved in revision of results, manuscript writing, and editing. J.N. coordinated the in vitro and in vivo studies carried out at KU Leuven. R.S.B. contributed to the design and evaluation of experiments to evaluate compound breadth using reporter viruses across multiple coronavirus strains. F.J.M.v.K. supervised broad-spectrum antiviral assays and contributed by performing data analysis, critical revision of results and manuscript editing. E.J.S. and M.J.v.H. supervised, performed data analysis, and contributed to the writing and editing of the manuscript.

M. Monshouwer, S.S., R.D.-A., A.K. and M.V.L. contributed to the conceptualization and study design, critical revision of results, manuscript writing, editing and supervision.

**Competing interests** M.V.L., E.V.D., M.V.G. and C.B. are named as inventors on a pending patent application claiming inhibitors of coronavirus (WO 2024/008909), which was filed by the Applicant Janssen Pharmaceutica. M.V.L., E.V.D., J.X., S.J., L.G., J.D., M.V.G., R.D.-A., A.D., S. Marsili, S. Miller, C.V.d.E., A.D.R., P.V., K.T., D.P. and C.B. were/are employees of Janssen Pharmaceutica and may possess stocks of Johnson & Johnson. S. Miller is an employee of Spark Therapeutics and may possess stocks of Roche. N.V.d.B., L.T., V.R. and I.D.P. are employees of Charles River Laboratories, a contract research organization and may possess stocks of Johnson & Johnson. M.S. is an employee of Gilead Sciences and may possess stocks of Gilead Sciences. A.A.L., H.L.M.d.G., E.J.S. and M.J.v.H. received funding from Janssen Pharmaceutica to perform contract research. M.D.T., M.Z. and F.J.M.v.K. received funding from Janssen Pharmaceutica to perform contract research. The other authors declare no competing interests.

**Additional information**
**Correspondence and requests for materials** should be addressed to Ruxandra Draghia-Akli, Anil Koul or Marnix Van Loock.

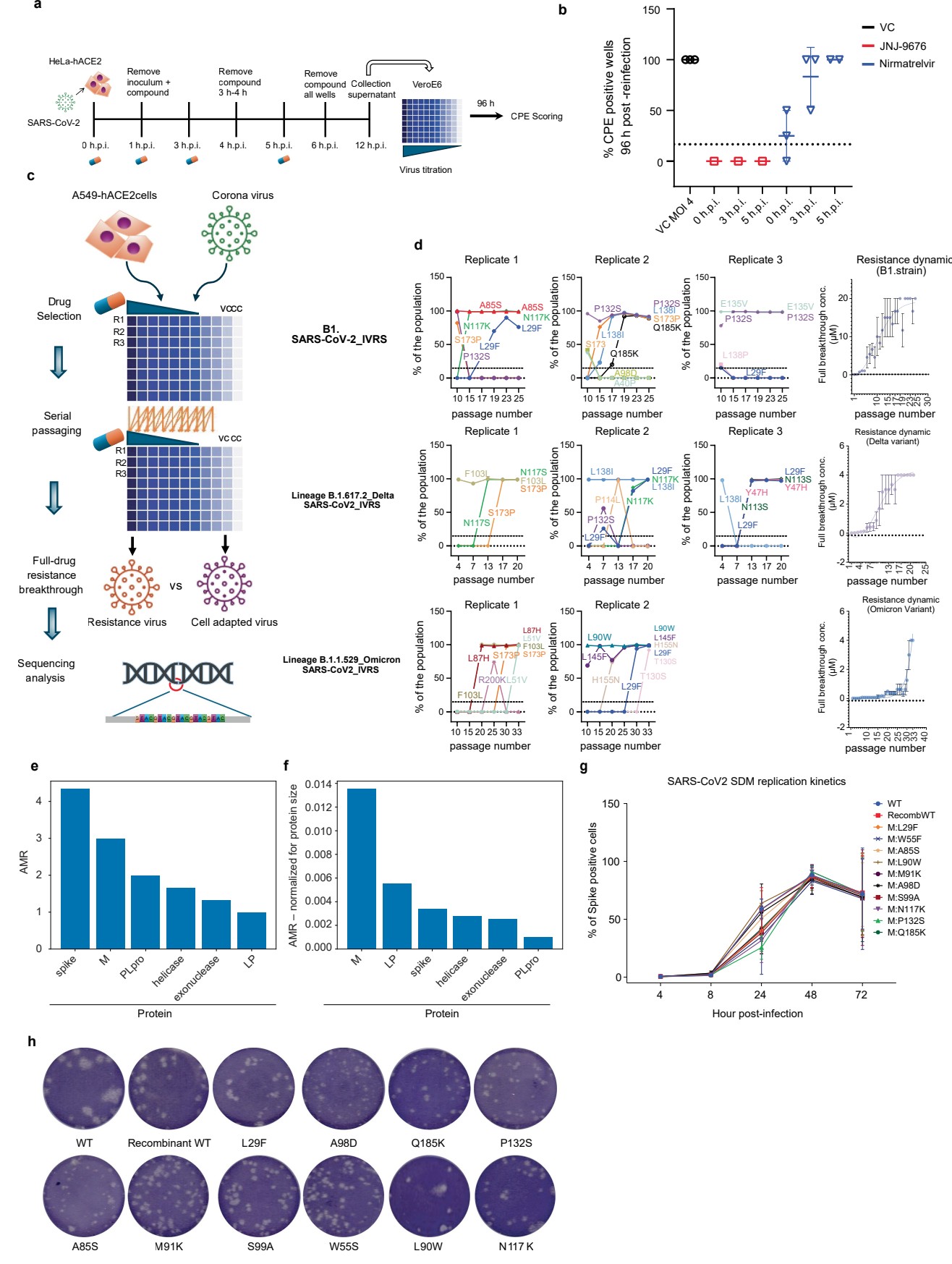

**Extended Data Fig. 1** | See next page for caption.

**Extended Data Fig. 1 | Time-of-addition (TOA) and in vitro resistance selection experiments with JNJ-9676. a**, Representation of ToA study with CPE scoring as a readout showing the presence of infectious progeny after compound treatment. **b**, ToA assay with 5 µM JNJ-9676 (n = 1) and nirmatrelvir (n = 2 for 5 h.p.i. and n = 3 for 0 and 3 h.p.i., error bars representing the standard deviation) reading out infectious virus. The dotted horizontal line represents the reinfection with supernatant directly collected after washing away the virus input at 1 h.p.i. (n = 3). **c**, Schematic representation of the IVRS procedure. **d**, Whole genome sequencing was utilized to investigate the emergence of mutations in various SARS-CoV-2 lineages/variants under selection of JNJ-9676 (n = 3 for B1 and B.1.617.2 lineages and n = 2 for the omicron lineage; error bars showing error of the mean). Each coloured line represents the appearance dynamics of a specific mutation during virus passaging in the presence of JNJ-9676, with the mutation colour-coded accordingly. Sequencing was conducted on SARS-CoV-2 variants collected at intervals of every 2nd to 6th passage and at the end of the experiment. The experiment involved two passages per week. The dotted line on the graph represents the 15% threshold for variant detection compared to the WT in the virus population. The resistance dynamics for each viral strain was plotted. **e**, The average number of IVRS mutations per replicate (AMR) was defined. Proteins with an AMR ≥ 1 were considered potential targets. **f**, AMR values normalized for protein size. **g**, Effect of resistance mutations on replication fitness (n = 3 with 8 technical replicates per experiment, error bars showing the standard deviation). **h**, Plaque assay showing representative images (n = 3, independent experiments) of plaque sizes from the site directed mutants in **g**. ToA, time-of-addition; CPE, cytopathic effect; h.p.i., hours post-infection; IVRS, in vitro resistance selection; WT, wild-type; AMR, average number of IVRS mutations per replicate.

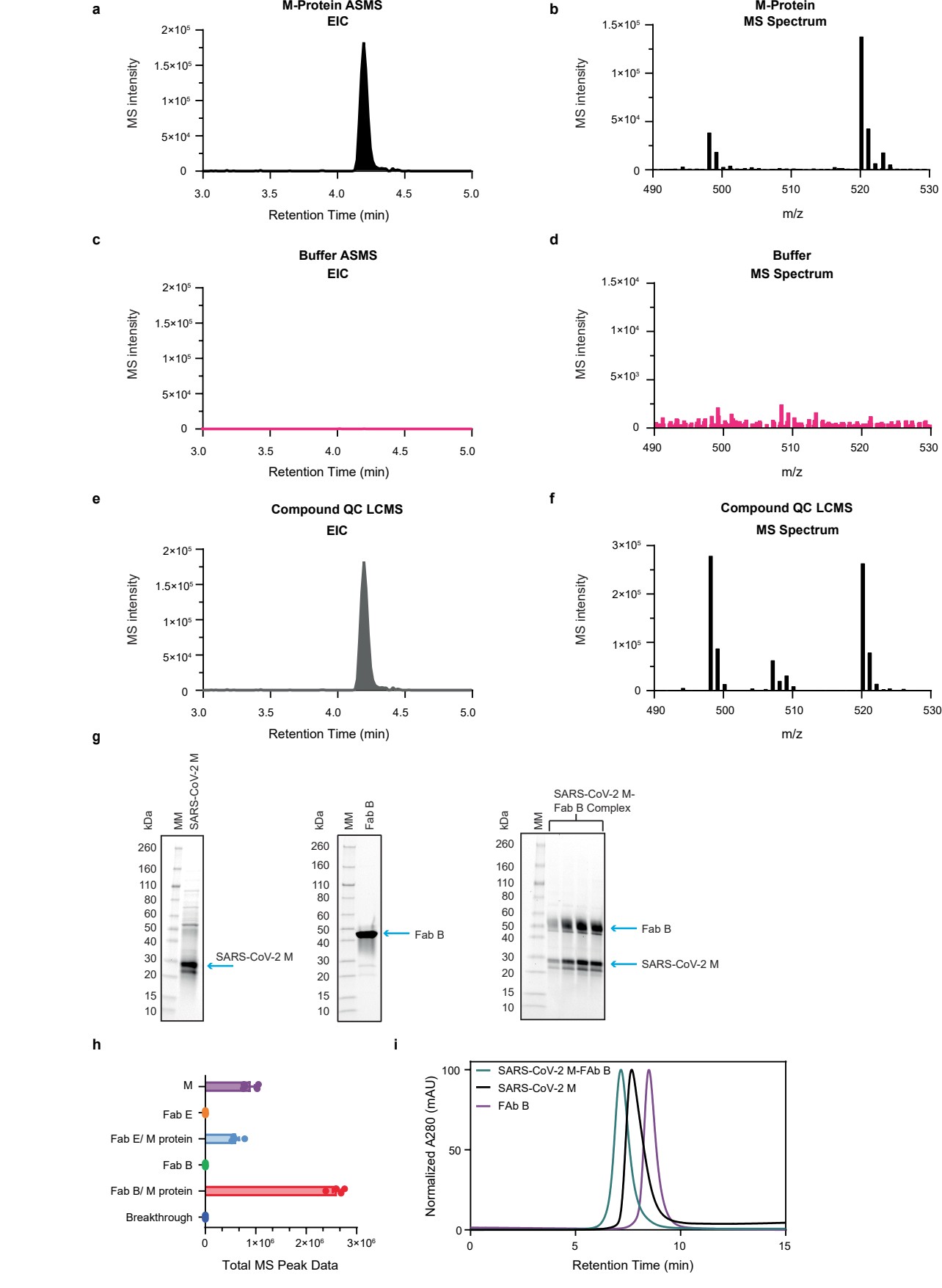

**Extended Data Fig. 2** | See next page for caption.

**Extended Data Fig. 2 | Characterization of JNJ-9676 binding to the M protein. a**, Representative M protein-enriched extracted ion chromatogram (EIC) with 3 ppm mass error tolerance window. **b**, Corresponding MS spectrum at M protein EIC peak apex. **c**, Buffer control EIC with 3 ppm mass error tolerance window. **d**, Corresponding MS spectrum at M protein EIC peak apex. **e**, Compound QC LCMS EIC with 3 ppm mass error tolerance window. **f**, Corresponding MS spectrum at compound QC LCMS EIC peak apex. **g**, SDS-PAGE of purified proteins. **h**, Offline ASMS recovery of 5 μM JNJ-9676 with 5 μM M-protein only, Fab-E only, Fab-B only, and Fab-B/Fab-E/M protein complex alongside buffer control with negligible breakthrough (n = 4 technical replicates, error bars indicate standard error of the mean). **i**, Analytical size exclusion chromatography of SARS-CoV-2 M, Fab-B and SARS-CoV-2 M/Fab-B complex. EIC, extracted ion chromatogram; ppm, part per million; MS, mass spectrometry; QC, quality control; LCMS, liquid chromatography–mass spectrometry; MM, molecular weight marker (kDa).

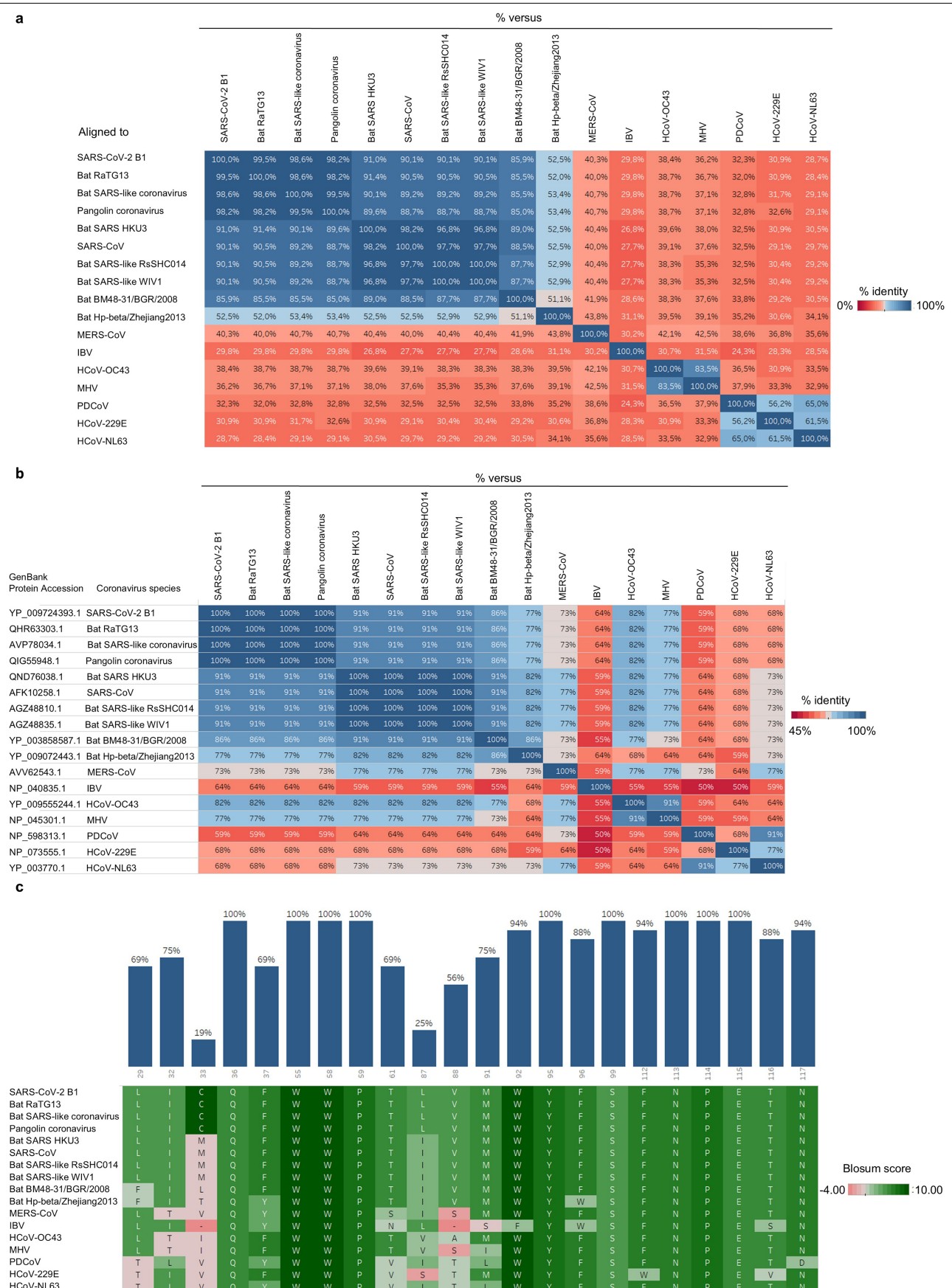

**Extended Data Fig. 3 |** See next page for caption.

**Extended Data Fig. 3 | Sequence conservation of the M protein and its binding pocket. a**, M protein sequence conservation across coronaviruses based on genomic sequence alignment. The B1 strain of SARS-CoV-2 genomic sequence of the M protein was aligned with corresponding sequences from viruses from the zoonotic reservoir from the sarbecovirus subfamily, Middle East respiratory syndrome virus (MERS-CoV), infectious bronchitis virus, HCoV-OC43, murine hepatitis virus, porcine epidemic diarrhoea virus, HCoV-229E, and HCoV-NL63. **b**, M protein sequence conservation of the binding pocket based on genomic sequence alignment. Sequence identity of the B1 strain of SARS-CoV-2 to sequences in the sarbecovirus family is > 90% for positions corresponding to the binding pocket of JNJ-9676. **c**, The M protein amino acid sequence in the binding pocket based on cryo-EM data was compared between the different viruses listed above based on the average Blosum score. To further assess the binding pocket conservation in sarbecoviruses, we downloaded the protein sequences classified into the sarbecovirus family (206 sequences, downloaded 25 June 2024) from the InterPro database. InterPro entry: IPR044361, M matrix/glycoprotein, SARS-CoV-like). Conservation analysis shows that 20 out of the 22 residues in the binding pocket of JNJ-9676 are completely conserved in proteins from the sarbecovirus family. The non-conserved positions correspond to SARS-CoV-2 residues C33 and I87.

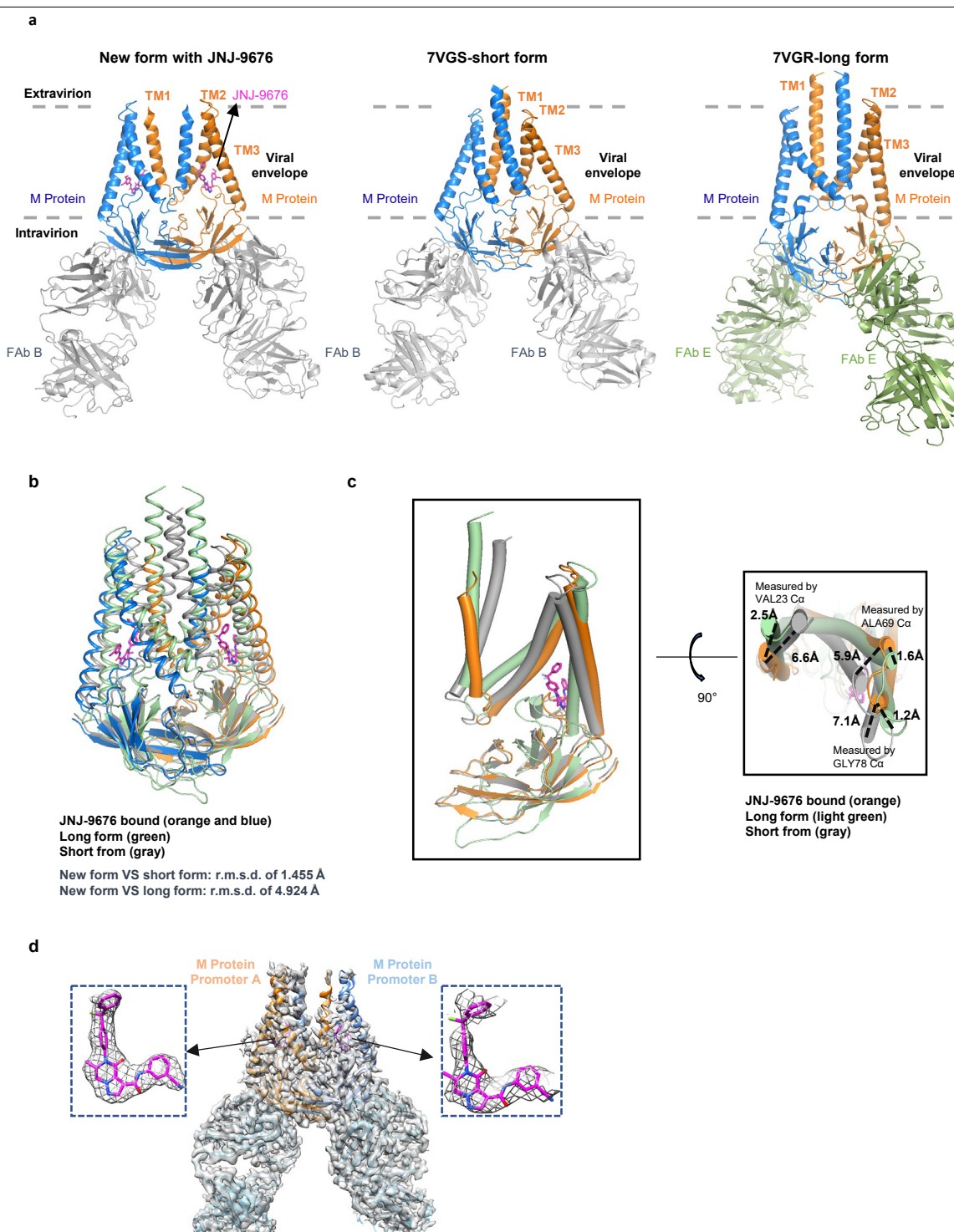

**Extended Data Fig. 4 | Three M protein dimer conformations. a**, Cryo-EM ribbon models: JNJ-9676 bound (new conformation) and unbound M/Fab-B complexes (short-form, PDB access code 7VGS), M/Fab-E (long-form, PDB access code 7VGR). **b**, Structural comparison by superimposition of the short-form (grey), the JNJ-9676 bound form (orange and blue), and the long-form (light green). Alignments based on all Cα atoms in the full M dimer structure. **c**, Structural comparison of protomer A by superimposition of the JNJ-9676 bound M protein, the unbound M protein (short-form), and the unbound long-form indicated in orange, grey, and light green, respectively. **d**, Cryo-EM map and model of the M protein dimer/Fab-B complex with JNJ-9676, featuring insets that shows JNJ-9676 density in individual protomer A and protomer B. The density map in the inset is displayed at a 0.145 threshold. Protomer A is coloured orange, protomer B is coloured blue, and JNJ-9676 is coloured magenta.

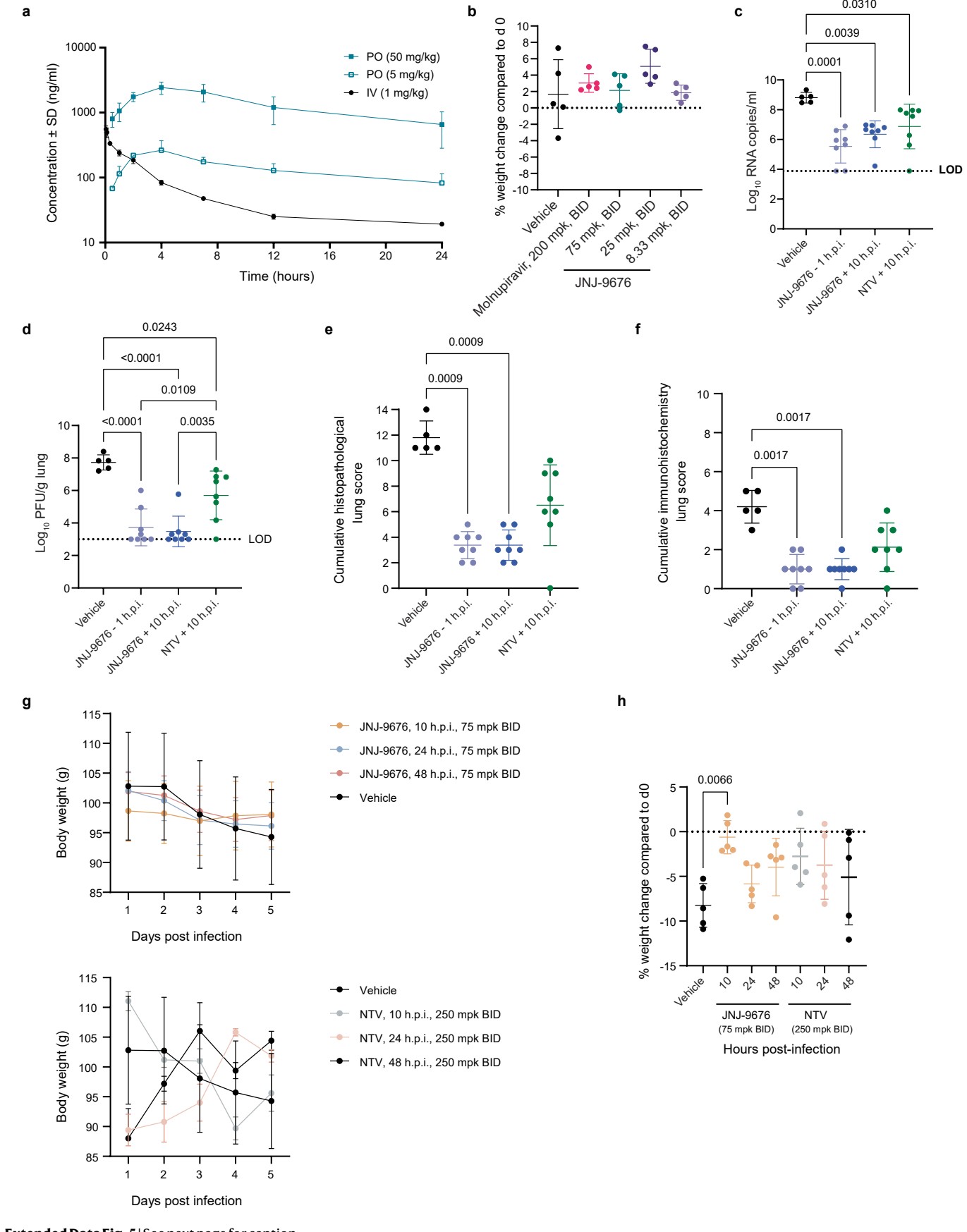

**Extended Data Fig. 5** | See next page for caption.

**Extended Data Fig. 5 | JNJ-9676 and reference compounds as pre- or post-exposure treatment in Syrian golden hamsters infected with SARS-CoV-2 B1.**
**a**, Single-dose pharmacokinetic profiles of JNJ-9676 in Syrian golden hamster. Mean ± standard deviations (n = 3 animals/group). **b**, Pre-exposure treatment with 8.33, 25 or 75 mpk JNJ-9676 or 200 mpk molnupiravir (n = 5 animals/group). Percentage bodyweight change on day 4. Individual data points (n = 5 animals/group) represent mean ± standard deviation. Mean differences between groups are calculated using one-way ANOVA with Šídák's multiplicity correction. **c-f**, Post-exposure treatment with 75mpk JNJ-9676 or 250 mpk nirmatrelvir (treatment one hour before infection or 10 h after infection) (n = 5 animals in the vehicle group, n = 8 animals in other groups). **c,d**, Individual data points (n = 5 animals/group) represent mean ± standard deviation. Mean differences between groups are estimated as in **b**. **c**, Viral load in the lung. Values below the limit of detection (LOD) were imputed to 3.89 $\log_{10}$ copies/mL (LOD value). p = 0.0001 for JNJ-9676-1 h.p.i. versus vehicle, p = 0.0039 for JNJ-9676+10 h.p.i. versus vehicle, p = 0.0310 for NTV+10 h.p.i. versus vehicle. **d**, Infectious virus in the lung. Values below the LOD were imputed to 3 $\log_{10}$ plaque-forming units/g (LOD value). p < 0.0001 for JNJ-9676-1 h.p.i. versus vehicle, p < 0.0001 for JNJ-9676+10 h.p.i. versus vehicle, p = 0.0243 for NTV+10 h.p.i. versus vehicle, p = 0.0109 for JNJ-9676-1 h.p.i. versus NTV+10 h.p.i. and p = 0.0035 for JNJ-9676+10 h.p.i. versus NTV+10 h.p.i. **e,f**, Individual data points per group (median and 95% confidence intervals). Differences between groups are calculated using the non-parametric Kruskal-Wallis test with the original false discovery rate method of Benjamini and Hochberg multiplicity correction. **e**, Histopathology score in the lung. p = 0.0009 for JNJ-9676-1 h.p.i. versus vehicle, p = 0.0009 for JNJ-9676+10 h.p.i. versus vehicle. **f**, Immunohistology in the lung. p = 0.0017 for JNJ-9676-1 h.p.i. versus vehicle, p = 0.0017 for JNJ-9676+10 h.p.i. versus vehicle. **g,h**, Post-exposure treatment with 75 mpk JNJ-9676 or 250 mpk nirmatrelvir (treatment 10-, 24- or 48-hours post-infection) (n = 5 animals/group). Mean differences between groups are estimated as in **b**. **g**, Mean body weight ± standard deviation over time. **h**, Mean percentage bodyweight change ± standard deviation on day 4. Mean differences between groups are estimated as in **b**. p = 0.0066 for JNJ-9676+10 h.p.i. 75 mkp BID. h.p.i., hours post-infection; BID, twice daily; mpk, milligrams/kilogram bodyweight; LOD, limit of detection. The doses reflect the amount of compound given each administration.

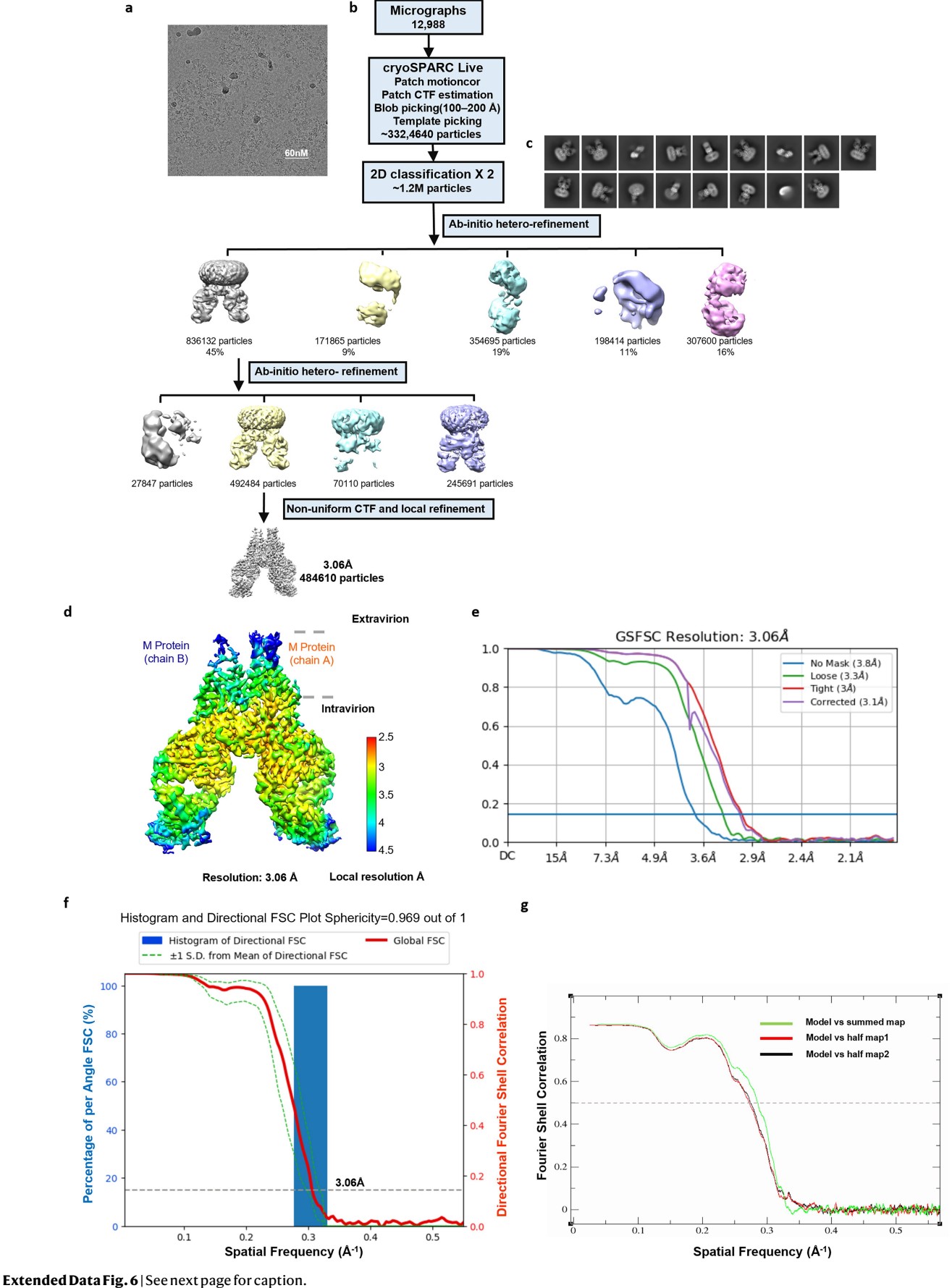

**Extended Data Fig. 6** | See next page for caption.

**Extended Data Fig. 6 | Cryo-EM analysis of SARS-CoV-2_MProtein_FAb B_ JNJ-9676. a**, Micrograph from SARS-CoV-2_MProtein_FAb B_JNJ-9676 data collection. **b**, Workflow: cryo-EM data analysis for SARS-CoV-2_MProtein_FAb B_JNJ-9676 map. CryoSPARC Live was used for real-time data processing. The non-uniform analysis yielded a 3.06-Å resolution 3D map. **c**, Representative 2D class averages of SARS-CoV-2_MProtein_FAb B_JNJ-9676. **d**, Local resolution coloured by ResMap estimation. **e**, Average resolutions estimated using 0.143 criterion of gold standard Fourier shell correlation (GSFSC). **f**, Anisotropy assessed by 3D-FSC server, showing complex sphericity of 0.969. **g**, Model validation through FSC curve comparisons: model versus half map 1 (work), model versus half map 2 (free), model versus full map in red, black, and green.

**Extended Data Table 1 | Antiviral activity of JNJ-9676 against different coronaviruses**

| Cell type | Virus | Median EC$_{50}$ (µM) | IQR (µM) | Median EC$_{90}$ (µM) | IQR (µM) | Median CC$_{50}$ (µM) | N |
|---|---|---|---|---|---|---|---|
| Huh7 | HCoV-229E (alphaCoV) | >30 | - | >30 | - | >30 | 2 |
| HeLa-hACE2 | HCoV-229E (alphaCoV | >12.5 | - | >12.5 | - | >12.5 | 2 |
| LLC-MK2 | HCoV-NL63 (alphaCoV) | >20 | - | >20 | - | >20 | 2 |
| MRC-5 | HCOV-OC43 (betaCoV) | 3.9 | 3.0–4.7 | 4.0[a] | - | >50 | 4 |
| A549-hACE2 | SARS-CoV (betaCoV) | 0.021 | 0.017–0.026 | 0.045 | 0.031–0.050 | >25 | 5 |
| Vero | SARS-CoV (betaCoV) | 0.016 | - | 0.020 | - | >50 | 2 |
| Huh7 | MHV (betaCoV) | 3.4 | - | 12.4 | - | >20 | 2 |
| HeLa | MHV (betaCoV) | 1.3 | - | 6.3 | - | >20 | 1 |
| Huh7 | MERS-CoV (betaCoV) | 0.6 | - | NA[b] | - | >20 | 2 |
| Huh7 | IBV (gammaCoV) | 8.0 | 5.3–12.6 | >30[b] | | >30 | 4 |
| Huh7 | PDCoV (deltaCoV) | >30[b] | - | >30[b] | - | >30 | 2 |

CC$_{50}$, 50% cytotoxic concentration; CoV, coronavirus; EC$_x$, x% effective concentration; IBV, infectious bronchitis virus; IQR, interquartile range; MERS-CoV, Middle East respiratory syndrome coronavirus; MHV, mouse hepatitis virus; N, number of independent technical repeats; NA, not available; PDCoV, porcine deltacoronavirus; SARS-CoV, severe acute respiratory syndrome coronavirus.

For N=2, no IQR could be calculated. For N=1, no median nor IQR could be calculated.

[a]In one experiment the EC$_{90}$ could be calculated, in other experiments HCoV-OC43 inhibition did not reach 100%, the EC$_{50}$ value is a relative EC$_{50}$, the EC$_{50}$ value is a relative EC$_{50}$.

[b]MERS-CoV inhibition did not reach 100%, median EC$_{90}$ values could not be calculated, the EC$_{50}$ value is a relative EC$_{50}$.

**Extended Data Table 2 | Summary of the mutations arising in in vitro resistance selection with SARS-CoV-2 B1, SARS-CoV-2 B1.617.2 (Delta) and SARS-CoV-2 B1.1.529 (Omicron)**

| Protein | JNJ-9676-*in vitro* selected SARS-CoV-2 variants | | |
| --- | --- | --- | --- |
| | **SARS-CoV-2 B1**<br>(from passages 10, 15, 17, 19, 23, 25) | **SARS-CoV-2 Delta**<br>(from passages 4, 7, 13, 17, 20) | **SARS-CoV-2 Omicron**<br>(from passages 10, 15, 20, 25, 30, 33) |
| nsp1 | G7V, <u>Q15K</u>, <u>R77Q</u> | | V84G, M85H, M85Q |
| nsp2 | P352S | Y49H, T153M | S68N, E563D |
| nsp3 | V207L, T459A, R883I, F1510V, <u>P1787S</u>, S1843F | Y693F, L822P, P1200S, C1223G | S660F, E1091A, E1245A, T1347I |
| nsp4 (DMV formation) | | S137L | T73I |
| nsp5 (3CLpro) | V186F | Q306H | |
| nsp10 | N114T, M122V | | |
| nsp12 (RdRp + NiRAN) | V720F | C84S, V102I | H801Q |
| nsp13 (helicase) | D204A, V348G, A362V, S535A, D578A | S229A | Y180C, T239I, N361Y |
| nsp14 (exonuclease/methyltransferase) | S134A, D179A, K304E, A425S | | P203S |
| nsp15 (endoRNAse) | G146V, D335E | | |
| nsp16 (2'-ORM) | K76R | D179E | D75G, L100P |
| Spike | <u>A27S</u>, <u>Y28N</u>, W64R, T76I, M153T, <u>E298N</u>, <u>Y313K</u>, F543I, <u>T678S</u>, <u>N679Y</u>, T724P, K795R, <u>S810L</u>, S813I | Y311K, Q990H, T1004R | R643S, R679Q, T729I |
| ORF3a | F114L, F114V, W128L, T269M | N257D | L52F, I62T |
| E | <u>T9I</u>, K53R | <u>T9I</u> | |
| M | <u>L29F</u>, A40P, A85S, A98D, <u>N117K</u>, <u>P132S</u>, E135V, <u>L138I</u>, L138P, <u>S173P</u>, Q185K | <u>L29F</u>, Y47H, <u>F103L</u>, N113S, P114L, <u>N117K</u>, N117S, <u>P132S</u>, <u>L138I</u>, <u>S173P</u> | <u>L29F</u>, L51V, L87H, L90W, <u>F103L</u>, T130S, L145F, H155N, <u>S173P</u>, R200K |
| ORF6 | T45I | T45P | T21I |
| ORF7a | | S45F, S45L, Y46E, Y46F | |
| ORF8 | | C83P | |
| N | T205I, T205S, <u>N213Y</u> | S37P, S194L, D402E | T195I, G209S |
| ORF10 | R24C | | |

Nsp, nonstructural protein; 3CLpro, 3-chymotrypsin-like cysteine protease; RdRp, RNA-dependent RNA polymerase; endoRNAse, endoribonuclease; 2-ORM, 2'-O-ribose methyltransferase; E, envelope; M, membrane; N, nucleocapsid; ORF, open reading frame.

Only mutations with a read frequency ≥15% are listed in the table. Identical mutations observed across different resistance virus samples are underlined.

**Extended Data Table 3 | Mean EC$_{50}$ fold changes in drug resistance potency of SDMs**

| Amino acid change in SARS-CoV-2 M | Potency shift (fold change ± SD) |
| --- | --- |
| A85S | 0.64 ± 0.21 |
| W55F | 0.62 ± 0.26 |
| A98D | 2.02 ± 0.58 |
| L90W | 2.47 ± 1.48 |
| L29F | 5.57 ± 0.42 |
| Q185K | 5.55 ± 1.92 |
| M91K | 10.03 ± 3.28 |
| P132S | 43.00 ± 13.42 |
| S99A | 96.90 ± 35.84 |
| N117K | 145.32 ± 52.05 |

EC$_{50}$, 50% effective concentration; SARS-CoV, severe acute respiratory syndrome coronavirus; SD, standard deviations; SDM, site-directed mutants.

**Extended Data Table 4 | Single-dose pharmacokinetic properties of JNJ-9676 in rat, Syrian golden hamster, dog and cynomolgus monkey (after intravenous oral [po] administration) and viral load and infectious virus reduction in Syrian golden hamsters**

(after intravenous oral [po] administration) and viral load and infectious virus reduction in Syrian golden hamsters

| | Rat | Hamster | Dog | Cynomolgus monkey |
|---|---|---|---|---|
| Clearance (ml/min/kg) | 5.2 | 9.0 | <1.5 | 1.9 |
| Volume of distribution (L/kg) | 8.4 | 7.9 | 8.2 | 5.1 |
| Plasma half-life (hours) | 19 | 13 | >40 | 35 |
| Oral bioavailability (%) | >70 | >50 | >80 | >30 |

**Viral load and infectious virus reduction for treatment relative to vehicle in Syrian golden hamsters infected with SARS-CoV-2 B1 and treated with JNJ-9676 pre- and post-exposure.**

| | JNJ-9676 treatment pre-exposure | | |
|---|---|---|---|
| | 75 mpk BID | 25 mpk BID | 8.33 mpk BID |
| Viral load in the lung ($\log_{10}$ copies/mg lung) | 3.01 (p=0.0019) | 3.51 (p=0.0003 | 2.01 (p>0.05) |
| Infectious virus in the lung ($\log_{10}$ $TCID_{50}$/mg lung) | 4.14 (p=0.03) | 4.41 (p=0.02) | 0.50 (p>0.05) |

| | JNJ-9676 treatment post-exposure (75 mpk BID) | | |
|---|---|---|---|
| | 10 hpi | 24 hpi | 48 hpi |
| Viral load in the lung ($\log_{10}$ copies/mg lung) | 2.38 (p=0.0004) | 2.32 (p=0.0005) | 1.38 (p>0.05) |
| Infectious virus in the lung ($\log_{10}$ $TCID_{50}$/mg lung) | 1.65 (p=0.0003) | 2.00 (p<0.0001) | 1.83 (p<0.0001) |

BID, twice daily; mpk, milligrams/ per kilogram body weight; SARS-CoV-2, severe acute respiratory syndrome coronavirus; $TCID_{50}$, 50% tissue culture infective dose.

**Extended Data Table 5 | Overview of site-directed mutant MOI in JNJ-9676 HCI-based antiviral assays**

| Recombinant virus strain | MOI |
| --- | --- |
| rcSARS-CoV-2 WT | 0.10 |
| SARS-CoV-2_M:L29F | 0.065 |
| SARS-CoV-2_M:P132S | 0.25 |
| SARS-CoV-2_M:N117K | 0.29 |
| SARS-CoV-2_M:M91K | 0.073 |
| SARS-CoV-2_M:S99A | 0.076 |
| SARS-CoV-2_M:Q185K | 0.12 |
| SARS-CoV-2_M:L90W | 0.014 |
| SARS-CoV-2_M:W55F | 0.010 |
| SARS-CoV-2_M:A85S | 0.072 |
| SARS-CoV-2 M:A98D | 0.060 |

MOI, multiplicity of infection; HCI, high-content imaging.

## Extended Data Table 6 | Cryo-EM data collection, refinement and validation statistics

| | SARS-CoV-2_MProtein_FAb7vgs_JNJ-9676 (EMDB-43745) (PDB 8W2E) |
|---|---|
| **Data collection and processing** | |
| Magnification | 105,000 |
| Voltage (kV) | 200 |
| Electron exposure (electrons/$Å^2$) | 40 |
| Defocus range (μm) | -0.6 to -2.0 |
| Pixel size (Å) | 0.910 |
| Symmetry imposed | C1 |
| Initial particle images (no.) | 3,324,640 |
| Final particle images (no.) | 484,616 |
| Map resolution (Å) | 3.06 |
| FSC threshold | 0.143 |
| Map resolution range (Å) | 2.5–5 |
| **Refinement** | |
| Initial model used (PDB code) | 7vgs |
| Model resolution (Å) | 3.06 |
| FSC threshold | 0.143 |
| Model resolution range (Å) | 2.9–3.3 |
| Map sharpening $B$ factor ($Å^2$) | |
| Model composition | |
| Non-hydrogen atoms | 9373 |
| Protein residues | 1205 |
| Ligands | 2 |
| $B$ factors ($Å^2$) (min/max/mean) | |
| Protein | 11.55./211.73/90.64 |
| Ligand | 69.39/136.73103.06 |
| R.m.s. deviations | |
| Bond lengths (Å) | 0.003 |
| Bond angles (°) | 0.658 |
| Validation | |
| MolProbity score | 1.98 |
| Clashscore | 8.01 |
| Rotamers Outlier (%) | 0.20 |
| Ramachandran plot | |
| Favored (%) | 90.21 |
| Allowed (%) | 9.79 |
| Outliers (%) | 0.00 |

EMDB, Electron Microscopy Data Bank; FSC, Fourier shell correlation; PDB, protein data bank; r.m.s., root mean square.

|  |  |
|---|---|

# Reporting Summary

## Statistics

For all statistical analyses, confirm that the following items are present in the figure legend, table legend, main text, or Methods section.

| n/a | Confirmed | |
|---|---|---|
| ☐ | ☒ | The exact sample size (*n*) for each experimental group/condition, given as a discrete number and unit of measurement |
| ☐ | ☒ | A statement on whether measurements were taken from distinct samples or whether the same sample was measured repeatedly |
| ☐ | ☒ | The statistical test(s) used AND whether they are one- or two-sided *Only common tests should be described solely by name; describe more complex techniques in the Methods section.* |
| ☒ | ☐ | A description of all covariates tested |
| ☐ | ☒ | A description of any assumptions or corrections, such as tests of normality and adjustment for multiple comparisons |
| ☐ | ☒ | A full description of the statistical parameters including central tendency (e.g. means) or other basic estimates (e.g. regression coefficient) AND variation (e.g. standard deviation) or associated estimates of uncertainty (e.g. confidence intervals) |
| ☐ | ☒ | For null hypothesis testing, the test statistic (e.g. *F*, *t*, *r*) with confidence intervals, effect sizes, degrees of freedom and *P* value noted *Give P values as exact values whenever suitable.* |
| ☒ | ☐ | For Bayesian analysis, information on the choice of priors and Markov chain Monte Carlo settings |
| ☒ | ☐ | For hierarchical and complex designs, identification of the appropriate level for tests and full reporting of outcomes |
| ☒ | ☐ | Estimates of effect sizes (e.g. Cohen's *d*, Pearson's *r*), indicating how they were calculated |

*Our web collection on statistics for biologists contains articles on many of the points above.*

## Software and code

Policy information about availability of computer code

| Data collection | Antiviral activity against SARS-CoV, SARS-CoV-2 and 229E was done by high content imaging in A549-hACE2 cells on a Cell Voyager 8000 (Yokogawa) confocal microscope whereas antiviral activity VeroE6-eGFP using HCI was done on a Arrayscan XTI (Thermofisher). MTS assays to assess toxicity were read out on a Spark plate reader (Tecan). |
|---|---|
| | Antiviral activity against zoonotic viruses was measured using NanoGlo on a Glomax plate reader (Promega). |
| | Antiviral activity against IBV, MHV and PDCoV was done using a GloMax® Discover Microplate Reader (Promega). |
| | Antiviral activity against OC43 and NL63 was measured on a BioTek spectrophotometer. |
| | Antiviral activity against MERS was measured on an Envision multimode plate reader (Perkin Elmer). |
| | All RNA extractions were automated on a MagNA Pure instrument (Roche) and RT-qPCR readouts were obtained on a LightCycler 480 real-time PCR instrument (Roche). |
| | For viral yield studies, RNA extraction and RT-qPCR results were obtained as listed above. The parallel toxicity readout was done on a Viewlux instrument (PerkinElmer). |
| | In ALI cultures, antiviral activity was measured by RT-qPCR as mentioned above. TEER toxicity measurements on ALI cultures were performed using the EVOM3 (World Precision Instruments). |
| | NanoDSF measurements were taken on a Prometheus NT.Plex instrument (NanoTemper Technologies). |
| | For ASMS, all liquid chromatography-mass spectrometry (LC-MS) analyses were performed on an 1290 Infinity II uHPLC system (Agilent) coupled to a 6545XT qTOF (Agilent). |
| | Cryo-EM data collection was automated on a 200 kV Glacios™ microscope (Thermo Scientific).Micrographs were taken at 105,000X magnification using a Facon4 detector (Gatan) in counting mode. |
| | 1H NMR spectra were recorded on a Bruker DPX-400 spectrometer. |
| | 1H-13C HMBC NMR spectra were recorded on a Bruker Avance-500 spectrometer. |

| Data analysis | All High content imaging analysis was done in Phaedra HCI analysis software (version 1.0.10.202309011029).<br>All antiviral data (EC50/90, CC50) was processed using Graphpad Prism (version 8 or 9).<br>Antiviral data in ALI cultures was processed in LightCycler software (Roche) and Graphpad Prism (version 8). Toxicity CC50 values were calculated in Graphpad Prism (version 8).<br>NanoDSF data were analyzed with PR. ThermControl v2.1.6 (NanoTemper Technologies).<br>ASMS data processing was performed using Agilent MassHunter Qualitative Analysis (v 10.0).<br>CryoEM structure representations were generated using Pymol (v2.0) and Chimera (v1.17.3).Cryo-EM data collection and image quality were monitored using cryoSPARC Live v3.2. Image. Local resolution was determined using ResMap. For the M/Fab-B complex model building, the M protein was manually built using COOT. The Fab-B was fitted into the 3D map using Chimera and then further refined manually with COOT followed by real-space refinement in Phenix.<br>The data was processed using the Bruker TOPSPIN program v4.1, and 1H and 13C chemical shifts were analyzed using ACD/Spectrus software 2023 v1.1.<br>All statistical analyses for in vivo experiments were performed in GraphPad Prism (version 9) and validated using R (version 3.6.1).<br>The amino acid sequences for the M protein were downloaded from https://www.ncbi.nlm.nih.gov/ (dated 2023/01/31) and aligned through a pairwise sequence alignment using the Needleman-Wunsch algorithm through the EMBOSS-Needle tool from EMBL-EBI (https://www.ebi.ac.uk/jdispatcher/psa/emboss_needle). All visualizations of the sequence alignments were made using Tableau Software (online version).<br>Graphs and figures were generated using Microsoft PowerPoint (Version 2308 Build 16731.20460), GraphPad Prism (v8 and 9), BioRender (free version, in vivo work), PyMOL Molecular Graphics System (Version 2.0), Chimera (version 1.17.3), CryoSparc (version 4.4.1), 3D-FSC (version 1.0), Grace (version 5.1.25) and Image Lab (version 6.0.1); the software is made available by Janssen Pharmaceutica NV. |

For manuscripts utilizing custom algorithms or software that are central to the research but not yet described in published literature, software must be made available to editors and reviewers. We strongly encourage code deposition in a community repository (e.g. GitHub). See the Nature Portfolio guidelines for submitting code & software for further information.

# Data

Policy information about availability of data

All manuscripts must include a data availability statement. This statement should provide the following information, where applicable:

- Accession codes, unique identifiers, or web links for publicly available datasets
- A description of any restrictions on data availability
- For clinical datasets or third party data, please ensure that the statement adheres to our policy

All data supporting the findings of this study are available within the article and all accession codes are provided in the manuscript.
Cryo-EM maps have been deposited in the Electron Microscopy Data Bank (accession code: EMD-43745), while the atomic coordinates of the M/Fab-B complex structures have been deposited in the Protein Data Bank (accession code: 8W2E). The PDB accession codes for the M/Fab-B complex short-form is 7VG and for the M/Fab-E long-form is 7VGR.
No cropped images of western blots are shown, the uncropped images of the western blots are presented in Extended Data Fig. 2g.

# Research involving human participants, their data, or biological material

Policy information about studies with human participants or human data. See also policy information about sex, gender (identity/presentation), and sexual orientation and race, ethnicity and racism.

| Reporting on sex and gender | No human participants were used in this study. |
| Reporting on race, ethnicity, or other socially relevant groupings | No human participants were used in this study. |
| Population characteristics | No human participants were used in this study. |
| Recruitment | No human participants were used in this study. |
| Ethics oversight | No human participants were used in this study. |

Note that full information on the approval of the study protocol must also be provided in the manuscript.

# Field-specific reporting

Please select the one below that is the best fit for your research. If you are not sure, read the appropriate sections before making your selection.

☒ Life sciences ☐ Behavioural & social sciences ☐ Ecological, evolutionary & environmental sciences

For a reference copy of the document with all sections, see nature.com/documents/nr-reporting-summary-flat.pdf

# Life sciences study design

All studies must disclose on these points even when the disclosure is negative.

| | |
|---|---|
| Sample size | For almost all antiviral in vitro studies, three or more independent experiments (using multiple technical replicates) were performed (Fig1b, Extended data table 1). Whenever possible, we strived to obtain data from three independent experiments, a common standard for biological experiments which allows to identify outliers or anomalies in the data. No formal size calculation was performed for in vitro experiments. In our analysis of the spectrum against which JNJ-9676 is active, some exceptions to the n=3 rule were made. Either because, in case of antiviral experiments with animal coronaviruses, the materials are very scarce and results were fully in line with those obtained with human coronaviruses. Or, in case of human coronaviruses other than SARS-CoV-2 or SARS-CoV, because antiviral activity was limited or absent and thus irrelevant for future human treatment. These data were merely used to showcase the spectrum against which JNJ-9676 acts. In the translational ALI model, three independent experiments were run to account for potential variation (Fig1c). Resistant selection experiments with the compound were obtained from three different (at the time) variants of concern of SARS-CoV-2 and includes analysis of multiple passages (Fig1d,e; Extended data table 2; Extended data fig1d). Three independent experiments were run with the site directed mutants to obtain EC50 fold changes (Fig1f). Fitness experiments were carried out in three independent experiments (8 technical replicates per experiment)(Extended data fig1g). One representative experiment is shown for the ASMS readout, three technical replicates are plotted (Fig1g). NanoDSF was carried out with the compound in three independent experiments (Fig1h). Time of addition studies were carried out multiple times, a representative experiment is shown with three technical replicates (Extended data fig1b). Given the elaborate evidence and the high labor intensity of these assays; this number of replicates was considered sufficient. Statistical power analysis as well the limitations of the study size warrented 5 animals per group to obtain statistical significance in in vivo Syrian golden hamster studies. |
| Data exclusions | No data was excluded from any of the studies reported in this paper. |
| Replication | Three or more independent experiments (either in duplicate or triplicate) were performed for almost all in vitro experiments and at least two independent experiments for almost all in vivo studies. All attempts at replication were consistent and reflect the intra and inter variability. |
| Randomization | Allocation of hamsters to experimental groups was performed randomly. For the in vitro experiments performed in this study, randomization was not relevant as no allocation to experimental treatment groups is required. Reference compounds and proper controls were taken along to assess consistency over time. We received consistent results over time with repeats performed on different days and by different people. |
| Blinding | For both the in vivo and in vitro experiments performed in the study, blinding was not applicable as no experimental treatment groups were used where the quality of the outcome could be influenced. |

# Reporting for specific materials, systems and methods

We require information from authors about some types of materials, experimental systems and methods used in many studies. Here, indicate whether each material, system or method listed is relevant to your study. If you are not sure if a list item applies to your research, read the appropriate section before selecting a response.

### Materials & experimental systems

| n/a | Involved in the study |
|---|---|
| ☐ | ☒ Antibodies |
| ☐ | ☒ Eukaryotic cell lines |
| ☒ | ☐ Palaeontology and archaeology |
| ☐ | ☒ Animals and other organisms |
| ☒ | ☐ Clinical data |
| ☒ | ☐ Dual use research of concern |
| ☒ | ☐ Plants |

### Methods

| n/a | Involved in the study |
|---|---|
| ☒ | ☐ ChIP-seq |
| ☒ | ☐ Flow cytometry |
| ☒ | ☐ MRI-based neuroimaging |

## Antibodies

| | |
|---|---|
| Antibodies used | SARS-CoV/SARS-CoV-2 nucleoprotein/nucleocapsid antibody, rabbit polyclonal antibody (Sino Biological, 40143-T62); primary anti- |

| Antibodies used | spike S1 monoclonal antibody (Recombinant, expressed from Hek293 cells; rabbit) (Sino Biological, cat. 40150-R007, Houston, TX, USA); primary anti-double-stranded RNA (dsRNA) monoclonal antibody J2 (mouse) (SCICONS, cat. 10010500, Jena Bioscience, Jena, Germany); secondary goat anti-mouse Polyclonal immunoglobulin G (IgG) secondary antibody, conjugated to Alexa Fluor 488 (goat anti-mouse) (cat. A11001, Invitrogen, Waltham, MA, USA); secondary goat anti-rabbit Polyclonal immunoglobulin G (IgG) secondary antibody, conjugated to Alexa Fluor 568 (cat. A11036, Invitrogen, Waltham, MA, USA), HQ Anti-Rabbit (cat. 07017812001, Roche), Anti-HQ HRP (cat. 07017936001, Roche). |
|---|---|
| Validation | All antibodies were obtained from commercial sources. The use of SARS-CoV/SARS-CoV-2 nucleoprotein/nucleocapsid antibody, rabbit polyclonal antibody (Sino Biological) was described here PMID: 38920116. The use of SARS-CoV/SARS-CoV-2 spike monoclonal antibody (Sino Biological); J2 mouse anti-dsRNA monoclonal antibody (Scicons); goat anti-rabbit polyclonal Ab conjugated to Alexa Fluor 5681 (Invitrogen); goat anti-mouse IgM polyclonal Ab conjugated to Alexa Fluor 488 (Invitrogen) was described here PMID: 38158129. |

## Eukaryotic cell lines

Policy information about cell lines and Sex and Gender in Research

| Cell line source(s) | Human epithelial cell line A549 stably expressing hACE2 (A549-hACE2) were obtained from InvivoGen (San Diego, USA) or from the American Type Culture Collection (ATCC; # CCL-185). VeroE6-eGFP were cloned and validated at Tibotec/Janssen Pharmaceutica NV (Beerse, Belgium). Pooled donor nasal epithelial cells grown in air-liquid interface (ALI) format were obtained from Epithelix as a fully differentiated culture (Plan-les-Ouates, Switserland). Hela cells were obtained from ATCC. HeLa-hACE2 cells were obtained from Creative Biogene (New York, USA). Huh7 were obtained from ATCC. LLC-MK2 cells were obtained from Evotec (Toulouse, France). MRC-5 cells were obtained from Evotec (Toulouse, France). |
|---|---|
| Authentication | Cell lines were not authenticated. |
| Mycoplasma contamination | All cell lines tested negative for mycoplasma contamination. |
| Commonly misidentified lines (See ICLAC register) | None of the commonly misidentified cell lines were used. |

## Animals and other research organisms

Policy information about studies involving animals; ARRIVE guidelines recommended for reporting animal research, and Sex and Gender in Research

| Laboratory animals | In pre-exposure studies, female Syrian golden hamsters of 6–8 weeks were used purchased from a Janvier Laboratories. In post-exposure studies, male Syrian golden hamsters of 6–8 weeks were used purchased from a Janvier Laboratories. |
|---|---|
| Wild animals | No wild animals were used in this study. |
| Reporting on sex | Findings do not apply to one sex only. |
| Field-collected samples | No field-collected samples were used in the study. |
| Ethics oversight | Pre-exposure experiments were performed at KU Leuven. Housing conditions and experimental procedures were performed as described in project 062/2020 as approved by the ethics committee of KU Leuven (Belgium) which is licensed under number LA1210186. Post-exposure experiments happened either at Evotec or in house at J&J Innovative Medicine. Housing conditions and experimental procedures were performed as described in project APAFIS#31467-2021041618563995 v3 and Proj 129-Proc 786 as approved by the ethics committee of Evotec (France) and Johnson&Johnson Innovative Medicine (Belgium) which are licensed under number E31555059 and LA1100119, respectively. |

Note that full information on the approval of the study protocol must also be provided in the manuscript.

## Plants

| | |
|---|---|
| Seed stocks | No plants were used in this study. |
| Novel plant genotypes | No plants were used in this study. |
| Authentication | No plants were used in this study. |

