## [Peer Review File · Nature]

A small-molecule SARS-CoV-2 inhibitor targeting the membrane protein

Corresponding Author: Dr Marnix Van Loock

Version 0:

Reviewer comments:

Referee #2

(Remarks to the Author)

The manuscript from Van Damme and colleagues described a small molecule that targets the coronavirus M protein. CryoEM was used to identify the binding pocket and conformational alteration upon binding to M protein dimers. The alteration prevents release of infectious virus as measure by amounts of RNA in the extracellular media off infected cells. A hamster model was used to show that the small molecule is effective in vivo. The inhibitor is broadly effect against SARS-1 and SAR-2 viruses, variants, and bat and panagolin SARs-like viruses. Overall, the results are very interesting and important in that a unique inhibitor targeting the M protein is described both structurally and functionally.

Comments

1. Abstract – Lines 53-54 – Based on the data it seems more accurate to indicate that JNJ-9676 binds within a pocket formed by the transmembrane domains of M dimerr which results in conformational changes.
2. Fig. 1d graphic annotation should be improved and the orientation corrected. The amino end should be shown as in the lumen and the carboxy tail in the cytoplasm as it is translated at the ER. However, it would be better to refer to orientation in the context of the virus instead (e.g. amino end in the extravirion or outside of virion envelope and carboxy tail as intravirion or inside the envelope. By simply moving the arrows a bit so they do not overlap the TMD, loops and tail (line) illustration and use a color other than black for the stars will improve the figure.
3. Line 164, page 9 – Q185L should be Q185K.
4. Lines 150-152, Extended Fig. 1 a-b – What concentration of inhibitor was added? It is not clear from the results that egress is inhibited. It seems more likely that virus assembly is inhibited. This could possibly be addressed with an inhibitor dose response experiment or EM of infected cells, but for this report it is sufficient to not specifically indicate that egress is inhibited and leave it at interference with biogenesis.
5. Page 10, lines 187-190 – Fab-B binds the short form of M? The specificity of Fab-B should be included here.

Referee #3

(Remarks to the Author)

A small-molecule SARS-CoV-2 inhibitor targeting the membrane protein

This manuscript reports JNJ-9676, which is a first-in-kind small compound targeting coronavirus M protein. The validity of the novel target is supported by the experiments conducted, and the effectiveness of coronavirus M protein inhibitor in the animal model is very compelling. Identification of a novel druggable target is a valuable addition to the existing collection of antiviral drugs against COVID-19, and the results presented are of significant interest and importance in coronavirus antiviral field. To further improve the quality of the manuscript, addressing specific points below is recommended. There are also many places such as figure legends and materials/methods section where descriptions are lacking or not clear.

Major points:

1. It is claimed that JNJ-9676 is broadly active against sarbecoviruses, which was partly supported by the strong inhibitory activity against bat coronaviruses SHC014 and WIV1, which are very close to SARS-CoV-2. It is assumed that all sarbecovirus M proteins share high homology (91%) but only a few sarbecoviruses were included in the analysis. More importantly, adding additional in-vitro inhibition data using more sarbecovirus M proteins would strengthen their claim of

broad-sarbecovirus activity.

2. Regarding viral resistance

It would be nice to include the EC50 changes from SARS-CoV-2 over passages, which would help to understand how fast resistance emerges in a live virus setting. The Materials/methods described that EC50s were determined at each passage number, so it would be easy to add a figure/table. Combined with Fig 1F table that shows fold changes due to single mutations introduced to recombinant mutant virus, and Ext. Fig1d that shows mutant virus population % over passages, a new figure/table will provide a more comprehensive view on the resistance barrier to this compound.

Ext. Fig 1d and g: The recombinant SARS-CoV-2 carrying each mutation show very similar replication efficiency (g), but the predominant mutations (d) vary among different SARS-CoV-2 strains and replicates. What would be the potential explanation for this observation?

Ext. Fig 1d and g: please add how these experiments were conducted and analyzed in detail. In Ext. Fig1d, multiple mutations were found in 100% of viral population at some passage numbers, which suggest multiple mutations with >50-100-fold increase in EC50 emerge quickly during 10-15 passages. Please add discussion on its clinical implications as antiviral drugs.

3. Time of Addition assay

Ext. Fig 1a/b, p150-152: Viral replication was inhibited when the compound is incubated with cells for 0-1 hr (viral entry step). Does it mean that the inhibitor blocks viral entry or uncoating as well by inhibiting generating mature virions? Or could there be another mechanism for this inhibition?

4. Mouse study: Are the extended Fig 5C table and the Fig 3b-h (log10 lung virus copies and lung TCID50) from the same experiment? The numbers in the table and the figures do not match.

Minor point:

Fig 1a is confusing – please revise the figure for clarification.

Fig. 3e – histopath photos have lower resolution. Photos of higher resolution and magnification would be helpful.

Fig. 3 – histopath data from post-exposure experiment is not included.

Fig. 3 – The inhibitor was given BID but the figure shows 8hr interval.

Fig. 3 - title sarbecovirus needs to be changed to SARS-CoV-2.

It is better to move the extended Fig 5C table to Fig 3g and h.

Please consider adding the resistance data to main manuscript (Extended fig1d and add EC50 values)

Treatment dose of mg/kg BID – 75 mg/kg BID is per oral treatment or per day (in total)? Please specify.

Ext. Fig 1b. What is the purpose of including 10-1 and 10-2 virus dilutions in the figure?

Referee #4

(Remarks to the Author)

Thanks for inviting me to review Nature manuscript 2024-02-03252. Let me first declare my limitations in synthetic chemistry and structural biology as a clinical virologist.

This international team synthesized a small molecule JNJ-9676 which is an analogue of a compound series with nanomolar activity against SARS-CoV-2 identified previously from chemical library screening. Vero E6 cell culture assay confirmed its low EC50 against many representative strains of Sarbecoviruses while it is less potent against other non-sarbeco-coronaviruses. Its potent activity is also confirmed in primary human nasal epithelial cells grown in air-liquid interface. Time of addition experiments suggested that it interferes with the production of virions. Serial passage of SARS-CoV-2 B1 strain in the presence of gradually increasing concentration of this compound selects for escape mutations mainly found in M protein which resides near the dimer interface. Reverse genetically engineered M viral mutants showed 43 to 145 fold increased phenotypic resistance to this compound. Drug target interaction between JNJ-9676 with M protein were clearly demonstrated with two different binding assays including offline affinity selection-mass spectrometry and nano differential scanning fluorimetry. Cryogenic EM controlled by a fiducial marker antibody fragment showed that SARS-CoV-2 M protomer A complexed better with the compound. The compound adopts a rotated L shaped configuration and binds to an induced pocket formed by TM2 and TM3 of protomer A and TM1 of protomer B with significant conformational change to a novel conformational state. Finally oral JNJ-9676 reduced virus titre and histopathological damage in the lung of hamsters challenged by SARS-CoV-2 when the compound is given before and even 48 hours after virus inoculation. The authors concluded that M protein is an attractive novel antiviral target and JNJ-9676 represents a potential drug candidate.

Strength

1. The finding of JNJ-9676 targeting a structural protein M is novel.
2. The study is very comprehensive with a representative set of sarbecoviruses included in the experiments.
3. The compound has an impressive potency of EC50 at nanomolar concentration and can be given orally.
4. The mechanism of its inhibitory activity is very well demonstrated by the time of addition experiments, induction of resistant escape mutant viruses, confirmation by reverse genetically engineered mutant virus with phenotypic resistance, binding and interaction assays, and finally the conformational changes of M induced by the compound under the cryoEM.

Weakness

1. The finding of broad inhibitor against structural M or N protein of virus is not a new concept. Both have been reported for

- influenza viruses. These seminal papers should be discussed. Broad Spectrum Anti-Influenza Agents by Inhibiting Self-Association of Matrix Protein 1 - PMC <https://www.ncbi.nlm.nih.gov/pmc/articles/PMC5004101/> (nih.gov); Identification of influenza A nucleoprotein as an antiviral target | Nature Biotechnology <https://www.nature.com/articles/nbt.1638>
2. The authors claim that JNJ-9676 was an analogue of a compound series that was identified in the screening campaign published in J Med Virol in reference 37. But there is no information in that paper which refers to this compound series which is now represented by the analogue JNJ9676.
 3. No positive control (such as nirmatrelvir even though it has a different antiviral mechanism) was included in the cell culture or hamster challenge experiments.
 4. The relative amount of short M and long M after JNJ-9686 treatment of transfected cell line for the production of virus like particles should be assayed to demonstrate the mechanism of inhibitor.
 5. An experiment should be performed to see if the compound can reduce the infectivity of cell free SARS-CoV-2 virion (with long M) after in vitro incubation with JNJ-9676.
 6. How the recombinant viruses with mutations were generated are not clear. By simply refer to the references are not enough. The authors should show the plaque pictures of each recombinant viruses. It is still not clear why resistant mutations, such as the T91 E protein which is consistently positive in 2 out of 3 columns, were not chosen for reverse genetic investigation (Extended Data Table 2).
 7. The exact binding affinity Kd value between JNJ-9676 and M protein, wild type and mutant, was not determined.
 8. The pharmacokinetic data of JNJ-9676 cannot be found in the file (Please indicate it as the reviewer may have missed it).

Minor comments

1. Line 52: SARS-CoV2 shall be SARS-CoV-2
2. mock-infection control is lacking in figure 2e.

Recommendation

The authors should address all the weakness before publication

Referee #5

(Remarks to the Author)

Van Damme, et al. report on the discovery of a small molecule inhibitor of SARS-CoV-2 assembly targeting the M protein. This is exciting work, well performed, and clearly and concisely presented. It provides important insights into the role of M protein in coronavirus assembly and strong evidence supporting its suitability as a novel target for therapeutics. I have only several minor concerns and suggestions for the manuscript.

1. Did the authors attempt to determine a structure or assess interaction of JNJ-9676 with M in the long conformation? If not, how was it decided to pursue only a cryo-EM structure of M in the short conformation bound to the small molecule?
2. A more thorough explanation of the various escape mutants identified would be an interesting addition. For example, can they be grouped into those expected to directly occlude binding through steric or electrostatic repulsion versus those rather expected to bias the conformational equilibrium between long and short conformations and differentially colored in Figure 1e?
3. Figure 2f should be improved for clarity. Perhaps change to make the overlaid apo structure gray or some other color?
4. The clear difference in resolvability of the two M subunits is interesting. Can the authors elaborate on how JNJ-9676 was docked and refined into the second subunit with weaker map features?
5. No large-scale issues are evident in the model, but the structure shows overall relatively poor refined geometry for the resolution and map quality (particularly in the Ramachandran plot and clash statistics). The authors should consider additional refinement with either an improved starting model or with increased geometry weight in real space refinement. If issues are confined to the less well-defined subunit, weak NCS constraints could be applied as an alternative approach.
6. This is subjective, but I would be inclined to describe the conformation of M captured as a distorted version the previously described short conformation rather than a "new" form. Conformational differences are seemingly well described by a simple coordinated movement of TM1s required for docking of the short arm of the JNJ-9676 "L" including the benzyl group and associated movement of the tops of TM2-3. Whether this is a populated alternative conformation or intermediate in the absence of JNJ-9676 is unclear.

Referee #6

(Remarks to the Author)

In this investigation, the authors discovered a new SARS-CoV-2 inhibitor, called JNJ-9676, by a phenotype based high-throughput screening. The biological target of this compound was identified as the M protein by an in vitro resistance selection (IVRS) assay. JNJ-9676 demonstrated nanomolar antiviral activity against various sarbecovirus types including SARS-CoV2, SARS-CoV, and others in in vitro antiviral assays. Cryo-EM was used to determine the complex structure of JNJ-9676 – the M protein, which showed that JNJ-9676 binds to a new pocket in the M protein's transmembrane domain, stabilizing the M protein in an altered conformational state between its long- and short-forms, hence preventing the release of infectious virus. This compound also displayed good in vivo anti-viral activity in preclinical animal models. From the reviewer's point of view, this study is a very preliminary work, needing to be further improved.

My major concerns:

- 1) How was JNJ-9676 discovered? More detailed description is needed, although structure-activity relationship (SAR)

analysis of this series compounds is not necessary.

2) JNJ-9676 was not well characterized. Only nanoDSF was used to measure the binding of JNJ-9676 with the M protein. A ΔT_m value of 0.9°C indicates a poor binding affinity, at least not good. Why did the authors not use more other approaches to confirm this binding, such as ITC, SPR?

3) The author only tested the antiviral activity of JNJ-9676 in hamsters. It is better to evaluate it in humanized transgenic mice. In addition, it is also necessary to explore the effects of the M protein inhibitor on the immune system during viral infection.

4) In vitro and in vivo antiviral experiments lack a positive control, making it difficult to objectively compare the effects of JNJ-9676. It is recommended to use paxlovid (nirmatrelvir) as a positive control, and show whether JNJ-9676 has some advantages.

5) As an oral medication, please evaluate the pharmacokinetic properties of JNJ-9676 in related species and provide detailed parameters.

Minors:

6) What is the solvent used in the in vivo assay?

7) How about the target selectivity of JNJ-9676?

8) Please provide the compound characterization data of JNJ-9676, including ¹H NMR and ¹³C NMR, HPLC spectra, etc.

Version 1:

Reviewer comments:

Referee #2

(Remarks to the Author)

Manuscript 2024-02-03252A from Van Damme and colleagues describes a small molecule that targets the coronavirus M protein. Results with the inhibitor in an animal model are promising to provide a new therapeutic antiviral against a new coronavirus target. The revised manuscript addresses the major comments on the initial submission. The manuscript overall is improved based on the responses to earlier questions and suggestions from the reviewers.

Referee #3

(Remarks to the Author)

The authors has put a great deal of efforts to address the points raised by the reviewers in general. However, there are some points that need further clarification as listed below. No additional data or experiments are deemed necessary with this revision in my view, but more careful perusal of the manuscript by the authors is required to make sure all figures and data sets are free of errors.

-Entry inhibition with JNJ-9676 (0-1 hr incubation): The authors mentioned in the rebuttal that the EM pictures showed viral entry is not affected by the compound because there is no virus in the reticulovesicular network or released virions at 10 hr post infection. Although it is possible that virion assembly is the major key mechanism of viral infection by the compound, the EM pictures do not serve as evidence that virion uncoating/entry is not affected. Endosomal pathway (2-6 hr post infection) needs to be monitored in ACE2-expressing cells for failure of viral uncoating/entry (as accumulated virus particles), and I wonder if the authors also looked at these subcellular locations. It is okay to leave this for future study, but as the time-of-addition clearly indicates that viral entry step is also affected (and the EM pictures do not disprove), this observed entry inhibition needs to be pointed out in the results as well as in discussion.

-Fig1b Y axis (EC50 in uM): what are the minor ticks on the Y-axis? For example, there are 7 ticks between 0.01 and 0.03 uM, which appears like log-scale but they could not be. How was the data processed to mark the axis in the figure?

-The authors mentioned in the rebuttal that for all variants of SARS-CoV-2, key mutations P132S or N117K were present. However, it is not what it shows in Extended Figure 1d (Omicron). This difference in resistance profile for Omicron is curious as the M proteins are almost identical for all variants, but I understand that this is not the main point of the manuscript.

- virucidal incubation: The figure in the rebuttal shows cell controls which have viral RNA copies. The rebuttal only mentioned virus-infected cells with or without compound treatment, which are also in the figure. There is no mention of cell-control, which is presumably control cells without virus infection, but it contains increasing viral RNA over time.

- In revised manuscript, line 305-307: 'computational data supporting a broader zoonotic sarbecovirus coverage' was added in the discussion. What kind of computational data were referred to?

- In revised manuscript, line 325-329: 'In addition, mutations through compound pressure arise over a course of weeks, while treatment of an acute infection would be limited to...' During that first week of treatment, viruses undergo numerous cycles in an individual, and from the perspective of population dynamics, it is not convincing to say viral resistance would not be likely. It would be better to remove line 324 to 327 and leave the last sentence on monitoring.

Referee #4

(Remarks to the Author)

Thanks for asking me to review revised Nature ms 2024-02-03252A. The authors have responded well to most of my

comments with additional experiments.

But there are some unanswered questions which warrant further attention:

1. In this Nature paper, they employed Fab B and Fab E as fiducial markers to 'lock' the M protein in short and long conformation, then claiming that the compound JNJ-9676 can only be found in the short form but not long form M protein after Cryo-EM analysis. Even though the long and short form of M-protein lack structurally distinguishable features, biochemical approaches be used to demonstrate the conformational change. One may consider distinct fluorescence-conjugated Fab B and Fab E to visualize the M protein conformations instead of making such a claim purely on structural data.
2. We should have asked the authors to provide mock-infection control for the section of lung H&E in figure 3e, but bot figure 2e (our typo error, not the fault of the author).
3. Minor: As the authors responded in #7, un-determined Kd value shall be discussed as a limitation of this study and such information might be important to Nature readership.

Referee #5

(Remarks to the Author)

One remaining issue is the register of M residues K162 to T175 in both A and B chains. The register for this region in PDB 8CTK better fits density and chemistry compared to 7VGS (which was used as an initial model and is likely just incorrectly modeled in this strand). Correcting this region should further improve model and fit statistics.

Referee #6

(Remarks to the Author)

All my concerns have been addressed by the authors. I do not have any more comments.

Response to referees

We thank the reviewers for their positive comments and careful review which improved the manuscript. Please find below our point-by-point responses to the reviewers' comments. **Please note that the line numbering in our responses refers to the track changes version of the document.**

Referees' comments:

Referee #2 (Remarks to the Author):

The manuscript from Van Damme and colleagues described a small molecule that targets the coronavirus M protein. CryoEM was used to identify the binding pocket and conformational alteration upon binding to M protein dimers. The alteration prevents release of infectious virus as measure by amounts of RNA in the extracellular media off infected cells. A hamster model was used to show that the small molecule is effective in vivo. The inhibitor is broadly effect against SARS-1 and SAR-2 viruses, variants, and bat and panagolin SARs-like viruses. Overall, the results are very interesting and important in that a unique inhibitor targeting the M protein is described both structurally and functionally.

We appreciate the constructive and insightful comments of referee #2. Please find below a detailed iteration of the changes made. We hope that referee #2 is supportive of publication after these revisions.

Comments

1. Abstract – Lines 53-54 – Based on the data it seems more accurate to inmodicate that JNJ-9676 binds within a pocket formed by the transmembrane domains of M dimerr which results in conformational changes.

The authors agree with the reviewer. Abstract and main text have been updated according to the comment of the reviewer, replacing the term “M protein” with “M protein dimer”. Changes were made on L57, 58, 108, 174, 218, 220, 224, 231 and 543.

2. Fig. 1d graphic annotation should be improved and the orientation corrected. The amino end should be shown as in the lumen and the carboxy tail in the cytoplasm as it is translated at the ER. However, it would be better to refer to orientation in the context of the virus instead (e.g. amino end in the extravirion or outside of virion envelope and carboxy tail as intravirion or inside the envelope. By simply moving the arrows a bit so they do not overlap the TMD, loops and tail (line) illustration and use a color other than black for the stars will improve the figure.

The authors agree with the reviewer's suggestions and have adapted Fig. 1d accordingly. We changed the orientation of the figure to reflect the location of the M protein within the virion. The amino end is now located outside the virion and the carboxy tail is situated inside the virion. We changed the colours of the different domains and replaced the stars indicating the mutations with dots or lines. We avoided overlapping lines with the topology of the M protein.

Fig 1d. now looks as follows:

Legend Extended Data Figure 1d. Membrane topology schematic of the SARS-CoV-2 M protein annotated with in vitro resistance selection (IVRS) mutation residues (indicated in black) and important residues in the CryoEM structure (indicated in blue), the intra-virion domain is indicated in green, the extra-virion domain is indicated in red, and transmembrane domains are indicated as grey boxes.

3. Line 164, page 9 – Q185L should be Q185K.

We thank the reviewer for pointing this out and apologize for the error. It should indeed be Q185K. We have corrected the error on L183.

4. Lines 150-152, Extended Fig. 1 a-b – What concentration of inhibitor was added? It is not clear from the results that egress is inhibited. It seems more likely that virus assembly is inhibited. This could possibly be addressed with an inhibitor dose response experiment or EM of infected cells, but for this report it is sufficient to not specifically indicate that egress is inhibited and leave it at interference with biogenesis.

The authors agree with referee #2 that two terms (egress and assembly) have been used interchangeably. We agree with the assumption that assembly is most likely inhibited instead of egress. This is also consistent with the described functions of the M protein as a key protein in viral assembly, by mediating the concerted recruitment of all virion protein components.

We replaced the term “egress” with “biogenesis” in L160-161:

“The biogenesis of infectious progeny was inhibited completely by JNJ-9676, even if compound treatment was delayed until 5 hours post infection (Extended Data Fig. 1a,b).”

We also added the concentration (5µM) of the inhibitor in the legend of Extended Data Figure 1.

We undertook transmission electron microscopy (TEM) studies to assess the effect of the compound on various phases of the life cycle. While we think these studies are not informative for the general reader of the manuscript, we do wish to share a summary of our findings in this rebuttal document.

TEM examination of ultra-thin sections of resin-embedded, mock-infected A549-hACE2 cells showed that treatment with 0.5 µM and 1 µM of a tool compound (belonging to the same series as JNJ-9676) did not influence the ultrastructure of the A549-hACE2 cells nor the appearance of cell organelles including the nucleus, the mitochondria, the endocytic vesicles and cell membrane. In SARS-CoV-2 infected cells (single round infection, 8h post-infection), representative TEMs showed no effect on the key structures including the reticulovesicular network (RVN) consisting of large numbers of double membrane vesicles, with mitochondria recruited at the edge of this network. In untreated cells, virions were assembled in these replication complexes and ultimately released on the surface of the cells.

Legend: Selected transmission electron micrographs of A549-hACE2 cells infected with SARS-CoV-2 (MOI5, single infection round, 10h post-infection), in the absence of compound treatment, showing formation of the reticulovesicular network (A), virions in the reticulovesicular network (B) and virions outside of the cells (C). Arrows indicate new virions.

In compound (analogue of JNJ-9676)-treated samples, no SARS-CoV-2 virions were observed. Large series of TEM micrographs were systematically recorded of sections of the cells of the different samples, and the presence of the reticulovesicular network was verified as a proxy to determine if cells are infected or not. Since no virions were observed in the compound-treated samples, no statistics on the virus assembly, budding and accumulation in intracytoplasmic vesicles, and exocytotic release of virions at the cell membrane could be provided. This underpins the hypothesis that the compound affects virus morphogenesis.

Legend: Selected transmission electron micrographs of A549-hACE2 cells infected with SARS-CoV-2 (MOI5, single infection round, 10h post-infection), treated with 1 μ M tool compound (analogue of JNJ-9676), showing formation of the reticulovesicular network (A). No virions were observed inside or outside of the cells (B,C).

5. Page 10, lines 187-190 – Fab-B binds the short form of M? The specificity of Fab-B should be included here.

The M protein exists in equilibrium between two conformational states- short and long, this is outlined in lines 106-114 in the manuscript. We agree with the reviewer that the specificity of FABB should be discussed in the manuscript results as well and have included this information – see page 11, line 209-217.

Referee #3 (Remarks to the Author):

A small-molecule SARS-CoV-2 inhibitor targeting the membrane protein

This manuscript reports JNJ-9676, which is a first-in-kind small compound targeting coronavirus M protein. The validity of the novel target is supported by the experiments conducted, and the effectiveness of coronavirus M protein inhibitor in the animal model is very compelling. Identification of a novel druggable target is a valuable addition to the existing collection of antiviral drugs against COVID-19, and the results presented are of significant interest and importance in coronavirus antiviral field. To further improve the quality of the manuscript, addressing specific points below is recommended. There are also many places such as figure legends and materials/methods section where descriptions are lacking or not clear.

We are grateful for the valuable suggestions and feedback from referee #3 which have greatly improved the manuscript. We have made several changes to improve clarity and hope that the manuscript qualifies for publication after these changes.

Major points:

1. It is claimed that JNJ-9676 is broadly active against sarbecoviruses, which was partly supported by the strong inhibitory activity against bat coronaviruses SHC014 and WIV1, which are very close to SARS-CoV-2. It is assumed that all sarbecovirus M proteins share high homology (91%) but only a few sarbecoviruses were included in the analysis. More importantly, adding additional data using more sarbecovirus M proteins would strengthen their claim of broad-sarbecovirus activity.

We understand the reviewer's request to test more sarbecoviruses in vitro. Nevertheless, the set tested and presented in the manuscript were, to our knowledge, the only sarbecoviruses available outside China for in vitro testing in robust assays. We are confident that JNJ-9676 covers other sarbecoviruses, due to the high sequence homology of the M protein. In Extended Data Figure 3, we show that the full sequence of the M protein (panel A) is >87% conserved between known sarbecoviruses. When we zoom in on the binding pocket of JNJ-9676, this sequence homology increases to >90% (panel B). Finally, key mutations such as in residue 117 are completely conserved (panel C).

In addition to the computational analysis, we show in Extended Data Figure 3, and to confirm this statistic, we also downloaded 206 sarbecovirus protein sequences from the InterPro database (InterPro entry: IPR044361, M matrix/glycoprotein, SARS-CoV-like, <https://www.ebi.ac.uk/interpro/entry/InterPro/IPR044361/protein/UniProt>). The only non-conserved residues in the binding pocket are confirmed to correspond to SARS-CoV-2 residues C33 and I87 (as shown in Extended Data Figure 3). The binding pocket sequence identity across sarbecoviruses is also confirmed as >90%. The caption of Extended Data Figure 3 has been updated to reflect this extended analysis of conservation. Based on these computational analyses it is highly likely that JNJ-9676 covers other zoonotic sarbecoviruses. We included these points in the discussion section of the manuscript (L305-307).

2. Regarding viral resistance

It would be nice to include the EC50 changes from SARS-CoV-2 over passages, which would help to understand how fast resistance emerges in a live virus setting. The Materials/methods described that EC50s were determined at each passage number, so it would be easy to add a figure/table. Combined with Fig 1F table that shows fold changes due to single mutations introduced to recombinant mutant virus, and Ext. Fig1d that shows mutant virus population % over passages, a new figure/table will provide a more comprehensive view on the resistance barrier to this compound.

The authors thank the reviewer for this suggestion to highlight the timing of emerging resistance. For this purpose, we have added additional panels to Extended Figure 1d to demonstrate the resistance dynamics of the IVRS performed with the original Wuhan, delta and Omicron variants of SARS-CoV-2. The resistance dynamic change curve shows that the resistance building up is reflected by full-breakthrough concentration increasing over the passing process. Noteworthy, for the Omicron variants, the resistance is building up slower than the original strain and delta variants meaning that only after passage 30, a big resistance shift occurred.

The following text was inserted in the manuscript on L170-171: *“The resistance dynamic change curve (Extended Data Fig. 1d) shows that the generation of resistance mutations continues to increase the concentration of JNJ-9676 needed for full breakthrough.”*

Ext. Fig 1d and g: The recombinant SARS-CoV-2 carrying each mutation show very similar replication efficiency (g), but the predominant mutations (d) vary among different SARS-CoV-2 strains and replicates. What would be the potential explanation for this observation?

In Extended Data Figure 1d we show the emergence of mutations, in most cases, multiple mutations in the M protein under compound pressure across the different passages. It is true that at different times, different mutations arise in different strains, however, key mutations such as P132S or N117K which cause the biggest shifts in EC₅₀ values are present in all strains. The interplay between the different

mutations results in resistant escape mutants. We then genetically introduced single point mutations of the most frequently observed mutations as well as key residues identified using cryoEM. These site-directed mutants underwent fitness tests and plaque assays. We show that single mutations do not impact fitness nor plaque size (Extended Data Figure 1g,h). Although in the manuscript, we opted to focus on single mutations observed in resistant viruses and mutants in key residues based on cryoEM, we did produce also mutants with combinations of multiple mutations i.e. M:P132S+N117K, M:A85S+N117K+L29F and M:P132S+L138I+S173P+Q185K. The fitness curve and plaque assay are shown below. As evident from the replication curves, only M:P132S+L138I+S173P+Q185K showed a decrease in fitness. This is confirmed by the plaque assay in which this quadruple mutant showed considerably smaller plaques.

SARS-CoV2 SDM replication kinetics

Ext. Fig 1d and g: please add how these experiments were conducted and analyzed in detail.

For Figure 1d, the description of the resistance selection is now included in the Supplementary Methods at Line 237- 259.

“In vitro resistance selection assay

In vitro resistance selection (IVRS) experiments were performed with SARS-CoV-2 B.1, Delta B.1.617.2 and Omicron B.1.1.529-BA.1 under pressure of increasing concentrations of JNJ-9676 in a 96-well plate. A549-hACE2 cells were seeded in assay medium at a density of 5,000 cells/well (3 days of incubation) or 3,000 cells/well (4 days of incubation) in a 96-well plate and were immediately inoculated with SARS-CoV-2 at MOI of 0.01 or 0.02 based on MOI optimization experiments in the presence of compound or DMSO for virus and cell control conditions. Compounds were added in nine 2-fold dilutions starting at a concentration of 4 μ M for JNJ-9676. Three replicates were performed per compound. After incubation, every well was scored microscopically for CPE and RT-qPCR was performed regularly as an additional control for the CPE readout. The virus was passaged on new A549-hACE2 compound plates until full infection as determined by CPE scoring was reached for the three replicates. The supernatants of the highest compound concentrations with virus-induced CPE were collected for RNA extraction together with a virus control that was passaged on the plates, and together with the original virus stocks used for infection. These RNA samples were used for the preparation of a library pool for Illumina next generation sequencing, based on the workflow of the NuGen Trio RNASeq™ kit (Tecan Genomics). FastQ data were analyzed to determine the amino acid changes observed as compared to the reference SARS-CoV-2 sequence (i.e., sequence of the virus inoculation stock) and the frequency of those substitutions. Only samples with an average coverage of more than 1,000 reads per position were proceeded for further bioinformatic analysis. A read frequency threshold of 15% was applied for variant calling. Mutations from compound-resistant viruses were filtered through by comparing to the virus control samples to filter out any potential cell adaptation related mutations.”

For Figure 1g, the experiment details were included in the Supplementary Methods at Line 292-319.

“Replication kinetic studies and plaque assay for site-directed mutant viruses

Virus replication kinetics of the recombinant virus strains was assessed in A549-hACE2 cells. Cells were pre-seeded at 12,500 cells/well into 96-well plates. 24 h after seeding, cells were infected with SARS-CoV-2 wild-type (WT), recombinant wild type M protein virus: SARS-CoV-2-recombWT, L29F, A85S, L90W, N117K, P132S, Q185K, W55F, M91K, S99A, N117K+P132S, N117K+L29F+A85S and P132S+L138I+S173P+Q185K, at an MOI of 0.1. Viruses were washed away after 1 h incubation. Viral supernatants were collected for RT-qPCR analysis and cells were fixed for staining of viral spike protein and dsRNA at indicated timepoints (1, 4, 8, 24, 48, 72 hpi). The kinetics profile for various recombinant strains of SARS-CoV-2 in aspects of dsRNA, spike and viral RNA abundancy throughout the duration of the experiment were compared with WT recombinant virus.

Plaque assay of the recombinant virus strains was performed in Vero E6 cells. Cells were pre-seeded at 25000 cells/well into 12-well plates. 24h after seeding, cells were infected SARS-CoV-2 wild-type (WT), recombinant wild type M protein virus: SARS-CoV-2-recombWT, L29F, A85S, L90W, N117K, P132S, Q185K, W55F, M91K, S99A, N117K+P132S, N117K+L29F+A85S and P132S+L138I+S173P+Q185K, with 250 μ L corresponding serial dilution (=inoculum). Cells were incubated with virus for 1 hour at 37°C. In the meantime, the agarose overlay was prepared (50% of 1% UltraPure™ Low Melting Point Agarose (ThermoFisher) + 50% of 2X medium (Gibco™ MEM (Temin's modification) (2X), 4% fetal bovine serum (Biowest), and 0.08% gentamicine (Gibco)). After 1 hour of incubation the inoculum was removed from the wells and 1mL of the agarose overlay was added to each well. Once the agarose overlay solidified, plates were placed upside down in the incubator for 3 days at 37°C, 5% CO₂. Cells were fixated with 2

ml/well of 4% formaldehyde (Polysciences Inc.) at 3 dpi and the plates were kept 2 hours under the laminar flow for soaking. Afterwards the agarose overlay and formaldehyde was removed and the plates were washed with water. Next, pre-made 0.1% crystal violet (Sigma) was added to the plates and incubated for 10 minutes. After staining, the plates were washed with PBS. The plaques size was visually assessed and compared between wild-type viruses and recombinant mutant viruses.”

In Ext.Fig1d, multiple mutations were found in 100% of viral population at some passage numbers, which suggest multiple mutations with >50-100-fold increase in EC50 emerge quickly during 10-15 passages. Please add discussion on its clinical implications as antiviral drugs.

To address this comment, we added the following to the discussion on L322-329:

“It has yet to be clarified how the emergence of resistance mutation in vitro will translate to in a clinical setting. In case of nirmatrelvir (Paxlovid), a subset of mutations occurring in vitro was found in patients. However, it is unclear what the impact on non-immunocompromised patients is as escape mutants often have compromised viral fitness. In addition, mutations through compound pressure arise over a course of weeks, while treatment of an acute infection would be limited to 5–7 days while the virus is quickly cleared. Nevertheless, the emergence of resistance mutations will need to be monitored in the clinic across human predicted doses and an extended period of time.”

3. Time of Addition assay

Ext. Fig 1a/b, p150-152: Viral replication was inhibited when the compound is incubated with cells for 0-1 hr (viral entry step). Does it mean that the inhibitor blocks viral entry or uncoating as well by inhibiting generating mature virions? Or could there be another mechanism for this inhibition?

This is indeed an interesting observation that we have also made. It is possible that the mode of action may be dual and that JNJ-9676 impacts both uncoating and the biogenesis of new virions. As uncoating is a poorly understood process, it is unclear at this stage what may happen in these early stages. Based on transmission electron microscopy (as shown above), we know that viral entry and the formation of replication organelles (double membrane vesicles) are not inhibited by the compound. Based on Extended Data Fig. 1a/b, where we specifically looked at the production of infectious viruses, we believe that the main driver of compound activity is the inhibition of virion assembly, congruent with the main function of the M protein. In addition, we performed experiments to test whether JNJ-9676 could interfere with the virion prior to infection; however, no virucidal effect could be observed.

4. Mouse study: Are the extended Fig 5C table and the Fig 3b-h (log10 lung virus copies and lung TCID50) from the same experiment? The numbers in the table and the figures do not match.

We thank the reviewer for spotting these discrepancies. Indeed, these data are from the same experiment and thus the following corrections were made in the table:

- Infectious virus in the lung at 75mg/kg
- infectious virus in the lung at 25mg/kg

The values in the table reflect the mean (average) of the vehicle minus the mean (average) of the tested condition.

Minor point:

Fig 1a is confusing – please revise the figure for clarification.

We show the chemical structure in Fig 1a, we have added the chemical name ((S) N (3 cyanophenyl)-5 (4 (difluoro(phenyl)methyl)phenyl)-6 methyl-4 oxo-4,5,6,7 tetrahydropyrazolo[1,5 a]pyrazine-3 carboxamide) to the legend.

Fig. 3e – histopath photos have lower resolution. Photos of higher resolution and magnification would be helpful.

We have exported the photos at the highest resolution possible and re-assembled the figure.

Fig. 3 – histopath data from post-exposure experiment is not included.

We do not have histopathology images from the post-exposure experiments. In these experiments all viral parameters were the same as in pre-exposure experiments and thus similar histopathology findings are expected.

Fig. 3 – The inhibitor was given BID but the figure shows 8hr interval.

Indeed, we dosed the animals each day in the morning and then again 8 hours later, repeating this every day. We have added the following clarifying sentence in the legend of Fig 3a and f: “Animals were dosed twice daily at 8 AM and 4 PM.”

Fig. 3 - title sarbecovirus needs to be changed to SARS-CoV-2.

We have corrected the title of Figure legend 3 as suggested.

It is better to move the extended Fig 5C table to Fig 3g and h.

We agree with the reviewer’s suggestion and have moved the table from Extended Data Figure 5 to Figure 3i.

Please consider adding the resistance data to main manuscript (Extended fig1d and add EC50 values)

While we appreciate that the referee’s suggestion to highlight the resistance data in the main manuscript, we prefer to keep the resistance data in the extended data as the initial goal of in vitro resistance experiments is to unravel the potential target of JNJ-9676. Based on the observed accumulation of mutations in the M protein, we gained an initial idea about the viral target, enabling us to generate additional experiments to unravel the mechanism of action at a molecular level. Therefore, we ran ASMS and NanoDSF studies and finally generated the cryoEM structures which are prominently featured in the manuscript. The importance of resistance mutations will be further evaluated during clinical development of the compound. However, if the referee prefers to include this information in the main manuscript, we can still make this change.

Treatment dose of mg/kg BID – 75 mg/kg BID is per oral treatment or per day (in total)? Please specify.

To improve clarity, we have added the following text to each figure legend with in vivo data: “The doses reflect the amount of compound given each administration”.

Ext. Fig 1b. What is the purpose of including 10⁻¹ and 10⁻² virus dilutions in the figure?

We appreciate this feedback and upon reflection, the 10⁻¹ and 10⁻² virus dilutions do not add a lot of value. We have changed this in panel b of Extended Data Fig. 1.

Referee #4 (Remarks to the Author):

This international team synthesized a small molecule JNJ-9676 which is an analogue of a compound series with nanomolar activity against SARS-CoV-2 identified previously from chemical library screening. Vero E6 cell culture assay confirmed its low EC50 against many representative strains of Sarbecoviruses while it is less potent against other non-sarbeco-coronaviruses. Its potent activity is also confirmed in primary human nasal epithelial cells grown in air-liquid interface. Time of addition experiments suggested that it interferes with the production of virions. Serial passage of SARS-CoV-2

B1 strain in the presence of gradually increasing concentration of this compound selects for escape mutations mainly found in M protein which resides near the dimer interface. Reverse genetically engineered M viral mutants showed 43 to 145 fold increased phenotypic resistance to this compound. Drug target interaction between JNJ-9676 with M protein were clearly demonstrated with two different binding assays including offline affinity selection-mass spectrometry and nano differential scanning fluorimetry. Cryogenic EM controlled by a fiducial marker antibody fragment showed that SARS-CoV-2 M protomer A complexed better with the compound. The compound adopts a rotated L shaped configuration and binds to an induced pocket formed by TM2 and TM3 of protomer A and TM1 of protomer B with significant conformational change to a novel conformational state. Finally oral JNJ-9676 reduced virus titre and histopathological damage in the lung of hamsters challenged by SARS-CoV-2 when the compound is given before and even 48 hours after virus inoculation. The authors concluded that M protein is an attractive novel antiviral target and JNJ-9676 represents a potential drug candidate.

We thank referee #4 for the comments and believe that implementing the changes below will significantly improve the manuscript, hopefully making it fit for publication.

Weakness

1. The finding of broad inhibitor against structural M or N protein of virus is not a new concept. Both have been reported for influenza viruses. These seminal papers should be discussed. Broad Spectrum Anti-Influenza Agents by Inhibiting Self-Association of Matrix Protein 1 – PMC <https://www.ncbi.nlm.nih.gov/pmc/articles/PMC5004101/> (nih.gov); Identification of influenza A nucleoprotein as an antiviral target | Nature Biotechnology <https://www.nature.com/articles/nbt.1638>

Add

We appreciate the suggestion of referee #4 to mention these findings. We have included a reference to the publications in the discussion. We added the following sentence on L348-349: *“Targeting conserved structural proteins has been described before when influenza matrix and nucleoprotein were found druggable^{61,62}.”*

2. The authors claim that JNJ-9676 was an analogue of a compound series that was identified in the screening campaign published in J Med Virol in reference 37. But there is no information in that paper which refers to this compound series which is now represented by the analogue JNJ9676.

We thank the reviewer for looking at our prior publication and we agree that there is ambiguity in what is currently written. Although the same methodology was used as in reference 37, in fact, JNJ-9676 was not identified in that screen. It was identified in an unpublished follow-up screen with a diversity set of compounds from the Janssen proprietary library.

We have included these clarifications in the manuscript (L132-138):

“Following the same methodology as in a high-throughput screen using phase one passed structures³⁷, a follow up screening campaign for small molecule inhibitors of SARS-CoV-2 was performed in VeroE6-eGFP cells using a diversity set of compounds from the Janssen proprietary library. JNJ-9676 is a representative analogue (Fig. 1a) of a compound series that was identified in the screen.”

3. No positive control (such as nirmatrelvir even though it has a different antiviral mechanism) was included in the cell culture or hamster challenge experiments.

We agree with the reviewer’s claim to include a positive control to benchmark both the in vitro and in vivo data in this manuscript. We have added the following data to the manuscript:

- Fig 1b: a multitude of assays was performed to assemble this figure, each using an internal positive control, however, positive controls differed depending on the assay. We have, however, added the EC50 value of nirmatrelvir in SARS-CoV-2 infected A549-hACE2 cells as a reference.
- Fig 1c: we have added the dose-response curve of nirmatrelvir in the ALI cultures as a reference and mentioned this as well in the text (L152-153)
- Fig 3b, c, d: at the time of this experiment, molnupiravir was the reference molecule of choice and thus the pre-exposure model was carried out with molnupiravir as a positive control. However, we could not determine a statistically significant decline in viral parameters. The figure was updated including the positive control. In Extended Data Figure 5b we show the bodyweight differences for all groups as well. A textual reference was made on L258-260.
- Fig 3g,h: in the fast-evolving landscape of SARS-CoV-2 research, we moved to nirmatrelvir as positive control molecule. We have adapted Figure 3g, h to include these data, Extended Data Figure 5i was also adapted. A textual reference was made on L268-269.
- Extended Data Figure 5: these studies were carried out with nirmatrelvir as a positive control molecule. We have adapted the figure to show these data.

4. The relative amount of short M and long M after JNJ-9686 treatment of transfected cell line for the production of virus like particles should be assayed to demonstrate the mechanism of inhibition.

We appreciate this feedback, however, visualization of relative amounts of long and short M proteins and their organization in the virion is extremely challenging due to small size of the M protein (25kDa) and the lack of distinguishable features between two states in their intra-virion domains. Even with the purified M protein, we needed the bound FAbs to aid with particle alignment to obtain our structure. In addition, we have attempted mechanism of inhibition of studies by generating minimal VLPs containing only M, N, S, and E proteins. However, this minimal VLP system did not recapitulate the complex virus assembly process.

5. An experiment should be performed to see if the compound can reduce the infectivity of cell free SARS-CoV-2 virion (with long M) after in vitro incubation with JNJ-9676.

We agree with the reviewer's suggestion. Due to the nature of the compound, we had already addressed this question using a virucidal experiment in which JNJ-9676 was pre-incubated at 5 μ M for 0.5 h in a SARS-CoV-2 B1 (early Belgian strain) viral inoculum of 200,000 TCID₅₀/mL. This corresponds to a concentration of >220x the EC₅₀ against SARS-CoV-2 B1 on A549-hACE2 cells. Subsequently, the compound and virus were diluted to theoretical concentrations of 1 nM compound and 20 TCID₅₀/mL virus. The diluted SARS-CoV-2 B1 inoculum, which does not contain enough compound to elicit an eminent antiviral effect, as well as the mock-treated inoculum were used to infect Vero E6 cells. Both visual CPE scoring and vRNA monitoring using qPCR were performed at 24, 48 and 72hpi. No difference between mock and JNJ-9676-treated condition could be observed in the qPCR readout nor was there a difference in CPE scoring between both conditions. Hence, we concluded that JNJ-9676 does not have a virucidal effect once the virion has been successfully formed. The figure below shows the results of the vRNA qPCR.

Figure legend. Assessment of virucidal effect using qPCR: vRNA copies in function of time of supernatant fractions from A549-hACE2 cells infected with virus inoculum that was pretreated with JNJ-9676 or mock.

6. How the recombinant viruses with mutations were generated are not clear. By simply refer to the references are not enough. The authors should show the plaque pictures of each recombinant viruses. It is still not clear why resistant mutations, such as the T91 E protein which is consistently positive in 2 out of 3 columns, were not chosen for reverse genetic investigation (Extended Data Table 2).

We appreciate the need for a clearer description of the generation of recombinant viruses in the materials and methods. We have added more details in the Supplementary Materials on L266-L285. The following details have been added:

“The primers for CPER reaction are listed in the Extended Data Table 5 (p59). Mutations were introduced via site- directed mutagenesis using either Q5[®] Site-Directed Mutagenesis Kit or Phusion PCR. In brief, the SARS-CoV-2 Belgian strain (B.1) genetic backbone was divided across seven different pUC57 plasmids. These plasmids (pUC57_F1-pUC57_F7) cover the complete SARS-CoV-2 genome. Mutations were introduced in the plasmid that aligned with the region of interest of the virus genome. SDM primers were designed with the NEBaseChanger tool (<https://nebasechanger.neb.com/>) and inverse PCR was performed using the Q5[®] Site-Directed Mutagenesis Kit (New England Biolabs, MA, USA) following the manufacturer’s instructions. After DpnI (Invitrogen, MA, USA) digestion of methylated fragments, 5’ ends of PCR products were phosphorylated with polynucleotide kinase (Invitrogen, MA, USA) and ultimately circularized by in vitro ligation with T4 DNA ligase (Invitrogen, MA, USA). Mutated plasmids were propagated in E. coli DH10B T1R cells, following ampicillin selection of positive clones and validation by Sanger sequencing. The mutated plasmids were used as PCR template in the CPER reaction for generation of the full-length circular site mutant recombinant construct. Only the high-frequency accumulated site mutations in the M protein were introduced as single mutations (L29F, A85S, L90W, N117K, P132S, and Q185K). The following top 3 single mutations (W55F, M91K, S99A) were also introduced after resolving the M protein-compound complex cryo-EM structure. For introducing these mutations, a simplified fusion PCR was used by using overlapping primers (Extended Data Table 5) covering the desired mutation followed by the same procedure of CPER reaction.”

We have carried out plaque assays for all the recombinant viruses and are showing the images in Extended Data Figure 1h. The largest plaques were observed for the wild type virus, between site directed mutants (including the recombinant wild type) there are no discernable differences in plaque

size between any of the single mutants. This underlines the similar replication kinetics shown in Extended Data Figure 1g. Text reflecting these observations was inserted in L185-186.

The experimental details were added in the Supplementary Methods in Line 303-319.

“Plaque assay of the recombinant virus strains was performed in Vero E6 cells. Cells were pre-seeded at 25000 cells/well into 12-well plates. 24h after seeding, cells were infected SARS-CoV-2 wild-type (WT), recombinant wild type M protein virus: SARS-CoV-2-recombWT, L29F, A85S, L90W, N117K, P132S, Q185K, W55F, M91K, S99A, N117K+P132S, N117K+L29F+A85S and P132S+L138I+S173P+Q185K, with 250 µL corresponding serial dilution (=inoculum). Cells were incubated with virus for 1 hour at 37°C. In the meantime, the agarose overlay was prepared (50% of 1% UltraPure™ Low Melting Point Agarose (ThermoFisher) + 50% of 2X medium (Gibco™ MEM (Temin's modification) (2X), 4% fetal bovine serum (Biowest), and 0.08% gentamicine (Gibco)). After 1 hour of incubation the inoculum was removed from the wells and 1 mL of the agarose overlay was added to each well. Once the agarose overlay solidified, plates were placed upside down in the incubator for 3 days at 37°C, 5% CO₂. Cells were fixed with 2 ml/well of 4% formaldehyde (Polysciences Inc.) at 3 dpi and the plates are kept 2 hours under the laminar flow for soaking. Afterwards the agarose overlay and formaldehyde was removed and the plates were washed with water. Next, pre-made 0.1% crystal violet (Sigma) was added to the plates and incubated for 10 minutes. After staining, the plates were washed with PBS. The plaques size was visually assessed and compared between wild-type viruses and recombinant mutant viruses.”

As we detected a variety of mutations, we selected the most frequent resistance mutations occurring only in the M protein that were observed in at least 3 independent replicates for further genetic cloning.

7. The exact binding affinity Kd value between JNJ-9676 and M protein, wild type and mutant, was not determined.

We have attempted to measure the binding affinity between JNJ-9676 and M protein via surface plasma resonance (SPR) using a Biacore instrument. These studies were not successful due to inefficient capture of M protein on the surface and inability to generate a stable baseline. In addition, we attempted to generate biotinylated M protein with a C-terminal Avi tag to address this issue, but that was not successful either due to poor quality of the avi tagged protein and the low efficiency of biotinylation.

8. The pharmacokinetic data of JNJ-9676 cannot be found in the file (Please indicate it as the reviewer may have missed it).

We would like to guide the referee to Extended Data Figure 6a where we show hamster PK of JNJ-9676 at 50mg/kg PO, 5mg/kg PO and 1mg/kg IV.

Minor comments

1. Line 52: SARS-CoV2 shall be SARS-CoV-2

We thank the reviewer for spotting this error and corrected it in L55.

2. mock-infection control is lacking in figure 2e.

We are not sure which figure the reviewer is referring to. In Figure 2e, we describe the molecular interactions of JNJ-9676 with the M protein.

Recommendation

The authors should address all the weakness before publication

Referee #5 (Remarks to the Author):

Van Damme, et al. report on the discovery of a small molecule inhibitor of SARS-CoV-2 assembly targeting the M protein. This is exciting work, well performed, and clearly and concisely presented. It provides important insights into the role of M protein in coronavirus assembly and strong evidence supporting its suitability as a novel target for therapeutics. I have only several minor concerns and suggestions for the manuscript.

We are very grateful for the comments and suggestions of referee #5 which we have addressed point-by-point below. We hope these revisions are sufficient to make our manuscript suitable for publication.

1. Did the authors attempt to determine a structure or assess interaction of JNJ-9676 with M in the long conformation? If not, how was it decided to pursue only a cryo-EM structure of M in the short conformation bound to the small molecule?

In our CryoEM study of the SARS-CoV-2 M protein complexed with JNJ-9676, we strategically employed FabB and FabE fragments as fiducial markers in cryo-EM analysis. This approach led to achieving a global nominal resolution of 3.1 Å for the SARS-CoV-2 M protein complexed with JNJ-9676 and FabE and 3.06 Å for the complex with FabB.

However, within the 3.1 Å resolution SARS-CoV-2 M protein complexed with JNJ-9676 and FabE map, we could not identify the ligand density corresponding to the JNJ-9676, and the M protein maintained its long-form conformation. In contrast, the 3.06 Å resolution SARS-CoV-2 M protein complexed with JNJ-9676 and FabB unveiled a clearly defined compound density for JNJ-9676.

We have updated the manuscript accordingly in L209-L217.

2. A more thorough explanation of the various escape mutants identified would be an interesting addition. For example, can they be grouped into those expected to directly occlude binding through steric or electrostatic repulsion versus those rather expected to bias the conformational equilibrium between long and short conformations and differentially colored in Figure 1e?

Following the reviewer's suggestion, out of the set of mutants inducing a potency shift, we identified two mutants (P132S and Q185K) that could potentially alter the conformational equilibrium between long and short form. Mutants with direct interactions with JNJ-9676 are shown in Figure 1e and the corresponding interactions are discussed in the main text at 174-178. Detailed ligand interactions are also shown in Fig. 2e,f.

3. Figure 2f should be improved for clarity. Perhaps change to make the overlaid apo structure gray or some other color?

We have made the suggested changes in Figure 2f.

4. The clear difference in resolvability of the two M subunits is interesting. Can the authors elaborate on how JNJ-9676 was docked and refined into the second subunit with weaker map features?

We included a new figure (Extended Data Fig. 4e) to illustrate the docking and refinement process of JNJ-9676 into the second subunit, specifically above protomer B. We have updated the manuscript accordingly (L221).

5. No large-scale issues are evident in the model, but the structure shows overall relatively poor refined geometry for the resolution and map quality (particularly in the Ramachandran plot and clash statistics). The authors should consider additional refinement with either an improved starting model or with increased geometry weight in real space refinement. If issues are confined to the less well-defined subunit, weak NCS constraints could be applied as an alternative approach.

Thank you for your comment. The main issues were limited to the less well-defined subunit. After performing additional refinement with increased geometry weight in real-space refinement, we observed improvements in the Ramachandran plot (Outliers: 0.17% to 0.00%; Allowed: 15.70% to 10.46%; Favoured: 84.14% to 89.54%) and clash statistics (9.77 to 7.25). We have updated the table in the manuscript accordingly.

6. This is subjective, but I would be inclined to describe the conformation of M captured as a distorted version the previously described short conformation rather than a “new” form. Conformational differences are seemingly well described by a simple coordinated movement of TM1s required for docking of the short arm of the JNJ-9676 “L” including the benzyl group and associated movement of the tops of TM2-3. Whether this is a populated alternative conformation or intermediate in the absence of JNJ-9676 is unclear.

We believe that JNJ-9676 traps the M protein in an alternative conformation that limits its plasticity to move between short and long forms to form viable virus. We agree that this JNJ-9676 bound conformation is more like short form than the long form, however with the significant shifts of TM’s 1, 2 and 3 and the loss of symmetric dimer that was observed in the short form, we were hesitant to call this a “distorted short form” but rather a third alternative conformation that is incompatible with proper

virus assembly. Nevertheless, should the reviewer strongly feel this should be named as a distorted short form, we are willing to make the adjustments.

Referee #6 (Remarks to the Author):

In this investigation, the authors discovered a new SARS-CoV-2 inhibitor, called JNJ-9676, by a phenotype based high-throughput screening. The biological target of this compound was identified as the M protein by an in vitro resistance selection (IVRS) assay. JNJ-9676 demonstrated nanomolar antiviral activity against various sarbecovirus types including SARS-CoV2, SARS-CoV, and others in in vitro antiviral assays. Cryo-EM was used to determine the complex structure of JNJ-9676 – the M protein, which showed that JNJ-9676 binds to a new pocket in the M protein's transmembrane domain, stabilizing the M protein in an altered conformational state between its long- and short-forms, hence preventing the release of infectious virus. This compound also displayed good in vivo anti-viral activity in preclinical animal models. From the reviewer's point of view, this study is a very preliminary work, needing to be further improved.

We wish to thank referee #6 for the insightful comments and hope that with the changes reflected in this document, we have sufficiently improved the manuscript to qualify for publication.

My major concerns:

1) How was JNJ-9676 discovered? More detailed description is needed, although structure-activity relationship (SAR) analysis of this series compounds is not necessary.

We appreciate the reviewer's interest in the discovery method of the molecule. We initially used a VeroE6-eGFP SARS-CoV-2 infection model to run a Phase One Passed Structures screen (reference: Chiu, W. et al. Development and optimization of a high-throughput screening assay for in vitro anti-SARS-CoV-2 activity: evaluation of 5676 Phase 1 passed structures. J. Med. Virol. 94, 3101-3111 (2022)). Subsequently, we used the same assay to screen a diversity set of the Janssen compound collection. We identified the initial hit in the series in this screen. However, to obtain JNJ-9676, significant chemical modifications were made. The evolution of the compound series is currently being written up in a Medicinal Chemistry journal article. To give more clarity on the discovery of the initial hit in the series, we added the following paragraph on L132-138: *"Following the same methodology as in a high-throughput screen using phase one passed structures³⁷, a follow up screening campaign for small molecule inhibitors of SARS-CoV-2 was performed in VeroE6-eGFP cells using a diversity set of compounds from the Janssen proprietary library. JNJ-9676 is a representative analogue (Fig. 1a) of a compound series that was identified in the screen."*

2) JNJ-9676 was not well characterized. Only nanoDSF was used to measure the binding of JNJ-9676 with the M protein. A ΔT_m value of 0.9°C indicates a poor binding affinity, at least not good. Why did the authors not use more other approaches to confirm this binding, such as ITC, SPR?

We understand that a ΔT_m value of 0.9°C is a subtle change, however, in combination with the Affinity Selection Mass Spectrometry (Extended Data Figure 2), which confirmed binding, we believed this was sufficient evidence of direct target engagement. In addition, we carried out photo-affinity labeling of a close analogue of JNJ-9676. Using this tagged molecule, we were able to determine once more that JNJ-9676 binds the M protein. Built on this evidence, we generated cryoEM structures which are described in this manuscript.

SPR was attempted, but unsuccessful. As using membrane proteins in SPR is notoriously difficult and given the clear binding of JNJ-9676 in the M protein cryoEM structure, we did not further this line of investigation.

3) The author only tested the antiviral activity of JNJ-9676 in hamsters. It is better to evaluate it in humanized transgenic mice. In addition, it is also necessary to explore the effects of the M protein inhibitor on the immune system during viral infection.

The hamster model was our chosen in vivo model because of the natural susceptibility of hamsters to SARS-CoV-2. Early in the pandemic, this model was frequently used (Muñoz-Fontela et al., 2020; Chan et al., 2020; Roberts et al., 2005; Sia et al., 2020) and most of these studies were carried out prior to the validation of mouse models. We hope the reviewer appreciates the efforts we have made to fully characterize the molecule in terms of in vivo efficacy (pre-exposure dose response, post-exposure at different time points).

We did not explore the effects of JNJ-9676 on hamster immunity given that the compound is a direct acting antiviral. While a role in cellular immunity has been attributed to the M protein e.g., through interaction with TLRs and IRFs (Kasuga et al., 2021), interference with RIG-I signaling (Zheng et al., 202) and even B-cell response (Martin et al., 2021), most of these pathways are poorly understood. Hamster immunity is not well described and the translatability to humans largely unknown. However, we have evaluated histopathological changes in lungs of hamsters and these findings are described in this manuscript. We see a clear reduction in inflammatory markers in the lungs of hamsters upon treatment with JNJ-9676 indicating that the compound contributes to viral clearance.

Chan, J. F.-W., et al. (2020). "Simulation of the Clinical and Pathological Manifestations of Coronavirus Disease 2019 (COVID-19) in a Golden Syrian Hamster Model: Implications for Disease Pathogenesis and Transmissibility." Clinical Infectious Diseases **71**(9): 2428-2446.

Kasuga, Y., et al. (2021). "Innate immune sensing of coronavirus and viral evasion strategies." Experimental and Molecular Medicine **53**(5): 723-736.

Martin, S., et al. (2021). "SARS-CoV-2 integral membrane proteins shape the serological responses of patients with COVID-19." iScience **24**(10): 103185.

Muñoz-Fontela, C., et al. (2020). "Animal models for COVID-19." Nature **586**(7830): 509-515.

Roberts, A., et al. (2005). "Severe Acute Respiratory Syndrome Coronavirus Infection of Golden Syrian Hamsters." Journal of Virology **79**(1): 503-511.

Sia, S. F., et al. (2020). "Pathogenesis and transmission of SARS-CoV-2 in golden hamsters." Nature **583**(7818): 834-838.

Zheng, Y., et al. (2020). "Severe acute respiratory syndrome coronavirus 2 (SARS-CoV-2) membrane (M) protein inhibits type I and III interferon production by targeting RIG-I/MDA-5 signaling." Signal Transduction and Targeted Therapy **5**(1): 299.

Figure Legend. Preliminary study in K18 hACE2 mice dosed with 32mg/kg BID Ensitrelvir and 50mg/kg BID JNJ-9676 showing viral RNA levels in the lung and trachea, infectious virus in the lung, histopathology scoring of the lung and immunohistochemistry evaluation of the lung.

4) *In vitro and in vivo antiviral experiments lack a positive control, making it difficult to objectively compare the effects of JNJ-9676. It is recommended to use paxlovid (nirmatrelvir) as a positive control, and show whether JNJ-9676 has some advantages.*

We appreciate the need for a positive control to benchmark both the in vitro and in vivo data in this manuscript. We have added the following data to the manuscript:

- Figure 1b: a multitude of assays was performed to assemble this figure, each using an internal positive control, however, positive controls differed depending on the assay. We have, however, added the EC₅₀ value of nirmatrelvir in SARS-CoV-2 infected A549-hACE2 cells as a reference.
- Figure 1c: we have added the dose-response curve of nirmatrelvir in ALI cultures as a reference and mentioned this as well in the text (L152-153)
- Figure 3b, c, d: at the time of this experiment, molnupiravir was the reference molecule of choice and thus the pre-exposure model was carried out with molnupiravir as a positive control. However, we could not determine a statistically significant (although visible) decline in viral parameters. The figure was updated including the positive control. In Extended Data Figure 5b we show the bodyweight differences for all groups as well. A textual reference was made on L258-260.

- Figure 3g,h: in the fast-evolving landscape of SARS-CoV-2 research, we moved to nirmatrelvir as positive control molecule. We have adapted Figure 3g, h to include these data as well, Extended Data Figure 5i was also adapted. A textual reference was made on L268-269.
- Extended Data Figure 5: these studies were carried out with nirmatrelvir as a positive control molecule. We have adapted the figure to show these data.

5) As an oral medication, please evaluate the pharmacokinetic properties of JNJ-9676 in related species and provide detailed parameters.

The PK JNJ-9676 was evaluated in hamster (in vivo efficacy species) and further in rat, dog and cynomolgus macaque in light of calculating human anticipated oral bioavailability and half-life. We have added this information to the manuscript as Extended Data Table 6 and to the text on L245-247.

	Rat	Hamster	Dog	Cyno
Clearance (ml/min/kg)	5.2	9.0	<1.5	1.9
Volume of distribution (L/kg)	8.4	7.9	8.2	5.1
Plasma half-life (hours)	19	13	>40	35
Oral bioavailability (%)	>70	>50	>80	>30

Minors:

6) What is the solvent used in the in vivo assay?

We used 100% PEG as a solvent for in vivo studies (please see also L619-621).

7) How about the target selectivity of JNJ-9676?

As a part of the target deconvolution of the compound series, we performed Photo Affinity Labeling experiments where a labeled probe and a tool compound (analogue of JNJ-9676) was used. Based on these experiments, the M protein was the only viral protein to be engaged. In addition, human proteins did not reproducibly engage the probe in uninfected conditions. Therefore, we consider that JNJ-9676 is selective for M protein as the target.

8) Please provide the compound characterization data of JNJ-9676, including 1H NMR and 13C NMR, HPLC spectra, etc.

1D ¹H, 1D ¹³C, 2D ¹H-¹H COSY, 2D ¹H-¹³C HSQC, and 2D ¹H-¹³C HMBC NMR spectra were added to the Supplementary Methods from page 5-10 with clarifying text on L96-L120.

Response to referees

We thank the reviewers for their positive comments on our revised version of the manuscript.

Referees' comments:

Referee #1 (Remarks to the Author):

Manuscript 2024-02-03252A from Van Damme and colleagues describes a small molecule that targets the coronavirus M protein. Results with the inhibitor in an animal model are promising to provide a new therapeutic antiviral against a new coronavirus target. The revised manuscript addresses the major comments on the initial submission. The manuscript overall is improved based on the responses to earlier questions and suggestions from the reviewers.

We thank the referee for the critical review and the positive feedback.

Referee #3 (Remarks to the Author):

The authors has put a great deal of efforts to address the points raised by the reviewers in general. However, there are some points that need further clarification as listed below. No additional data or experiments are deemed necessary with this revision in my view, but more careful perusal of the manuscript by the authors is required to make sure all figures and data sets are free of errors.

We thank the referee for the critical review and positive assessment.

-Entry inhibition with JNJ-9676 (0-1 hr incubation): The authors mentioned in the rebuttal that the EM pictures showed viral entry is not affected by the compound because there is no virus in the reticulovesicular network or released virions at 10 hr post infection. Although it is possible that virion assembly is the major key mechanism of viral infection by the compound, the EM pictures do not serve as evidence that virion uncoating/entry is not affected. Endosomal pathway (2-6 hr post infection) needs to be monitored in ACE2-expressing cells for failure of viral uncoating/entry (as accumulated virus particles), and I wonder if the authors also looked at these subcellular locations. It is okay to leave this for future study, but as the time-of-addition clearly indicates that viral entry step is also

affected (and the EM pictures do not disprove), this observed entry inhibition needs to be pointed out in the results as well as in discussion.

We appreciate the thorough review of the EM images. We indeed did not observe any differences in the very first steps of the viral life cycle (entry, localization of the virus in DMV), however, the reviewer is correct that we cannot exclude that another early event may be disrupted. The endolysosomal pathway was not investigated in detail in the EM studies at this stage.

We reflected the possibility of a dual mode of action in the results section on L157-158 (“*This suggests that JNJ-9676 may interfere with early events as well as the biogenesis of infectious viral progeny.*”).

-Fig1b Y axis (EC50 in uM): what are the minor ticks on the Y-axis? For example, there are 7 ticks between 0.01 and 0.03 uM, which appears like log-scale but they could not be. How was the data processed to mark the axis in the figure?

In the y-axis of Fig1b, we use antilog10 number format. In this format, we have the option to visualize 0-8 minor ticks, we choose to reflect the highest granularity (8 ticks).

-The authors mentioned in the rebuttal that for all variants of SARS-CoV-2, key mutations P132S or N117K were present. However, it is not what it shows in Extended Figure 1d (Omicron). This difference in resistance profile for Omicron is curious as the M proteins are almost identical for all variants, but I understand that this is not the main point of the manuscript.

We have done Omicron IVRS experiment with JNJ-9676 as well as with other analogues of the series. It is indeed true that with JNJ-9676, we did not observe P132S or N117K mutations; however, we did find these mutations back in experiments with close analogues. While for JNJ-9676 in Omicron, these mutations do not seem to develop as readily, for the compound series these residues are still of high importance.

- virucidal incubation: The figure in the rebuttal shows cell controls which have viral RNA copies. The rebuttal only mentioned virus-infected cells with or without compound treatment, which are also in the figure. There is no mention of cell-control, which is presumably control cells without virus infection, but it contains increasing viral RNA over time.

We ran a cell control (uninfected cells) in the same experiment. We assessed the raw RT-qPCR data and observed that the Cp values from the cell controls at time points 48 and 72 were higher

than the Cp value obtained for the lowest concentration of the standard. Therefore, we concluded that these values were outside of the limit of quantification (LOQ). Therefore, it was not included in further calculations to construct the 'log₁₀ copies/mL in function of time plot'. The result is a bar plot in which no value is shown for the cell controls, whilst bars for JNJ-87009676 and the DMSO control remained unaltered. For completeness, we adapted the figure for this response letter.

-Figure 1

-In revised manuscript, line 305-307: 'computational data supporting a broader zoonotic sarbecovirus coverage' was added in the discussion. What kind of computational data were referred to?

We adapted the sentence slightly for readability (L293-295): *"Furthermore, computational data show that the full sequence of the M protein is >87% conserved between known sarbecoviruses, and the binding pocket of JNJ-9676 shows >90% homology, thereby supporting a broader zoonotic sarbecovirus coverage."*

The computational data meant here pertain to Ext. Data Figure 3. In this figure, we show a computational analysis of the M protein and the compound binding pocket in the M protein to highlight the high degree of conservation of the target.

- In revised manuscript, line 325-329: 'In addition, mutations through compound pressure arise over a course of weeks, while treatment of an acute infection would be limited to...' During that first week of treatment, viruses undergo numerous cycles in an individual, and from the perspective of population dynamics, it is not convincing to say viral resistance would not be likely. It would be better to remove line 324 to 327 and leave the last sentence on monitoring.

We agree with the reviewer. To limit speculation, we deleted lines 324-327 and left only the final sentence.

Referee #4 (Remarks to the Author):

Thanks for asking me to review revised Nature ms 2024-02-03252A. The authors have responded well to most of my comments with additional experiments.

We thank the referee for the critical review and the positive feedback.

But there are some unanswered questions which warrant further attention:

1. In this Nature paper, they employed Fab B and Fab E as fiducial markers to 'lock' the M protein in short and long conformation, then claiming that the compound JNJ-9676 can only be found in the short form but not long form M protein after Cryo-EM analysis. Even though the long and short form of M-protein lack structurally distinguishable features, biochemical approaches be used to demonstrate the conformational change. One may consider distinct fluorescence-conjugated Fab B and Fab E to visualize the M protein conformations instead of making such a claim purely on structural data.

Thank you for your feedback. We agree that the absence of compound density in the Fab E bound M protein complex does not provide enough evidence for the claim. However, we also performed affinity selection mass spectrometry (ASMS) studies with M protein bound to Fab B and E (Extended Figure 2h). In the ASMS experiment, JNJ-9676 recovery is significantly higher when it was incubated with M-protein:Fab B complex compared to M protein alone or M-protein:Fab E complex indicating JNJ-9676 binds better to M-protein:Fab B complex. These ASMS data, in combination with our observation in the Cryo-EM results, were the basis for our claim. Even with a significant effort to set-up a surface plasma resonance (SPR) system to study these interactions in detail, we were not successful due to insufficient capture of the protein and the inability to generate a stable baseline. This is now noted as a limitation of this study in the discussion (L319-321).

2. We should have asked the authors to provide mock-infection control for the section of lung H&E in figure 3e, but bot figure 2e (our typo error, not the fault of the author).

For this particular study, no mock infected animals were included. The pathologist scored the histopathology findings based on what is normally observed in a healthy lung (absence of infiltrations and inflammation). However, since these images are not available, we removed part of the sentence on L246-248, now stating: “JNJ-9676 significantly reduced the cumulative histopathological lung score at 75 mg/kg BID ($P=0.0015$) and 25 mg/kg BID ($P=0.0093$) (Fig. 3d,e).”

3. Minor: As the authors responded in #7, un-determined Kd value shall be discussed as a limitation of this study and such information might be important to Nature readership.

We added the following sentence in the discussion on L319-321: “A limitation of this study is that the affinity of JNJ-9676 to purified M protein could not be determined using surface plasma resonance (SPR) due to inefficient capture of M protein on the surface and inability to generate a stable baseline.”

Referee #5 (Remarks to the Author):

One remaining issue is the register of M residues K162 to T175 in both A and B chains. The register for this region in PDB 8CTK better fits density and chemistry compared to 7VGS (which was used as an initial model and is likely just incorrectly modeled in this strand). Correcting this region should further improve model and fit statistics.

Thank you for the valuable suggestion. We re-evaluated the region (K162-175) and made corrections, which have further improved the quality of our model. Specifically, the Ramachandran favored value increased from 84.14% to 90.21%, and the CaBLAM outlier value improved from 7.38% to 5.84%.

Referee #6 (Remarks to the Author):

All my concerns have been addressed by the authors. I do not have any more comments.

We thank the referee for the critical review and positive assessment.